# GENERATIVE ENTROPIC NEURAL OPTIMAL TRANSPORT TO MAP WITHIN AND ACROSS SPACES

## ABSTRACT

Learning measure-to-measure mappings is a crucial task in machine learning, featured prominently in generative modeling. Recent years have witnessed a surge of techniques that draw inspiration from optimal transport (OT) theory. Combined with neural network models, these methods collectively known as *Neural OT* use optimal transport as an inductive bias: such mappings should be optimal w.r.t. a given cost function, in the sense that they are able to move points in a thrifty way, within (by minimizing displacements) or across spaces (by being isometric). This principle, while intuitive, is often confronted with several practical challenges that require adapting the OT toolbox: cost functions other than the squared-Euclidean cost can be challenging to handle, the deterministic formulation of Monge maps leaves little flexibility, mapping across incomparable spaces raises multiple challenges, while the mass conservation constraint inherent to OT can provide too much credit to outliers. While each of these mismatches between practice and theory has been addressed independently in various works, we propose in this work an elegant framework to unify them, called *generative entropic neural optimal transport* (GENOT). GENOT can accommodate any cost function; handles randomness using conditional generative models; can map points across incomparable spaces, and can be used as an *unbalanced* solver. We evaluate our approach through experiments conducted on various synthetic datasets and demonstrate its practicality in single-cell biology. In this domain, GENOT proves to be valuable for tasks such as modeling cell development, predicting cellular responses to drugs, and translating between different data modalities of cells.

## 1 INTRODUCTION

Mapping a probability distribution onto another is a ubiquitous challenge in machine learning, with many implications in the field of generative modeling. Optimal transport (OT) has arisen in a few years as a major purveyor of tools to better address these challenges, both in theory and practice. The focus of OT lies on finding maps that can effectively transform a distribution of matter onto another, by minimizing a certain notion of cost (Santambrogio, 2015). Originally rooted in physics, the application of OT to large-dimensional problems arising in machine learning and sciences has necessitated various modifications and adaptations. Starting with solvers that can solve approximate matching problems at large scales (Cuturi, 2013; Peyré et al., 2016; Scetbon et al., 2021; 2022), a recent plethora of OT-inspired training approaches for neural networks has emerged (Makkuva et al., 2020; Korotin et al., 2020; Asadulaev et al., 2022; Fan et al., 2020; Uscidda & Cuturi, 2023; Lipman et al., 2023; Tong et al., 2020; 2023b). As an illustration of this overall trend, the applications of OT to single-cell genomics have evolved from advanced matching problems (Schiebinger et al., 2019; Demetci et al., 2022), towards neural-based approaches that can, for instance, predict the response of cells to various perturbations (Bunne et al., 2021; 2022). Our goal in this paper is to address the various challenges that still stand in the way of applying OT to the most pressing scientific tasks.

**From Linear to Quadratic Neural OT Maps.** Optimal transport is primarily used through the Kantorovich problem to put in correspondence distributions taking values in the same space $\mathcal{X}$, pend the existence of a cost $c(x, y)$ for any two points $x, y \in \mathcal{X}$. Most of the theory is available in that regime, notably for simpler costs such as the squared Euclidean distance (Santambrogio, 2015, §1.3). We refer to such problems as *linear* OT problems. Yet, more challenging applicative scenarios sought by practitioners involve source and target distributions that do *not* live in the same

space, e.g. $\mathcal{X}$ and $\mathcal{Y}$ have differing dimensions, as in (Demetci et al., 2022). The challenge in that case is that no cost functions are known, requiring the use of quadratic losses (Mémoli, 2011; Sturm, 2020), yielding the so-called Gromov-Wasserstein (GW) problem. While theory is far more scarce in these regimes, practitioners expressed major interest in that flexibility, going as far as proposing, with the Fused Gromov-Wasserstein (FGW) distance, a tool that blends both linear and quadratic approaches (Vayer et al., 2018), as in (Klein et al., 2023; Lange et al., 2023; Nitzan et al., 2019; Zeira et al., 2022). There exists, however, to our knowledge, only one formulation of a neural quadratic OT method, which is limited to learning deterministic maps for the inner product costs and whose training procedure involves a min-max-min optimization procedure (Nekrashevich et al., 2023).

**From Deterministic to Stochastic Maps.** The classic (Monge) deterministic map can lack flexibility in practice, both at estimation and inference time. In the quadratic case, that map may not exist (Dumont et al., 2022). Practitioners may favor, instead, stochasticity, which would account naturally for instance, for the non-determinism of cell evolutions (Elowitz et al., 2002). Stochastic formulations can also produce a conditional distribution that can be used to quantify uncertainty. In the discrete setting, this property is fulfilled by entropy-regularized OT (EOT) (Cuturi, 2013).

**Flexibility in Mass Conservation.** In numerous real-world applications, the data acquisition process can be error-prone, resulting in outliers. To mitigate this, unbalanced OT (UOT) formulations that can discard observations have been proposed (Frogner et al., 2015; Chizat et al., 2018; Séjourné et al., 2021), with numerous applications to generative modeling (Balaji et al., 2020; Yang & Uhler, 2019) and single-cell genomics (Schiebinger et al., 2019; Eyring et al., 2022; Lübeck et al., 2022).

**Contributions.** We propose a flexible neural OT framework that satisfies all requirements above:

- We propose the first method to compute neural EOT couplings in both Kantorovich and GW settings by fitting stochastic maps to their conditional distributions (Prop. 3.1) using conditional flow matching (Lipman et al., 2023) as a building block. In particular, GENOT works with any cost function between samples.
- By showing that solving an unbalanced EOT problem is equivalent to solving a balanced one between re-weighted measures (Prop. 3.2) that can be estimated consistently (Prop. 3.3), we introduce U-GENOT to solve unbalanced EOT problems.
- We extend (U-)GENOT to solve the (unbalanced) entropic Fused GW problem (§ 3.3). To our knowledge, GENOT is the first neural OT method to solve a continuous Fused GW problem.
- We demonstrate the applicability of GENOT in various single-cell biology problems. In particular, we (i) quantify lineage branching events in the developing mouse pancreas, (ii) predict cellular responses to drug perturbations along with a well-calibrated uncertainty estimation, and (iii) introduce a novel method to translate ATAC-seq data to RNA-seq data.

## 2 BACKGROUND

**Notations.** We consider throughout this work two compact subsets $\mathcal{X} \subset \mathbb{R}^p, \mathcal{Y} \subset \mathbb{R}^q$, referred to as the source and the target domain, respectively. In general, $p \neq q$. The sets of positive measures and probability measures on $\mathcal{X}$ are denoted by $\mathcal{M}^+(\mathcal{X})$ and $\mathcal{M}_1^+(\mathcal{X})$, respectively. For $\pi \in \mathcal{M}^+(\mathcal{X} \times \mathcal{Y})$, we denote its marginals by $\pi_1 := p_1 \sharp \pi$ and $\pi_2 := p_2 \sharp \pi$. Then, for $\mu \in \mathcal{M}^+(\mathcal{X}), \nu \in \mathcal{M}^+(\mathcal{Y})$, $\Pi(\mu, \nu)$ is the set of probability measures with respective marginals $\mu$ and $\nu$, i.e. $\Pi(\mu, \nu) = \{\pi : \pi_1 = \mu, \pi_2 = \nu\} \subset \mathcal{P}(\mathcal{X} \times \mathcal{Y})$. We define $\frac{d\mu}{d\nu}$ to be the relative density of $\mu$ w.r.t. $\nu$ and write $\mu = \frac{d\mu}{d\nu} \cdot \nu$ accordingly. For $\rho, \gamma \in \mathcal{M}^+(\mathcal{X})$, $\mathrm{KL}(\rho|\gamma) = \int_{\mathcal{X}} \log(\frac{d\rho}{d\gamma}) \, d\rho - \int_{\mathcal{X}} d\gamma + \int_{\mathcal{X}} d\rho$.

### 2.1 ENTROPIC OPTIMAL TRANSPORT

**The Entropic Kantorovich Problem.** Let $c : \mathcal{X} \times \mathcal{Y} \to \mathbb{R}$ be a cost function, $\mu \in \mathcal{M}_1^+(\mathcal{X}), \nu \in \mathcal{M}_1^+(\mathcal{Y})$ and $\varepsilon \geq 0$. The entropy-regularized OT problem reads

$$\min_{\pi \in \Pi(\mu, \nu)} \int_{\mathcal{X} \times \mathcal{Y}} c(\mathbf{x}, \mathbf{y}) \, d\pi(\mathbf{x}, \mathbf{y}) + \varepsilon \mathrm{KL}(\pi | \mu \otimes \nu) \,. \qquad \text{EK}$$

A solution $\pi_\varepsilon^\star$ of (EK) always exists. With $\varepsilon = 0$, we recover the classical Kantorovich (1942) problem. When $\varepsilon > 0$, the optimal coupling $\pi_\varepsilon^\star$ is unique. If $\mu$ and $\nu$ are discrete, (EK) can be solved with the Sinkhorn algorithm (Cuturi, 2013).

**The Entropic Gromov-Wasserstein Problem.** As opposed to considering an *inter-domain* cost defined on $\mathcal{X} \times \mathcal{Y}$, the entropic Gromov-Wasserstein problem is concerned with seeking couplings based on *intra-domain* cost functions $c_{\mathcal{X}} : \mathcal{X} \times \mathcal{X} \to \mathbb{R}$ and $c_{\mathcal{Y}} : \mathcal{Y} \times \mathcal{Y} \to \mathbb{R}$:

$$\min_{\pi \in \Pi(\mu,\nu)} \int_{(\mathcal{X} \times \mathcal{Y})^2} |c_{\mathcal{X}}(\mathbf{x}, \mathbf{x}') - c_{\mathcal{Y}}(\mathbf{y}, \mathbf{y}')|^2 \, \mathrm{d}\pi(\mathbf{x}, \mathbf{y}) \, \mathrm{d}\pi(\mathbf{x}', \mathbf{y}') + \varepsilon \mathrm{KL}(\pi|\mu \otimes \nu). \qquad \text{EGW}$$

With $\varepsilon = 0$, we recover the Gromov-Wasserstein problem (Mémoli, 2011). As in the Kantorovich setting, using $\varepsilon > 0$ comes with favorable computational properties, since for discrete $\mu,\nu$, we can solve (EGW) with a mirror-descent scheme based on the Sinkhorn algorithm (Peyré et al., 2016).

**Unbalanced Extensions.** The EOT formulations presented above can only handle measures with the same total mass. Unbalanced optimal transport (UOT) (Liero et al., 2018; Chizat et al., 2018) lifts this constraint by penalizing the deviation of $p_1 \sharp \pi$ to $\mu$ and $p_2 \sharp \pi$ to $\nu$ with a divergence. Using the KL divergence and introducing $\lambda_1, \lambda_2 > 0$ controlling how much mass variations are penalized as opposed to transportation, the unbalanced extension of (EK) seeks a measure $\pi \in \mathcal{M}^+(\mathcal{X} \times \mathcal{Y})$:

$$\min_{\pi \in \mathcal{M}^+(\mathcal{X} \times \mathcal{Y})} \int_{\mathcal{X} \times \mathcal{Y}} c(\mathbf{x}, \mathbf{y}) \, \mathrm{d}\pi(\mathbf{x}, \mathbf{y}) + \varepsilon \mathrm{KL}(\pi|\mu \otimes \nu) + \lambda_1 \mathrm{KL}(\pi_1|\mu) + \lambda_2 \mathrm{KL}(\pi_2|\nu). \qquad \text{UEK}$$

This problem can be solved efficiently in a discrete setting using a variant of the Sinkhorn algorithm (Frogner et al., 2015; Séjourné et al., 2023a). Analogously, the GW formulation (EGW) also admits an unbalanced generalization, which reads

$$\min_{\pi \in \mathcal{M}^+(\mathcal{X} \times \mathcal{Y})} \int_{(\mathcal{X} \times \mathcal{Y})^2} |c_{\mathcal{X}}(\mathbf{x}, \mathbf{x}') - c_{\mathcal{Y}}(\mathbf{y}, \mathbf{y}')|^2 \, \mathrm{d}\pi(\mathbf{x}, \mathbf{y}) \, \mathrm{d}\pi(\mathbf{x}', \mathbf{y}')$$

$$+ \varepsilon \mathrm{KL}^{\otimes}(\pi|\mu \otimes \nu) + \lambda_1 \mathrm{KL}^{\otimes}(\pi_1|\mu) + \lambda_2 \mathrm{KL}^{\otimes}(\pi_2|\nu), \qquad \text{UEGW}$$

where $\mathrm{KL}^{\otimes}(\rho|\gamma) = \mathrm{KL}(\rho \otimes \rho|\gamma \otimes \gamma)$. This can also be solved using an extension of Peyré et al. (2016)'s scheme introduced by Séjourné et al. (2023b). For both unbalanced problems (EK) and (UEGW), instead of directly selecting $\lambda_i$, we introduce $\tau_i = \frac{\lambda_i}{\lambda_i + \varepsilon}$ s.t. we recover the hard marginal constraint for $\tau_i = 1$, when $\lambda_i \to +\infty$. We write $\tau = (\tau_1, \tau_2)$ accordingly.

## 2.2 CONDITIONAL FLOW MATCHING

Provided a prior distribution $\rho_0 \in \mathcal{M}_1^+(\mathbb{R}^d)$ and a time-dependent vector field $v_t$, one can define a probability path $(p_t)_{t \in [0,1]}$ starting from $\rho_0$ using the flow $(\phi_t)_{t \in [0,1]}$ induced by the ODE

$$\frac{\mathrm{d}}{\mathrm{d}t} \phi_t(\mathbf{z}) = v_t(\phi_t(\mathbf{z})), \quad \phi_0(\mathbf{z}) = \mathbf{z}, \qquad (1)$$

by setting $p_t = \phi_t \sharp \rho_0$. In that case, we say that $v_t$ generates the path $p_t$ through the flow $\phi_t$. Continuous Normalizing Flows (Chen et al., 2018) model the vector field with a neural network $v_{t,\theta}$, leading to a deep parametric model of the flow, which is trained to match a terminal condition defined by a target distribution $p_1 = \rho_1 \in \mathcal{M}_1^+(\mathbb{R}^d)$. (Conditional) Flow Matching (CFM) (Lipman et al., 2023) is a simulation-free technique to train CNFs by constructing probability paths between individual data samples $\mathbf{z}_0 \sim \rho_0$, $\mathbf{z}_1 \sim \rho_1$, and minimizing the loss

$$\mathcal{L}_{\mathrm{CFM}}(\theta) = \mathbb{E}_{t \sim \mathcal{U}([0,1]), Z_0 \sim \rho_0, Z_1 \sim \rho_1} [\|v_{t,\theta} (tZ_0 + (1-t)Z_1) - (Z_1 - Z_0)\|_2^2]. \qquad (2)$$

If this loss is 0, then $v_{t,\theta}$ generates a probability path between $\rho_0$ and $\rho_1$, i.e. the induced flow satisfies $\phi_1 \sharp \rho_0 = \rho_1$ (Lipman et al., 2023)[Theorem 1]. To sample from $\rho_1$, we solve the ODE (1) with $\mathbf{z}_0$ sampled from $\rho_0$, and therefore $\phi_1(\mathbf{z}_0)$ is a sample from $\rho_1$

## 3 GENERATIVE ENTROPIC NEURAL OPTIMAL TRANSPORT

In this section, we introduce GENOT, a method to learn EOT couplings by learning their conditional distributions. In § (3.1), we first focus on the balanced OT case, when the source and the target measures have the same mass, and show that GENOT can solve (EK) or (EGW). Second, in § (3.2), we extend GENOT to the unbalanced setting by loosening the conservation of mass constraint and defining U-GENOT, which can be used to solve problems (UEK) and (UEGW). Finally, in § 3.3, we highlight that GENOT also adresses a fused problem, combining (EK) and (EGW).

### 3.1 Learning Entropic Optimal Couplings with GENOT

Let $\mu \in \mathcal{M}_1^+(\mathcal{X})$, $\nu \in \mathcal{M}_1^+(\mathcal{Y})$ and $\pi_\varepsilon^\star$ be an EOT coupling between $\mu$ and $\nu$, which can be a solution of problem (EK) or (EGW). The measure disintegration theorem yields

$$\mathrm{d}\pi_\varepsilon^\star(\mathbf{x}, \mathbf{y}) = \mathrm{d}\pi_{\varepsilon,1}^\star(\mathbf{x})\,\mathrm{d}\pi_\varepsilon^\star(\mathbf{y}|\mathbf{x}) = \mathrm{d}\mu(\mathbf{x})\,\mathrm{d}\pi_\varepsilon^\star(\mathbf{y}|\mathbf{x})\,. \tag{3}$$

Knowing $\mu$, we can hence fully describe $\pi_\varepsilon^\star$ via the conditional distributions $(\pi_\varepsilon^\star(\cdot|\mathbf{x}))_{\mathbf{x}\in\mathcal{X}}$. The latter are also of great practical interest, as they provide a way to transport a source sample $\mathbf{x} \sim \mu$ to the target domain $\mathcal{Y}$; either *stochastically* by sampling $\mathbf{y}_1, ..., \mathbf{y}_n \sim \pi_\varepsilon^\star(\cdot|\mathbf{x})$, or *deterministically* by averaging over conditional samples:

$$T_\varepsilon(\mathbf{x}) := \mathbb{E}_{Y \sim \pi_\varepsilon^\star(\cdot|\mathbf{x})}[Y] = \mathbb{E}_{(X,Y) \sim \pi_\varepsilon^\star}[Y | X = \mathbf{x}]\,. \tag{4}$$

Moreover, we can compute any statistic of $\pi_\varepsilon^\star(\cdot|\mathbf{x})$ to assess the uncertainty surrounding this prediction. In the following, we elaborate on our approach for calculating these conditional distributions.

**Noise Outsourcing.** Let $\rho \in \mathcal{M}_1^+(\mathcal{Z})$ be an atomless distribution on an arbitrary Borel space $\mathcal{Z}$, refer to as the noise. The noise outsourcing lemma (Kallenberg, 2002) states that there exists a collection of maps $\{T^\star(\cdot|\mathbf{x})\}_{\mathbf{x}\in\mathcal{X}}$ with $T^\star(\cdot|\mathbf{x}) : \mathcal{Z} \to \mathcal{Y}$ s.t. for each $\mathbf{x}$ in the support of $\mu$, $\pi_\varepsilon^\star(\cdot|\mathbf{x}) = T^\star(\cdot|\mathbf{x})\sharp\rho$, i.e. if $Z \sim \rho$, then $Y = T^\star(Z|\mathbf{x}) \sim \pi_\varepsilon^\star(\cdot|\mathbf{x})$. Each $T^\star(\cdot|\mathbf{x})$ generates a distribution from a point $\mathbf{x}$, by "outsourcing" the noise vectors $Z \sim \rho$. We refer to $\{T^\star(\cdot|\mathbf{x})\}_{\mathbf{x}\in\mathcal{X}}$ as a collection of *optimal conditional generators* since they generate the conditional distributions of $\pi_\varepsilon^\star$. Conversely, noise outsourcing provides a way to define neural couplings $\pi_\theta$ by parameterizing their conditional generators $\{T_\theta(\cdot|\mathbf{x})\}_{\mathbf{x}\in\mathcal{X}}$ with neural networks. To obtain $\pi_\theta \approx \pi_\varepsilon^\star$, we then need $T_\theta(\cdot|\mathbf{x})$ to generate $\pi_\varepsilon^\star(\cdot|\mathbf{x})$ by outsourcing the noise $\rho$, for any source sample $\mathbf{x}$ in the support of $\mu$.

**Learning the Conditional Generators.** In the following, we learn a collection of maps $\{T_\theta(\cdot|\mathbf{x})\}_{\mathbf{x}\in\mathcal{X}}$ fitting the constraint $T_\theta(\cdot|\mathbf{x})\sharp\rho \approx \pi_\varepsilon^\star(\cdot|\mathbf{x})$ for any $\mathbf{x}$ in the support of $\mu$. Instead of directly modeling $T_\theta(\cdot|\mathbf{x})$ with a neural network, we employ the CFM framework discussed in § 2.2. To that end, we first set $\mathcal{Z} = \mathbb{R}^q$ and the noise $\rho = \mathcal{N}(0, I_q)$. Recall that $q$ is the dimension of the target domain $\mathcal{Y}$. Then, we parameterize each $T_\theta(\cdot|\mathbf{x})$ implicitly as the flow induced by a neural vector field $v_{t,\theta}(\cdot|\mathbf{x}) : \mathbb{R}^q \to \mathbb{R}^q$. Namely $T_\theta(\cdot|\mathbf{x}) = \phi_1(\cdot|\mathbf{x})$ where $\phi_t(\cdot|\mathbf{x})$ solves

$$\frac{\mathrm{d}}{\mathrm{d}t}\phi_t(\mathbf{z}|\mathbf{x}) = v_{t,\theta}(\phi_t(\mathbf{z}|\mathbf{x})|\mathbf{x}), \quad \phi_0(\mathbf{z}|\mathbf{x}) = \mathbf{z}. \tag{5}$$

We stress that while $\mathbf{x} \in \mathcal{X} \subset \mathbb{R}^d$, the flow from $\rho$ to $\pi_\varepsilon^\star(\cdot|\mathbf{x})$ is defined on $\mathbb{R}^q \supset \mathcal{Y}$. Hence, we can map samples *within* the same space when $p = q$, but also *across* incomparable spaces when $p \neq q$. In particular, this allows us to solve the Gromov-Wasserstein problem (EGW). Thus, for each $\mathbf{x}$, we optimize $v_{t,\theta}(\cdot|\mathbf{x})$ by minimizing the CFM loss (2) with source $\rho$ and target $\pi_\varepsilon^\star(\cdot|\mathbf{x})$, i.e.

$$\mathbb{E}_{t\sim\mathcal{U}([0,1]), Z\sim\rho, Y\sim\pi_\varepsilon^\star(\cdot|\mathbf{x})}[\|v_{t,\theta}\left((1-t)Z + tY|\mathbf{x}\right) - (Y - Z)\|_2^2]\,. \tag{6}$$

Averaging for all $\mathbf{x}$ in the support of $\mu$ and using Fubini's Theorem, we arrive at the GENOT loss

$$\mathcal{L}_{\mathrm{GENOT}}(\theta) = \mathbb{E}_{t\sim\mathcal{U}([0,1]), Z\sim\rho, X\sim\mu, Y\sim\pi_\varepsilon^\star(\cdot|X)}[\|v_{t,\theta}\left((1-t)Z + tY|X\right) - (Y - Z)\|_2^2]\,. \tag{7}$$

GENOT is a well-posed loss in the sense that, in the idealized asymptotic, infinite sample setting, assuming neural network architectures that are expressive enough, one could provably recover the original entropic coupling (and its conditional distributions), as shown in the Proposition below.

**Proposition 3.1** (Well-posedness of GENOT Loss). *Suppose that $\mathcal{L}_{\mathrm{GENOT}}(\theta) = 0$. Then the flows $\{\phi_1(\cdot|\mathbf{x})\}_{\in\mathcal{X}}$, induced by the velocity fields $\{v_{t,\theta}(\cdot|\mathbf{x})\}_{\in\mathcal{X}}$, are a collection of optimal conditional generators. Namely, for $\mathbf{x}$ in the support of $\mu$, $Z \sim \rho$ and $Y = \phi_1(Z|\mathbf{x})$ denoting the solution of the ODE (5), then $Y \sim \pi_\varepsilon^\star(\cdot|\mathbf{x})$, therefore this ideal conditional vector field $v_{t,\theta}$ recovers $\pi_\varepsilon^\star$.*

We optimize the *sample-based* GENOT loss, using mini-batches. This involves (i) estimating a *discrete coupling* $\hat{\pi}_\varepsilon$ from samples $\mathbf{x}_1, \ldots, \mathbf{x}_n$ from $\mu$ and $\mathbf{y}_1, \ldots, \mathbf{y}_n$ from $\nu$, and (ii) sampling its discrete conditional distributions, to recover paired samples. Algorithm 1 details the overall procedure, using noise and time samples. GENOT can be thought of as a conditional CFM model: For each $\mathbf{x}$, using CFM, train a conditional vector field $v_{t,\theta}(\cdot|\mathbf{x})$ to generate $\pi_\varepsilon^\star(\cdot|\mathbf{x})$ from noise $\rho$.

**Bias and Mini-batches.** Quantifying non-asymptotically the bias resulting from minimizing a sample-based GENOT loss, and *not* its population value, is a challenging task. The OT-inspired

generative modeling literature (Genevay et al., 2019a; Salimans et al., 2018; Uscidda & Cuturi, 2023; Tong et al., 2023b) mentions recurrently this aspect, see also (Fatras et al., 2021). Analyzing these non-asymptotic properties becomes even harder when considering conditional mappings across spaces, in a GW setting, as we do here since discrete solvers do not return, in general, a globally optimal sample-based coupling. Yet, our goal in this paper is *not* to estimate a deterministic Monge map or vector field (Benamou & Brenier, 2000), we target explicitly the *entropic* coupling. In that sense, using a large $\varepsilon$ does help, because of two qualitative factors: In the Kantorovich problem, all statistical recovery rates that relate to entropic costs (Genevay et al., 2019b; Mena & Niles-Weed, 2019) or maps (Rigollet & Stromme, 2022), *for a fixed $\varepsilon > 0$*, have a far more favorable regime, with a parametric rate that dodges the curse of dimensionality. While these statistics are less studied for the GW case, Rioux et al. (2023) have recently shown that for sufficiently large $\varepsilon$, GW becomes a convex problem, making optimization more stable. Qualitatively, large $\varepsilon$ will be therefore useful on both statistical and computational fronts. The simpler alternative of independent sampling boils down effectively to an infinite $\varepsilon$.

**GENOT Addresses Any Cost.** Thanks to Prop. 3.1, we can use GENOT to solve (EK) and (EGW) problems. In both cases, we make no assumptions on the cost functions, and only need to evaluate these costs to estimate $\pi_\varepsilon^\star$. In particular, we can use costs that are implicitly defined and whose evaluation requires a non-differentiable sub-routine. For instance, recent works have proposed using the geodesic distance on the data manifold as cost, which can be approximated from samples by considering the shortest path distance on the $k$-nn graph induced by the Euclidean distance (Demetci et al., 2022). Using data-driven cost functions is crucial for many applications as in some single-cell genomic tasks (Huguet et al., 2022; Klein et al., 2023).

### 3.2 U-GENOT: EXTENSION TO THE UNBALANCED SETTING

**Re-Balancing the UOT Problems.** In its standard form, GENOT respects marginal constraints, so it cannot directly tackle unbalanced formulations (UEK) or (UEGW). We show that such unbalanced problems can be *re-balanced*. (Lübeck et al., 2022; Yang & Uhler, 2019) introduced previously these ideas in the Monge map estimation setting, namely, in a static and deterministic setup. Besides this important conceptual difference, various aspects differentiate further our approach: (Lübeck et al., 2022) define unbalanced couplings between *mapped* instances of the source measure using an ICNN (significantly closer to the target), and vice-versa, whereas we directly target the unbalanced coupling between source and target for any cost; (Yang & Uhler, 2019) provide an asymmetric formulation that only considers modulations of the source distribution to the target distribution. Our method stems from the fact that, for Kantorovich and GW cases, we can show that the unbalanced EOT coupling $\pi_{\varepsilon,\tau}^\star$ between $\mu \in \mathcal{M}^+(\mathcal{X})$ and $\nu \in \mathcal{M}^+(\mathcal{Y})$ solves a balanced EOT problem between its marginals, which are re-weighted versions of $\mu$ and $\nu$ that have the same mass.

**Proposition 3.2** (Re-Balancing the unbalanced problems.). *Let $\pi_{\varepsilon,\tau}^\star$ be an unbalanced EOT coupling, solution of* (UEK) *or* (UEGW) *between $\mu \in \mathcal{M}^+(\mathcal{X})$ and $\nu \in \mathcal{M}^+(\mathcal{Y})$. We note $\tilde{\mu} = p_1 \sharp \pi_{\varepsilon,\tau}^\star$ and $\tilde{\nu} = p_2 \sharp \pi_{\varepsilon,\tau}^\star$ its marginals. Then, in both cases, $\tilde{\mu}$ (resp. $\tilde{\nu}$) has a density w.r.t $\mu$ (resp. $\nu$) i.e. it exists $\eta, \xi : \mathbb{R}^d \to \mathbb{R}^+$ s.t. $\tilde{\mu} = \eta \cdot \mu$ and $\tilde{\nu} = \xi \cdot \nu$. Moreover, $\tilde{\mu}$ and $\tilde{\nu}$ have the same mass and*

1. *(Kantorovich) $\pi_{\varepsilon,\tau}^\star$ solves the balanced problem* (EK) *between $\tilde{\mu}$ and $\tilde{\nu}$ with the same $\varepsilon$.*

2. *(Gromov-Wasserstein) Provided that $c_\mathcal{X}$ and $c_\mathcal{Y}$ are conditionally positive (or conditionally negative) kernels (see Def. B.1), $\pi_{\varepsilon,\tau}^\star$ solves the balanced problem* (EGW) *between $\tilde{\mu}$ and $\tilde{\nu}$ with $\varepsilon' = m(\pi_{\varepsilon,\tau}^\star) \varepsilon$, where $m(\pi_{\varepsilon,\tau}^\star) = \pi_{\varepsilon,\tau}^\star(\mathcal{X} \times \mathcal{Y})$ is the total mass of $\pi_{\varepsilon,\tau}^\star$.*

**Remark.** In various experimental settings, $\mu$ and $\nu$ have mass 1 and we impose one of the two hard marginal constraints, for instance on $\mu$, by setting $\tau_1 = 1$. Then $\tilde{\nu}$ has also mass 1 and $m(\pi_{\varepsilon,\tau}^\star) = 1$, so we keep the same regularization strength $\varepsilon$ by re-balancing (UEGW).

**Learning the Coupling and the Re-Weightings Simultaneously.** Thanks to Prop. 3.2, we aim to (i) learn a balanced EOT coupling between $\tilde{\mu}$ and $\tilde{\nu}$ along with (ii) the re-weighting functions $\eta, \xi$. The latter are crucial since they model the creation and destruction of mass. We do both simultaneously by adapting the GENOT procedure. More formally, we seek to optimize the U-GENOT loss

$$\mathcal{L}_{\text{U-GENOT}}(\theta) = \mathbb{E}_{t \sim \mathcal{U}([0,1]), Z \sim \rho, X \sim \tilde{\mu}, Y \sim \pi_{\varepsilon,\tau}^\star(\cdot|X)}[\|v_{t,\theta}(tZ + (1-t)Y|X) - (Y - Z)\|_2^2] \quad \text{(i)}$$

$$+ \mathbb{E}_{X \sim \mu}[(\eta(X) - \eta_\theta(X))^2] + \mathbb{E}_{Y \sim \nu}[(\xi(Y) - \xi_\theta(Y))^2]. \quad \text{(ii)}$$

As with GENOT, we simply need to estimate the unbalanced OT coupling $\hat{\pi}_{\varepsilon,\tau}$ from samples $X_1, \ldots \mathbf{x}_n$ from $\mu$ and $\mathbf{y}_1, \ldots \mathbf{y}_n$ from $\nu$ to estimate that loss. We build upon theoretical insights from the Kantorovich case, which we extend in practice to the Gromov-Wasserstein case.

**Proposition 3.3** (Estimation of the re-weightings.). *Let $\hat{\pi}_{\varepsilon,\tau}$ the solution of* (UEK) *computed on samples. Let $\mathbf{a} = \hat{\pi}_{\varepsilon,\tau} \mathbf{1}_n$ and $\mathbf{b} = \hat{\pi}_{\varepsilon,\tau}^\top \mathbf{1}_n$ be its marginal weights and let $\hat{\eta}_n(\mathbf{x}_i) := n\, a_i$ and $\hat{\xi}_n(\mathbf{y}_i) := n\, b_i$. Then, almost surely, $\hat{\eta}_n(\mathbf{x}_i) \to \eta(\mathbf{x}_i)$ and $\hat{\xi}_n(\mathbf{x}_i) \to \xi(\mathbf{y}_i)$.*

Using Prop. 3.2, $\hat{\pi}_{\varepsilon,\tau}$ is a balanced EOT coupling between its marginals, which are empirical approximations of $\tilde{\mu}$ and $\tilde{\nu}$. We hence estimate the term (i) of the loss as we do in the balanced case by sampling from the discrete conditional distribution. Furthermore, Prop.3.3 highlights that the estimation of $\hat{\pi}_{\varepsilon,\tau}$ also provides a consistent estimate of the re-weighting function evaluations at each $\mathbf{x}_i$ and $\mathbf{y}_i$. This enables the estimation of the term (ii). Therefore, as with GENOT, each U-GENOT iteration only requires a call to a discrete solver. We detail our training procedure in algorithm 2.

### 3.3 COMBINING KANTOROVICH AND GROMOV-WASSERSTEIN TO THE FUSED SETTING

We show in § 3.1 and § 3.2 how to use our method to map samples within the same space, or across incomparable spaces, by solving (EK) or (EGW) and their unbalanced extensions. On the other hand, there are cases where the source and the target domains are only *partially* incomparable, leading to a problem that combines both OT formulations (Vayer et al., 2018). Suppose that the source and target space can be decomposed as $\mathcal{X} = \Omega \times \tilde{\mathcal{X}}$ and $\mathcal{Y} = \Omega \times \tilde{\mathcal{Y}}$, respectively. Moreover, assume we are given an inter-domain cost $c : \Omega \times \Omega \to \mathbb{R}$ along with the intra-domain costs $c_{\tilde{\mathcal{X}}}, c_{\tilde{\mathcal{Y}}}$. The entropic fused-Gromov-Wassertein (FGW) problem can then be defined as

$$\min_{\pi \in \Pi(\mu,\nu)} \int_{((\Omega \times \tilde{\mathcal{X}}) \times (\Omega \times \tilde{\mathcal{Y}}))^2} L\left((\mathbf{u}, \mathbf{x}), (\mathbf{v}, \mathbf{y}), \mathbf{x}', \mathbf{y}'\right) d\pi\left((\mathbf{u}, \mathbf{x}), (\mathbf{v}, \mathbf{y})\right) d\pi(\mathbf{x}', \mathbf{y}') + \varepsilon \mathrm{KL}(\pi | \mu \otimes \nu),$$

EFGW

where $L\left((\mathbf{u}, \mathbf{x}), (\mathbf{v}, \mathbf{y}), \mathbf{x}', \mathbf{y}'\right) := (1 - \alpha)\, c(\mathbf{u}, \mathbf{v}) + \alpha\, |c_{\tilde{\mathcal{X}}}(\mathbf{x}, \mathbf{x}') - c_{\tilde{\mathcal{Y}}}(\mathbf{y}, \mathbf{y}')|^2$ and $\alpha \in [0, 1]$ determines the influence of the components of the space decompositions. When $\alpha = 1$, we recover the pure GW setting. The above fused problem admits an unbalanced extension, which can be derived exactly in the same way as (UEGW) using the quadratic $\mathrm{KL}^{\otimes}$ (Thual et al., 2023).

**(U-)GENOT Addresses the Fused Setting.** Whether in the balanced or unbalanced setting, we can use our method to learn a specific coupling as soon as it can be estimated from samples. We stress that the discrete solvers we use for problems (EGW) and (UEGW) are still applicable in the fused setting. As a result, we can compute discrete fused couplings and then solve (EFGW) and its unbalanced counterpart with (U-)GENOT. To illustrate this idea more precisely, take a solution $\pi_\alpha^\star$ of (EFGW). Learning $\pi_\alpha^\star$ with our method amounts to training vector fields that are conditioned on pairs of modality from the source domain $v_{t,\theta}(\cdot, |\mathbf{u}, \mathbf{x})$, to sample pairs of modality from the target domain via the induced flow: $\mathbf{z} \sim \rho$, $\phi_1(\mathbf{z}|\mathbf{u}, \mathbf{x}) = (\mathbf{v}, \mathbf{y}) \sim \pi_\alpha^\star(\cdot|\mathbf{u}, \mathbf{x})$. Given each term of the fused problem (EFGW), the sampled modalities $(\mathbf{v}, \mathbf{y})$ minimize transport cost quantified by $c$ along the first modality, while being "isometric" w.r.t. $c_{\tilde{\mathcal{X}}}$ and $c_{\tilde{\mathcal{Y}}}$ on the second modality.

## 4 RELATED WORK

**Neural EOT.** While GENOT is the first model to learn neural EOT couplings in the (Fused) Gromov-Wasserstein or the unbalanced setting, various methods have been proposed in the (balanced) Kantorovich setting. The first class of methods solves the (EK) dual problem. While some of them (Genevay et al., 2019a) do not allow direct sampling according to $\pi_\varepsilon^\star$, Daniels et al. (2021) model the conditional distribution $\pi_\varepsilon^\star(\cdot|\mathbf{x})$. However, this method is (i) costly as it employs Langevin sampling at inference and (ii) numerically unstable as it requires the exponentiation of large numbers. Mokrov et al. (2023) proposed another approach modeling $\pi_\varepsilon^\star(\cdot|\mathbf{x})$ leveraging energy-based models, but is computationally expensive since it relies on Langevin sampling in each training iteration. Other Kantorovich EOT solvers build upon the link between (EK) and the Schrödinger bridge (SB) problem. They model the EOT plan as a time-evolving stochastic process with fixed marginal constraints, endowed with learnable drift and diffusion terms (De Bortoli et al., 2021; Chen et al., 2021; Vargas et al., 2021; Gushchin et al., 2022). Although these methods have shown good performance on image data, they are very costly since they require simulation-based training. A recent line

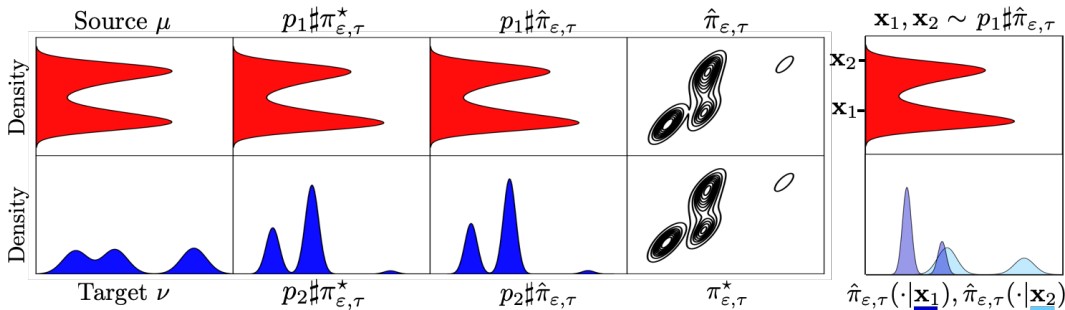

Figure 1: Prediction by UGENOT-K and ground truth of the unbalanced entropy-regularized transport plan between mixtures of Gaussians. The first column shows the source (top) and target (bottom) distribution. The second and third column show the marginal distributions of the true and the learnt transport plan, respectively. The fourth column compares the learnt (top) with the true (bottom) transport plan, while the fifth column plots conditional distributions. Here, $\varepsilon = 0.05$.

of work proposed to train such models in a completely simulation-free manner (Tong et al., 2023a;b; Shi et al., 2023; Liu et al., 2023) via score or flow matching. However, these methods can only be used for the squared Euclidean cost. Indeed, they rely on the fact that the marginals of the SB can be characterized as a mixture of Brownian bridges weighted by an EOT plan. However, this property is true only when we choose the Wiener process as a reference measure in the SB problem, which is limited to using $c(\mathbf{x}, \mathbf{y}) = \|\mathbf{x} - \mathbf{y}\|_2^2$ in (EK) (Léonard, 2013)[Eq. 1.2]. On the other hand, GENOT is the first neural EOT framework that can handle any cost function, even those defined implicitly, and whose evaluation requires a call to a non-differentiable sub-routine, like the geodesic distance on the data manifold. This point allows us to emphasize that our method fundamentally differs from theirs since we do not exploit the link between EOT and SB. Our approach is purely conditional and uses flow matching only as a powerful generative black box to learn, for each $\mathbf{x}$, a flow from $\rho$ to each $\pi_\varepsilon^\star(\cdot|\mathbf{x})$. Notably, since we set $\rho \in \mathcal{M}_1^+(\mathcal{Y})$ each flow occurs in the target domain $\mathcal{Y}$, which allows us to map distributions across spaces, while (Tong et al., 2023a;b; Shi et al., 2023; Liu et al., 2023) model (stochastic) flow directlty from $\mu$ to $\nu$, requiring from $\mu$ and $\nu$ to lie in the same space.

**Computation of Neural Couplings.** Another line of work considers computing neural couplings through the weak OT paradigm Korotin et al. (2022a;b); Asadulaev et al. (2022); Gazdieva et al. (2022), by solving a challenging min-max problem. However, (i) their method only enables mapping within the same space, (ii) in the balanced setting, and (iii) cannot handle EOT problems since they would require estimating the entropy of the neural coupling from samples at each iteration.

# 5 EXPERIMENTS

We demonstrate the applicability and versatility of the GENOT framework on toy data and single-cell data to map within the same space and across incomparable spaces. Metrics are discussed in appendix C and details on the single-cell datasets can be found in appendix D. Further experimental details or results for each experiment are reported in appendix E. Setups for competing methods are listed in appendix F. Details on the implementation of GENOT can be found in appendix G. We introduce the notation GENOT-K for the GENOT model solving problem (EK) while GENOT models solving the tasks (EGW) and (EFGW) are referred to as GENOT-GW and GENOT-FGW, respectively. The prefix U is used whenever consider an unbalanced problem, as described in § 3.2. Moreover, when reporting results based on the conditional mean of a GENOT model, we add the suffix *CM* to the model name. If not stated otherwise, we use the squared Euclidean distance as cost.

## 5.1 GENOT-K TO MAP WITHIN SPACES

**U-GENOT-K on simulated data** To visualize the capabilities of UGENOT-K to learn unbalanced entropy-regularized transport plans and rescaling functions, we compare its predictions with the OT plan obtained from a discrete EOT solver. Fig. 1 shows that the unbalanced entropy-regularized transport plan with $\varepsilon = 0.05$ and $\tau_1 = \tau_2 = 0.98$ between mixtures of Gaussians is accurately learnt by U-GENOT-K. The influence of the unbalancedness parameters $\tau_1, \tau_2$ is visualized in Fig. 7. The performance of GENOT-K is further assessed, in E, on its ability to learn the entropic OT coupling between Gaussian distributions.

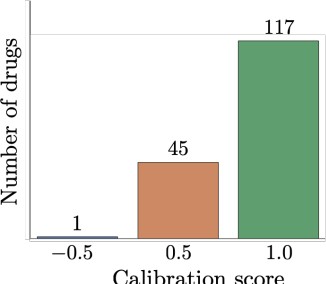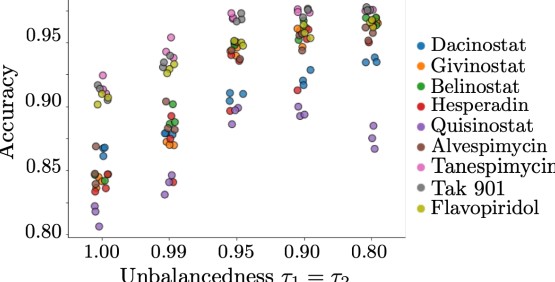

Figure 3: Left: Calibration score for the predictions of GENOT-K for modeling cellular responses to 163 cancer drugs (appendix C.1). Right: Accuracy of cellular response predictions of U-GENOT-K for different cancer drugs with varying unbalancedness parameter $\tau = \tau_1 = \tau_2$. For each $\tau$, U-GENOT-K was run three times with different seeds.

**U-GENOT-K for modeling single-cell trajectories** OT has been successfully applied to recover cellular trajectories in time-resolved single-cell data (Schiebinger et al., 2019). Due to the ever increasing size of these datasets (Haniffa et al., 2021), neural OT solvers are of particular interest and deterministic Monge map estimators have been successfully applied to millions of cells (He et al., 2023). We apply GENOT-K to a dataset capturing gene expression of the developing mouse pancreas at embryonic days 14.5 and 15.5 (Bastidas-Ponce et al., 2019). We assess the fitting property of the learnt plan by computing the Sinkhorn divergence (Feydy et al., 2019a) between the predicted target distribution $p_2 \sharp \hat{\pi}_\varepsilon$ and the target distribution see (E.2). Fig. 12 shows that GENOT-K outperforms competing methods.

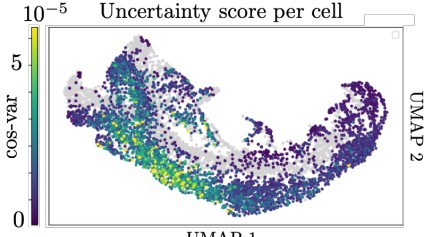

Figure 2: UMAP (McInnes et al., 2018) of the mouse pancreas development dataset colored by uncertainty per cell in the source distribution. Cells in the target distribution are colored in gray.

A key feature of all GENOT models is the ability to sample from the conditional distribution. Indeed, it is indispensable to stochastically model cellular trajectories, as cells are known to evolve non-deterministically (Elowitz et al., 2002). Following Gayoso et al. (2022), we compute $\text{cos-var}(\hat{\pi}_\varepsilon(\cdot|\mathbf{x})) = \text{Var}_{Y \sim \hat{\pi}_\varepsilon(\cdot|\mathbf{x})}[\text{cos-sim}(Y, \mathbb{E}_{Y \sim \hat{\pi}_\varepsilon(\cdot|\mathbf{x})}[Y])]$, where $\text{cos-sim}(\cdot, \cdot)$ denotes the cosine similarity, to assess the uncertainty of cell trajectories in the developing mouse pancreas (appendix C.1). We expect high uncertainty in cell types with fate decisions and low variance in mature cell types or cell types with a homogeneous descending population. Indeed, Fig. 2 and Fig. 13 show that GENOT-K helps to uncover lineage branching events.

The pancreas dataset considered so far subsets the original dataset to one cell lineage (endocrine) to prevent obtaining biologically implausible couplings. Indeed, table 1 shows that in the balanced case, the cell lineage transition score (see C.2) shows that only $66\%$ of the cells are mapped to the correct lineage. By loosening the conservation of mass constraint, U-GENOT-K helps to counteract the distributional shift introduced by different proliferation rates of cells and experimental biases.

**Prediction of cellular responses to drug perturbations with U-GENOT-K** In-silico perturbation prediction is a promising approach to accelerate drug discovery and improve gene therapies (Ji et al., 2021; Hetzel et al., 2022). Neural OT has been successfully applied to model cellular responses to such perturbations, using deterministic Monge maps (Bunne et al., 2021; Uscidda & Cuturi, 2023). GENOT has the comparative advantage that it can *sample* from the conditional distribution, which allows for uncertainty quantification. We consider single-cell RNAseq data measuring the response of cells to 163 cancer drugs (Srivatsan et al., 2020). Each drug has been applied to a population of cells that can be partitioned into 3 different cell types. While there is no ground truth in the matching between unperturbed and perturbed cells due to the destructive nature of sequencing technologies, we know which unperturbed subset of cells is supposed to be mapped to which perturbed subset of cells. We use this to define an accuracy metric (Appendix C.2). For the uncertainty metric, we choose again cos-var. Fig. 3 shows that for 117 out of 163 drugs the model is perfectly calibrated (Appendix C.1), while it yields a negative correlation between error and uncertainty only for one drug. To improve the accuracy of GENOT-K, we leverage its unbalanced formulation. Fig. 3 shows that allowing for mass variation improves the performance for nine different cancer drugs which are known to have a strong effect. Fig. 17 and 18 confirm the results visually.

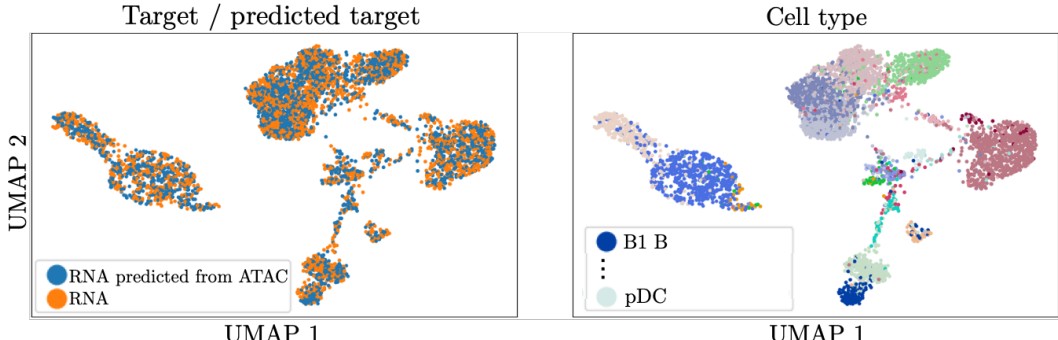

Figure 5: UMAP embedding of transported cells and cells in the target distribution (left), and jointly colored by cell type (right).

## 5.2 GENOT-GW AND GENOT-FGW TO MAP ACROSS SPACES

**GENOT-GW on simulated data.** We transport a Swiss role in $\mathbb{R}^3$ to a spiral in $\mathbb{R}^2$. Fig. 4 shows that GENOT-GW successfully mimics an isometric alignment. Here, we set $\varepsilon = 0.01$ and investigate its influence in more detail in Fig. 19.

**GENOT-GW for translating modalities of single cells** The number of modalities which can be simultaneously measured in a single cell is limited due to technical limitations. At the same time, new technologies allow to capture a more diverse set of modalities Baysoy et al. (2023). Yet, it is important to match measurements of different modalities to obtain a more holistic view of the profile of a cell. The discrete GW formulation has been used to match

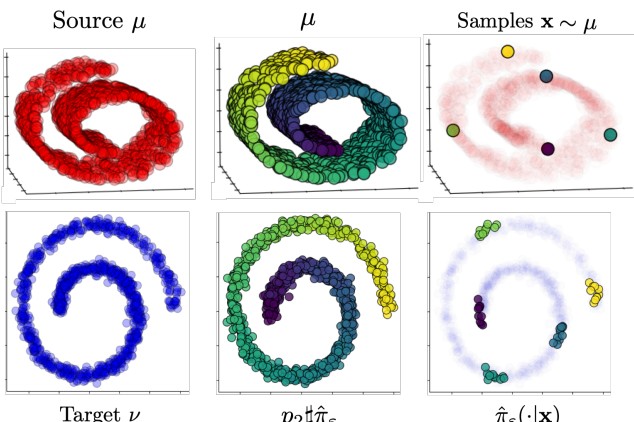

Figure 4: Mapping a Swiss roll in $\mathbb{R}^3$ (top left) to a spiral in $\mathbb{R}^2$ (bottom left). Center: Color code tracks where samples from the source (top) are mapped to (bottom). Right column: samples (top) and their conditional distributions.

measurements of cells in different modalities (Demetci et al., 2022). We use GENOT-GW to translate ATAC measurements to gene expression space on a bone marrow dataset (Luecken et al., 2021). As both modalities were measured in the same cell, the true match of each cell is known. We compare GENOT-GW with the discrete GW formulation (see F.2) and assess the performance with the FOSCTTM ("Fractions of Samples Closer to the True Match") score (see C.2). We leverage the flexibility of GENOT and use an approximated geodesic distance (Crane et al., 2013) rather than the Euclidean distances, which is not meaningful within embeddings of single-cell measurements (Moon et al., 2018). Fig. 6 shows 3 results related to the FOSCTTM score. First, using a graph-based cost is crucial in higher dimensions. Second, out-of-sample prediction for discrete GW based on regression (GW-LR) is competitive in low-dimensions, but not for higher. Third, taking the conditional mean as prediction improves the result with respect to the FOSCTTM score. Regarding the distributional fitting property, GENOT models are clearly superior. Crucially, Fig. 6 shows that the fitting property of GENOT models is not affected by the cost.

**GENOT-FGW improves modality translation of single cells** As the predictions yielded by GW-based models are not satisfactory, we introduce a novel method for translating between ATAC and RNA measurements by extending the model proposed by Demetci et al. (2022) to the fused setting. Therefore, we infer approximate gene expression from the ATAC measurements using gene activity (Stuart et al., 2021). We construct a joint space of the two modalities using a conditional VAE (Lopez et al., 2018a). Fig. 20 shows that the additional fused term helps to obtain a significantly better alignment compared to GENOT-GW, with the best GENOT-FGW CM model (weight parameter $\alpha = 0.7$) attaining a FOSCTTM score of below 0.05. It is important to note that incorporating the GW terms is necessary for attaining good results as discussed in appendix E.3. Fig. 5 visualizes

the push-forward of the learnt coupling. The intertwinement of samples of the target and the predicted target in the left panel visualizes the distribution fitting property, while the separation into cell types on the right confirms the optimality of the learnt coupling. See figures 23 and 24 for further visualizations. When aligning multiple modalities of single cells, we cannot assume to have the same proportion of cell types in both datasets, for example due to experimental biases caused by sequencing technologies. We simulate this setting by removing cells belonging to either of the cell types *Proerythroblasts*, *Erythroblasts* or *Normoblasts* in the source distribution. Table 3 shows that U-GENOT-FGW preserves high accuracy while learning meaningul rescaling functions.

**Conclusion.** We introduce GENOT, a versatile neural OT framework to learn cost-efficient stochastic maps within the same space and/or across incomparable spaces. GENOT is flexible to the extent that the mass

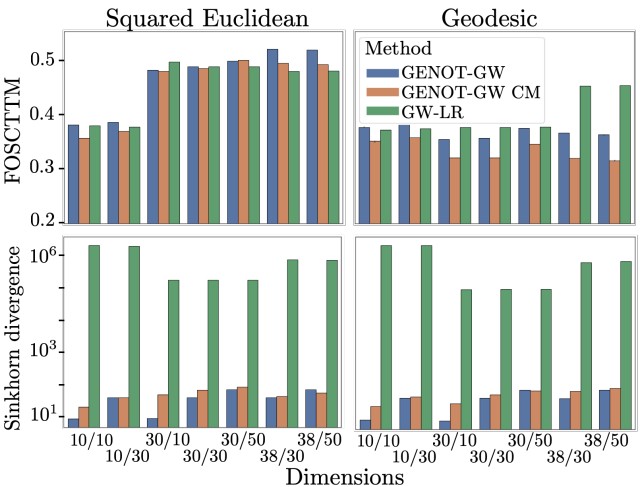

Figure 6: Benchmark (mean and std across three runs) of GENOT-GW models against discrete GW (GW-LR, appendix F) on translating cells between ATAC space of dimension $d_1$ and RNA space of dimension $d_2$ for experiment $d_1/d_2$. Performance is measured with the FOSCTTM score (appendix C.2) and the Sinkhorn divergence between target and predicted target distribution. While on the left, we learn the EOT coupling for the squared Euclidean cost, we use the geodesic cost on the right (Crane et al., 2013).

conservation constraint can be loosened, and provides tools to sample targets from an input. GENOT can be used within a wide array of tasks in single-cell biology.

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

APPENDIX

## A  ALGORITHMS

---

**Algorithm 1** GENOT

---

**Require:** Source distribution $\mu$; target distribution $\nu$; batch size $n$; number of (per $\mathbf{x}$) conditional sample $k$; number of iterations $T_{\text{iter}}$, entropic regularization strength $\varepsilon$, discrete solver $\text{Solver}_\varepsilon$ to estimate $\pi_\varepsilon^\star$ from samples, parameterized time-dependent velocity field $v_{t,\theta}$.

1: **for** $t = 1, \ldots, T_{\text{iter}}$ **do**
2:    Sample batches $\mathbf{x}_1, \ldots \mathbf{x}_n \sim_{\text{i.i.d}} \mu$ and $\mathbf{y}_1, \ldots \mathbf{y}_n \sim_{\text{i.i.d}} \nu$.
3:    Compute $\hat{\pi}_\varepsilon = \text{Solver}_\varepsilon \left( \{\mathbf{x}_i\}_{i=1}^n, \{\mathbf{y}_i\}_{i=1}^n \right) \in \mathbb{R}_+^{n \times n}$.
4:    **for** $i = 1, \ldots, n$ **do**
5:       Sample from the discrete conditional distribution $\hat{\mathbf{y}}_{i,1}, \ldots \hat{\mathbf{y}}_{i,k} \sim \hat{\pi}_\varepsilon(\cdot | \mathbf{x}_i)$.
6:       Sample noise vectors $\mathbf{z}_{i,1}, \ldots, \mathbf{z}_{i,k} \sim \rho$.
7:       Sample time-steps $\mathbf{t}_{i,1}, \ldots, \mathbf{t}_{i,k} \sim \mathcal{U}([0,1])$.
8:    **end for**
9:    Estimate GENOT loss:

$$\hat{\mathcal{L}}_{\text{GENOT}}(\theta) \leftarrow \frac{1}{n} \sum_{i=1}^n \frac{1}{k} \sum_{j=1}^k \| v_{t,\theta}(\mathbf{t}_{i,j} \, \mathbf{z}_{i,j} + (1 - \mathbf{t}_{i,j}) \, \hat{\mathbf{y}}_{i,j} \, | \, \mathbf{x}_i) - (\hat{\mathbf{y}}_{i,j} - \mathbf{z}_{i,j}) \|_2^2 .$$

10:    Update $\theta$ to minimize $\hat{\mathcal{L}}_{\text{GENOT}}$.
11: **end for**

---

---

**Algorithm 2** U-GENOT

---

**Require:** Source distribution $\mu$; target distribution $\nu$; batch size $n$; number of (per $\mathbf{x}$) conditional sample $k$; number of iterations $T_{\text{iter}}$, entropic regularization strength $\varepsilon$, left and right unbalancedness parameter $\tau = (\tau_1, \tau_2)$, discrete solver $\text{Solver}_{\varepsilon,\tau}$ to estimate $\pi_{\varepsilon,\tau}^\star$ from samples, parameterized time-dependent velocity field $v_{t,\theta}$, parameterized re-weighting functions $\eta_\theta, \xi_\theta$.

1: **for** $t = 1, \ldots, T_{\text{iter}}$ **do**
2:    Sample batches $\mathbf{x}_1, \ldots \mathbf{x}_n \sim_{\text{i.i.d}} \mu$ and $\mathbf{y}_1, \ldots \mathbf{y}_n \sim_{\text{i.i.d}} \nu$.
3:    Compute $\hat{\pi}_{\varepsilon,\tau} \leftarrow \text{Solver}_{\varepsilon,\tau} \left( \{\mathbf{x}_i\}_{i=1}^n, \{\mathbf{y}_i\}_{i=1}^n \right)$ $O(n^2)$ or $O(n^3)$, depending on solver.
4:    Set the marginal weights: $\mathbf{a} \leftarrow \hat{\pi}_{\varepsilon,\tau} \mathbf{1}_n$  and  $\mathbf{b} \leftarrow \hat{\pi}_{\varepsilon,\tau}^\top \mathbf{1}_n$.
5:    Set $\tilde{\mu}_n \leftarrow \sum_{i=1}^n a_i \delta_{\mathbf{x}_i}$ the left marginal of $\hat{\pi}_{\varepsilon,\tau}$.
6:    Sample $\tilde{\mathbf{x}}_1, \ldots, \tilde{\mathbf{x}}_n \sim \tilde{\mu}_n$.
7:    **for** $i = 1, \ldots, n$ **do**
8:       Sample from the discrete conditional distribution $\hat{\mathbf{y}}_{i,1}, \ldots \hat{\mathbf{y}}_{i,k} \sim \hat{\pi}_{\varepsilon,\tau}(\cdot | \tilde{\mathbf{x}}_i)$.
9:       Sample noise vectors $\mathbf{z}_{i,1}, \ldots, \mathbf{z}_{i,k} \sim \rho$.
10:       Sample time-steps $\mathbf{t}_{i,1}, \ldots, \mathbf{t}_{i,k} \sim \mathcal{U}([0,1])$.
11:    **end for**
12:    Estimate U-GENOT loss:

$$\hat{\mathcal{L}}_{\text{U-GENOT}}(\theta) \leftarrow \frac{1}{n} \sum_{i=1}^n \frac{1}{k} \sum_{j=1}^k \| v_{t,\theta}(\mathbf{t}_{i,j} \, \mathbf{z}_{i,j} + (1 - \mathbf{t}_{i,j}) \, \hat{\mathbf{y}}_{i,j} \, | \, \tilde{\mathbf{x}}_i) - (\hat{\mathbf{y}}_{i,j} - \mathbf{z}_{i,j}) \|_2^2 .$$

$$+ \frac{1}{n} \sum_{i=1}^n (\eta_\theta(\mathbf{x}_i) - n\mathbf{a}_i)^2 + \frac{1}{n} \sum_{i=1}^n (\xi_\theta(\mathbf{y}_i) - n\mathbf{b}_i)^2$$

13:    Update $\theta$ to minimize $\hat{\mathcal{L}}_{\text{U-GENOT}}$.
14: **end for**

---

# B PROOFS

## B.1 PROOFS OF § 3

**Proposition 3.1** (Well-posedness of GENOT Loss). *Suppose that $\mathcal{L}_{\text{GENOT}}(\theta) = 0$. Then the flows $\{\phi_1(\cdot|\mathbf{x})\}_{\in\mathcal{X}}$, induced by the velocity fields $\{v_{t,\theta}(\cdot|\mathbf{x})\}_{\in\mathcal{X}}$, are a collection of optimal conditional generators. Namely, for $\mathbf{x}$ in the support of $\mu$, $Z \sim \rho$ and $Y = \phi_1(Z|\mathbf{x})$ denoting the solution of the ODE (5), then $Y \sim \pi_\varepsilon^\star(\cdot|\mathbf{x})$, therefore this ideal conditional vector field $v_{t,\theta}$ recovers $\pi_\varepsilon^\star$.*

*Proof.* This result follows directly from the construction of the loss. Suppose that $\mathcal{L}_{\text{GENOT}}(\theta) = 0$. By Fubini's Theoreom, this implies that:

$$\mathbb{E}_{X\sim\mu}\mathbb{E}_{t\sim\mathcal{U}([0,1]), Z\sim\rho, Y\sim\pi_\varepsilon^\star(\cdot|\mathbf{x})}[\|v_{t,\theta}\left(tZ + (1-t)Y|X\right) - (Y - Z)\|_2^2] = 0\,. \tag{8}$$

Since it is positive, the function

$$\ell : \mathbf{x} \mapsto \mathbb{E}_{t\sim\mathcal{U}([0,1]), Z\sim\rho, Y\sim\pi_\varepsilon^\star(\cdot|)}[\|v_{t,\theta}\left(tZ + (1-t)Y|\mathbf{x}\right) - (Y - Z)\|_2^2]$$

is therefore zero $\mu$-a.e. which means that for any source sample $\mathbf{x} \sim \mu$, one has $\ell(\mathbf{x}) = 0$. Moreover, $\ell(\mathbf{x})$ is the CFM loss with source $\rho$ and target $\pi_\varepsilon^\star(\cdot|\mathbf{x})$, applied to the vector field $v_{t,\theta}(\cdot|\mathbf{x})$. Therefore, $\ell(\mathbf{x}) = 0$ implies that $v_{t,\theta}(\cdot|\mathbf{x})$ generates a probability path between $\rho$ and $\pi^\star(\cdot|\mathbf{x})$ thanks to (Lipman et al., 2023)[Theorem 1], which means that if $\phi_t(\cdot|\mathbf{x})$ is the flow induced by $v_{t,\theta}(\cdot|\mathbf{x})$, $\phi_1(\cdot|\mathbf{x})\sharp\rho = \pi_\varepsilon^\star(\cdot|\mathbf{x})$. Therefore, $\{\phi_1(\cdot|\mathbf{x})\}_{\mathbf{x}\in\mathcal{X}}$ is a collection of optimal conditional generator. $\square$

**Proposition 3.2** (Re-Balancing the unbalanced problems.). *Let $\pi_{\varepsilon,\tau}^\star$ be an unbalanced EOT coupling, solution of* (UEK) *or* (UEGW) *between $\mu \in \mathcal{M}^+(\mathcal{X})$ and $\nu \in \mathcal{M}^+(\mathcal{Y})$. We note $\tilde{\mu} = p_1\sharp\pi_{\varepsilon,\tau}^\star$ and $\tilde{\nu} = p_2\sharp\pi_{\varepsilon,\tau}^\star$ its marginals. Then, in both cases, $\tilde{\mu}$ (resp. $\tilde{\nu}$) has a density w.r.t $\mu$ (resp. $\nu$) i.e. it exists $\eta, \xi : \mathbb{R}^d \to \mathbb{R}^+$ s.t. $\tilde{\mu} = \eta \cdot \mu$ and $\tilde{\nu} = \xi \cdot \nu$. Moreover, $\tilde{\mu}$ and $\tilde{\nu}$ have the same mass and*

1. *(Kantorovich) $\pi_{\varepsilon,\tau}^\star$ solves the balanced problem* (EK) *between $\tilde{\mu}$ and $\tilde{\nu}$ with the same $\varepsilon$.*

2. *(Gromov-Wasserstein) Provided that $c_\mathcal{X}$ and $c_\mathcal{Y}$ are conditionally positive (or conditionally negative) kernels (see Def. B.1), $\pi_{\varepsilon,\tau}^\star$ solves the balanced problem* (EGW) *between $\tilde{\mu}$ and $\tilde{\nu}$ with $\varepsilon' = m(\pi_{\varepsilon,\tau}^\star)\,\varepsilon$, where $m(\pi_{\varepsilon,\tau}^\star) = \pi_{\varepsilon,\tau}^\star(\mathcal{X} \times \mathcal{Y})$ is the total mass of $\pi_{\varepsilon,\tau}^\star$.*

**Definition B.1.** A kernel $k : \mathbb{R}^d \times \mathbb{R}^d \to \mathbb{R}$ is conditionally positive (resp. negative) if it is symmetric and for any $\mathbf{x}_1, ..., \mathbf{x}_n \in \mathbb{R}^d$ and $\mathbf{a} \in \mathbb{R}^n$ s.t. $\mathbf{a}^\top \mathbf{1}_n = 0$, one has

$$\sum_{i,j=1}^n a_i a_j\, k(\mathbf{x}_i, \mathbf{x}_j) \geq 0 \quad (\text{resp. } \leq 0)$$

*Proof of 3.2.* **Step 1: Re-weightings.** We first show that for $\pi_{\varepsilon,\tau}^\star$ solution of EK problem, it exists $\eta, \xi : \mathbb{R}^d \to \mathbb{R}^+$ s.t. $\tilde{\mu} = p_1\sharp\pi_{\varepsilon,\tau}^\star = \eta \cdot \mu$ and $\tilde{\nu} = p_1\sharp\pi_{\varepsilon,\tau}^\star = \xi \cdot \nu$. We then remind the EK problem between $\mu$ and $\nu$:

$$\min_{\pi\in\mathcal{M}^+(\mathcal{X}\times\mathcal{Y})} \int_{\mathcal{X}\times\mathcal{Y}} c(\mathbf{x}, \mathbf{y})\, \mathrm{d}\pi(\mathbf{x}, \mathbf{y}) + \varepsilon\mathrm{KL}(\pi|\mu \otimes \nu) + \tau_1\mathrm{KL}(\pi_1|\mu) + \tau_2\mathrm{KL}(\pi_2|\nu)\,. \tag{9}$$

The relative entropy term $\mathrm{KL}(\pi|\mu\otimes\nu)$ in (9) is finite if and only if $\pi$ admits a density with respect to $\mu \otimes \nu$. Therefore, one can reformulate 9 by restricting to the plan $\pi = h \cdot \mu \otimes \nu$ with (non-negative) relative density $h \in L_1^+(\mathcal{X} \times \mathcal{Y})$ w.r.t. $\mu \otimes \nu$. Moreover, in that case, the left marginal $\pi_1$ of $\pi$ has a density $h_1$ w.r.t. $\mu$ and the right marginal $\pi_2$ has a density $h_2$ w.r.t. $\nu$, and these densities are given by: $h_1(\cdot) := \int_\mathcal{Y} h(\cdot, \mathbf{y})\, \mathrm{d}\nu(\mathbf{y})$ and $h_2(\cdot) := \int_\mathcal{X} h(\mathbf{x}, \cdot)\, \mathrm{d}\mu(\mathbf{x})$. Indeed, for any Borel set $A \subset \mathcal{X}$, one has:

$$\begin{aligned}
\pi_1(A) &= \pi(A \times \mathcal{Y}) \\
&= \int_{\mathcal{X}\times\mathcal{Y}} 1_A(\mathbf{x}) f(\mathbf{x}, \mathbf{y})\, \mathrm{d}\mu(\mathbf{x})\, \mathrm{d}\nu(\mathbf{y}) \\
&= \int_\mathcal{X} 1_A(\mathbf{x}) \left( \int_\mathcal{Y} f(\mathbf{x}, \mathbf{y})\, \mathrm{d}\nu(\mathbf{y}) \right) \mathrm{d}\mu(\mathbf{x}) \\
&= \int_\mathcal{X} 1_A(\mathbf{x}) h_1(\mathbf{x})\, \mathrm{d}\mu(\mathbf{x})
\end{aligned} \tag{10}$$

where the penultimate line follows from Fubini's theorem, so $\pi_1 = h_1 \cdot \mu$. We show similarly that $\pi_2 = h_2 \cdot \nu$. Therefore, one can reformulate (9) as:

$$\min_{h \in L_1^+(\mathcal{X} \times \mathcal{Y})} \int_{\mathcal{X} \times \mathcal{Y}} c(\mathbf{x}, \mathbf{y}) h(\mathbf{x}, \mathbf{y}) \, \mathrm{d}\mu(\mathbf{x}) \, \mathrm{d}\nu(\mathbf{y}) + \varepsilon \mathrm{KL}(h|\mu \otimes \nu) + \tau_1 \mathrm{KL}(h_1|\mu) + \tau_2 \mathrm{KL}(h_2|\nu) \quad (11)$$

where we extend the KL divergence for densities: $\mathrm{KL}(r|\gamma) = \int (r \log(r) + r - 1) \, \mathrm{d}\gamma$. As a result, it exists $h^\star \in L_1^+(\mathcal{X} \times \mathcal{Y})$ s.t. $\pi_{\varepsilon,\tau}^\star = h^\star \cdot \mu \otimes \nu$ and then $\tilde{\mu} = \eta \cdot \mu$ and $\tilde{\nu} = \xi \cdot \nu$ with $\eta = h_1^\star$ and $\xi = h_2^\star$. Moreover, by definition, $\pi_{\varepsilon,\tau}^\star$ is a coupling between its marginal $\tilde{\mu}$ and $\tilde{\nu}$, so they have the same total mass. Indeed, using Fubini's Theorem twice, we get:

$$\int_{\mathcal{X} \times \mathcal{Y}} \mathrm{d}\pi_{\varepsilon,\tau}^\star(\mathbf{x}, \mathbf{y}) = \int_{\mathcal{X}} \mathrm{d}\tilde{\mu}(\mathbf{x}) = \int_{\mathcal{Y}} \mathrm{d}\tilde{\nu}(\mathbf{y}) \quad (12)$$

We now handle the Gromov-Wasserstein case, when $\pi_{\varepsilon,\tau}^\star$ solve problem UEGW. We then remind problem UEGW between $\mu$ and $\nu$:

$$\min_{\pi \in \mathcal{M}^+(\mathcal{X} \times \mathcal{Y})} \int_{(\mathcal{X} \times \mathcal{Y})^2} |c_{\mathcal{X}}(\mathbf{x}, \mathbf{x}') - c_{\mathcal{Y}}(\mathbf{y}, \mathbf{y}')|^2 \, \mathrm{d}\pi(\mathbf{x}, \mathbf{y}) \, \mathrm{d}\pi(\mathbf{x}', \mathbf{y}')$$
$$+ \varepsilon \mathrm{KL}^\otimes(\pi|\mu \otimes \nu) + \lambda_1 \mathrm{KL}^\otimes(\pi_1|\mu) + \lambda_2 \mathrm{KL}^\otimes(\pi_2|\nu), \quad (13)$$

The relative entropy penalty is now quadratic: $\mathrm{KL}^\otimes(\pi|\mu \otimes \mu) = \mathrm{KL}(\pi \otimes \pi|(\mu \otimes \nu)^2)$, where $(\mu \otimes \nu)^2 = (\mu \otimes \nu) \otimes (\mu \otimes \nu)$ to lighten notations. Therefore, the above objective function is finite i.f.f. $\pi \otimes \pi$ has a density w.r.t. $(\mu \otimes \nu)^2$. Using arguments similar to the ones used above, it means that the marginals of $\pi \otimes \pi$ have a density w.r.t. the marginals of $(\mu \otimes \nu)^2$, which implies that $\pi$ has a density w.r.t. $(\mu \otimes \nu)$. Now, we get the result by following exactly the same strategy as above.

**Remark B.2.** Note that in both cases, since $\mathrm{d}\tilde{\mu}(\mathbf{x}) = \eta(\mathbf{x}) \, \mathrm{d}\mu(\mathbf{x})$ and $\mathrm{d}\tilde{\nu}(\mathbf{y}) = \xi(\mathbf{y}) \, \mathrm{d}\nu(\mathbf{y})$, the equality of mass of $\tilde{\mu}$ and $\tilde{\nu}$ yields $\mathbb{E}_{X \sim \mu}[\eta(X)] = \mathbb{E}_{Y \sim \nu}[\xi(Y)]$.

**Step 2: Optimality in the balanced problem for the Kantorovich case.** We now prove **point 1**, stating that if $\pi_{\varepsilon,\tau}^\star$ solves problem UEK between $\mu$ and $\nu$, then it solves problem EK between $\tilde{\mu}$ and $\tilde{\nu}$ for the same entropic regularization strength $\varepsilon$.

We rely on duality and the specific structure of the optimal density $h^\star$ s.t. $\pi_{\varepsilon,\tau}^\star = h^\star \cdot \mu \otimes \nu$. Thanks to Séjourné et al. (2023a, Prop. 2), one has the existence of the so-called entropic potentials $f^\star \in \mathcal{C}(\mathcal{X}), g^\star \in C(\mathcal{Y})$ s.t.

$$h^\star(\mathbf{x}, \mathbf{y}) = \frac{\mathrm{d}\pi_{\varepsilon,\tau}^\star}{\mathrm{d}(\mu \otimes \nu)}(\mathbf{x}, \mathbf{y}) = \exp\left(\frac{f^\star(\mathbf{x}) + g^\star(\mathbf{y}) - c(\mathbf{x}, \mathbf{y})}{\varepsilon}\right) \quad (14)$$

Moreover, by Nutz, Theorem 4.2, such a decomposition is equivalent to the optimality in problem EK. Therefore, $\pi_{\varepsilon,\tau}^\star$ is solves problem EK between its marginals $\tilde{\mu}$ and $\tilde{\nu}$, i.e.

$$\pi_{\varepsilon,\tau}^\star = \arg\min_{\pi \in \Pi(\tilde{\mu}, \tilde{\nu})} \int_{\mathcal{X} \times \mathcal{Y}} c(\mathbf{x}, \mathbf{y}) \, \mathrm{d}\pi(\mathbf{x}, \mathbf{y}) + \varepsilon \mathrm{KL}(\pi|\mu \otimes \nu). \quad (15)$$

**Step 3: Optimality in the balanced problem for the Gromov-Wasserstein case.** We now prove **point 2**, stating that, provided that the costs $c_{\mathcal{X}}$ and $c_{\mathcal{Y}}$ are conditionally positive (or conditionally negative), if $\pi_{\varepsilon,\tau}^\star$ is a solves problem UEGW between $\mu$ and $\nu$, then it solves problem EGW between $\tilde{\mu}$ and $\tilde{\nu}$ for the entropic regularization strength $\varepsilon' = m(\pi_{\varepsilon,\tau}^\star)\,\varepsilon$.

Define the functional:

$$F : (\gamma, \pi) \in \mathcal{M}^+(\mathcal{X} \times \mathcal{Y})^2 \mapsto \int_{(\mathcal{X} \times \mathcal{Y})^2} |c_{\mathcal{X}}(\mathbf{x}, \mathbf{x}') - c_{\mathcal{Y}}(\mathbf{y}, \mathbf{y}')|^2 \, \mathrm{d}\pi(\mathbf{x}, \mathbf{y}) \, \mathrm{d}\gamma(\mathbf{x}', \mathbf{y}')$$
$$+ \varepsilon \mathrm{KL}(\pi \otimes \gamma|(\mu \otimes \nu)^2) + \lambda_1 \mathrm{KL}(\pi_1 \otimes \gamma_1|\mu \times \mu) + \lambda_2 \mathrm{KL}^\otimes(\pi_2 \otimes \gamma_2|\nu \times \nu), \quad (16)$$

s.t. $\pi_{\varepsilon,\tau}^\star \in \arg\min_{\pi \in \mathcal{M}(\mathcal{X} \times \mathcal{Y})} F(\pi, \pi)$. By first-order condition, one has:

$$\pi_{\varepsilon,\tau}^\star \in \arg\min_{\gamma \in \mathcal{M}(\mathcal{X} \times \mathcal{Y})} F(\gamma, \pi_{\varepsilon,\tau}^\star) \quad (17)$$

We then define the linearized cost

$$c_{\varepsilon,\tau}^{\star} : (\mathbf{x}, \mathbf{y}) \in \mathcal{X} \times \mathcal{Y} \mapsto \int_{\mathcal{X} \times \mathcal{Y}} |c_{\mathcal{X}}(\mathbf{x}, \mathbf{x}') - c_{\mathcal{Y}}(\mathbf{y}, \mathbf{y}')|^2 \, \mathrm{d}\pi_{\varepsilon,\tau}^{\star}(\mathbf{x}', \mathbf{y}'), \tag{18}$$

s.t. from (Séjourné et al., 2023a)[Proposition 9], (18) implies that $\pi_{\varepsilon,\tau}^{\star}$ solves:

$$\pi_{\varepsilon,\tau}^{\star} \in \operatorname*{arg\,min}_{\pi \in \mathcal{M}(\times \mathcal{Y})} \int_{\mathcal{X} \times \mathcal{Y}} c_{\varepsilon,\tau}^{\star}(\mathbf{x}, \mathbf{y}) \, \mathrm{d}\pi(\mathbf{x}, \mathbf{y}) + \varepsilon m(\pi_{\varepsilon,\tau}^{\star}) \mathrm{KL}(\pi | \mu \otimes \nu)$$
$$+ \lambda_1 m(\pi_{\varepsilon,\tau}^{\star}) \mathrm{KL}(\pi_1 | \mu) + \lambda_2 m(\pi_{\varepsilon,\tau}^{\star}) \mathrm{KL}(\pi_2 | \nu). \tag{19}$$

so $\pi_{\varepsilon,\tau}^{\star}$ solves problem UEK between $\mu$ and $\nu$ for a new cost $c_{\varepsilon,\tau}^{\star}$, and the regularization strength $\varepsilon' = \varepsilon m(\pi_{\varepsilon,\tau}^{\star})$. We seek to apply point 1, to get that $\pi_{\varepsilon,\tau}^{\star}$ solves problem EK between $\tilde{\mu} = p_1 \sharp \pi_{\varepsilon,\tau}^{\star}$ and $\tilde{\nu} = p_2 \sharp \pi_{\varepsilon,\tau}^{\star}$ for the same entropic regularization strength $\varepsilon'$. To that end, we first verify that $c_{\varepsilon,\tau}^{\star}$ is continuous. For every $\mathbf{x}, \mathbf{x}' \in \mathcal{X}$ and $\mathbf{y}, \mathbf{y}' \in \mathcal{Y}$, one has

$$|c_{\mathcal{X}}(\mathbf{x}, \mathbf{x}') - c_{\mathcal{Y}}(\mathbf{y}, \mathbf{y}')|^2 \leq |c_{\mathcal{X}}(\mathbf{x}, \mathbf{x}')|^2 + c_{\mathcal{Y}}(\mathbf{y}, \mathbf{y}')|$$
$$\leq \sup_{\mathbf{x}, \mathbf{x}' \in \mathcal{X}} |c_{\mathcal{X}}(\mathbf{x}, \mathbf{x}')|^2 + \sup_{\mathbf{y}, \mathbf{y}' \in \mathcal{Y}} |c_{\mathcal{Y}}(\mathbf{y}, \mathbf{y}')|^2 \tag{20}$$
$$< +\infty$$

where the last line follows from the fact that $c_{\mathcal{X}}$ and $c_{\mathcal{Y}}$ are continuous on $\mathcal{X} \times \mathcal{X}$ and $\mathcal{Y} \times \mathcal{Y}$, which are compact sets as product of compact sets. Then, since $\pi_{\varepsilon,\tau}^{\star}$ has finite mass, Lebesgue's dominated convergence yields the continuity of $c_{\varepsilon,\tau}^{\star}$. We then to apply point 1 and get:

$$\pi_{\varepsilon,\tau}^{\star} \in \operatorname*{arg\,min}_{\pi \in \Pi(\tilde{\mu}, \tilde{\nu})} \int_{\mathcal{X} \times \mathcal{Y}} c_{\varepsilon,\tau}^{\star}(\mathbf{x}, \mathbf{y}) \, \mathrm{d}\pi(\mathbf{x}, \mathbf{y}) + \varepsilon' \mathrm{KL}(\pi | \mu \otimes \nu) \tag{21}$$

Since the costs are conditionally positive (or conditionally negative) kernels, (21) finally yields the desired result by applying (Séjourné et al., 2023b)[Theorem 3]:

$$\pi_{\varepsilon,\tau}^{\star} \in \operatorname*{arg\,min}_{\pi \in \Pi(\tilde{\mu}, \tilde{\nu})} \int_{\mathcal{X} \times \mathcal{Y}} |c_{\mathcal{X}}(\mathbf{x}, \mathbf{x}') - c_{\mathcal{Y}}(\mathbf{y}, \mathbf{y}')| \, \mathrm{d}\pi(\mathbf{x}', \mathbf{y}') \, \mathrm{d}\pi(\mathbf{x}, \mathbf{y}) + \varepsilon' \mathrm{KL}(\pi | \mu \otimes \nu) \tag{22}$$

$\square$

**Proposition 3.3** (Estimation of the re-weightings.)**.** *Let $\hat{\pi}_{\varepsilon,\tau}$ the solution of* (UEK) *computed on samples. Let $\mathbf{a} = \hat{\pi}_{\varepsilon,\tau} \mathbf{1}_n$ and $\mathbf{b} = \hat{\pi}_{\varepsilon,\tau}^{\top} \mathbf{1}_n$ be its marginal weights and let $\hat{\eta}_n(\mathbf{x}_i) := n \, a_i$ and $\hat{\xi}_n(\mathbf{y}_i) := n \, b_i$. Then, almost surely, $\hat{\eta}_n(\mathbf{x}_i) \to \eta(\mathbf{x}_i)$ and $\hat{\xi}_n(\mathbf{x}_i) \to \xi(\mathbf{y}_i)$.*

*Proof.* We remind that we here consider $\pi_{\varepsilon}^{\star}$ the solution of problem EK between $\mu$ and $\nu$, and $\tilde{\mu} = \eta \cdot \mu$ and $\tilde{\nu} = \xi \cdot \nu$ denote its marginals. As we saw in the proof of Prop.3.2, using Séjourné et al. (2023a, Prop. 2), one has the existence of $f^{\star} \in C(\mathcal{X})$ and $g^{\star} \in C(\mathcal{Y})$ s.t.

$$\frac{\mathrm{d}\pi_{\varepsilon}^{\star}}{\mathrm{d}(\mu \otimes \nu)}(\mathbf{x}, \mathbf{y}) = \exp\left( \frac{f^{\star}(\mathbf{x}) + g^{\star}(\mathbf{y}) - c(\mathbf{x}, \mathbf{y})}{\varepsilon} \right)$$

Therefore, the relative densities are

$$\eta : \mathbf{x} \mapsto \int_{\mathcal{Y}} \exp\left( \frac{f^{\star}(\mathbf{x}) + g^{\star}(\mathbf{y}) - c(\mathbf{x}, \mathbf{y})}{\varepsilon} \right) \mathrm{d}\nu(\mathbf{y})$$
$$\xi : \mathbf{y} \mapsto \int_{\mathcal{X}} \exp\left( \frac{f^{\star}(\mathbf{x}) + g^{\star}(\mathbf{y}) - c(\mathbf{x}, \mathbf{y})}{\varepsilon} \right) \mathrm{d}\mu(\mathbf{x}) \tag{23}$$

$\square$

Now, let consider $\hat{\pi}_{\varepsilon,\tau}$ the solution of problem EK between $\hat{\mu}_n$ and $\hat{\nu}_n$. Still applying Séjourné et al. (2023a, Prop. 2), one has the existence of $f_n^{\star} \in C(\mathcal{X})$ and $g_n^{\star} \in C(\mathcal{Y})$ s.t.

$$\frac{\mathrm{d}\hat{\pi}_{\varepsilon,\tau}}{\mathrm{d}(\hat{\mu}_n \otimes \hat{\nu}_n)}(\mathbf{x}, \mathbf{y}) = \exp\left( \frac{f_n^{\star}(\mathbf{x}) + g_n^{\star}(\mathbf{y}) - c(\mathbf{x}, \mathbf{y})}{\varepsilon} \right) \tag{24}$$

Writing $\hat{\pi}_{\varepsilon,\tau} = \sum_{i,j=1}^{n} \hat{\pi}_{\varepsilon,\tau}^{ij} \delta_{(\mathbf{x}_i, \mathbf{y}_j)}$, (24) implies that

$$\hat{\pi}_{\varepsilon,\tau}^{ij} = \frac{1}{n^2} \exp\left(\frac{f_n^{\star}(\mathbf{x}_i) + g_n^{\star}(\mathbf{y}_j) - c(\mathbf{x}_i, \mathbf{y}_j)}{\varepsilon}\right) \tag{25}$$

where the potentials $f_n^{\star}$, $g_n^{\star}$ now appear from their values on the samples $\mathbf{x}_i$, $\mathbf{y}_j$. Reminding that $\mathbf{a} = \hat{\pi}_{\varepsilon,\tau} \mathbf{1}_n$ and $\mathbf{b} = \hat{\pi}_{\varepsilon,\tau}^{\top} \mathbf{1}_n$ and combining that with (25), one has:

$$
\begin{aligned}
n\, a_i &= \frac{1}{n} \sum_{j=1}^{n} \exp\left(\frac{f_n^{\star}(\mathbf{x}_i) + g_n^{\star}(\mathbf{y}_j) - c(\mathbf{x}_i, \mathbf{y}_j)}{\varepsilon}\right) \\
n\, b_i &= \frac{1}{n} \sum_{i=1}^{n} \exp\left(\frac{f_n^{\star}(\mathbf{x}_i) + g_n^{\star}(\mathbf{y}_j) - c(\mathbf{x}_i, \mathbf{y}_j)}{\varepsilon}\right)
\end{aligned}
\tag{26}
$$

Almost surely, $\hat{\mu}_n \rightharpoonup \mu$ and $\hat{\nu}_n \rightharpoonup \nu$, so using (Séjourné et al., 2021)[Proposition 10], $f_n^{\star} \to f^{\star}$ and $g_n^{\star} \to g^{\star}$ in sup-norm. Then, define:

$$
\begin{aligned}
\hat{\eta}_n : \mathbf{x} &\mapsto \frac{1}{n} \sum_{j=1}^{n} \exp\left(\frac{f_n^{\star}(\mathbf{x}) + g_n^{\star}(\mathbf{y}_j) - c(\mathbf{x}, \mathbf{y}_j)}{\varepsilon}\right) \\
\hat{\xi}_n : \mathbf{y} &\mapsto \frac{1}{n} \sum_{j=1}^{n} \exp\left(\frac{f_n^{\star}(\mathbf{x}_i) + g_n^{\star}(\mathbf{y}) - c(\mathbf{x}_i, \mathbf{y})}{\varepsilon}\right)
\end{aligned}
\tag{27}
$$

s.t. $\hat{\eta}_n(\mathbf{x}_i) = n\, a_i$ and $\hat{\xi}_n(\mathbf{y}_i) = n\, b_i$. Instead of just showing the convergence in each of these points, we can even show that almost surely, $\hat{\eta}_n \to \eta$ and $\hat{\xi}_n \to \xi$ pointwise. Let us show this result for $\hat{\eta}_n$. First, we define:

$$
\begin{aligned}
h_n : (\mathbf{x}, \mathbf{y}) &\mapsto \exp\left(\frac{f_n^{\star}(\mathbf{x}) + g_n^{\star}(\mathbf{y}) - c(\mathbf{x}, \mathbf{y})}{\varepsilon}\right) \\
h : (\mathbf{x}, \mathbf{y}) &\mapsto \exp\left(\frac{f^{\star}(\mathbf{x}) + g^{\star}(\mathbf{y}) - c(\mathbf{x}, \mathbf{y})}{\varepsilon}\right)
\end{aligned}
\tag{28}
$$

such that $\hat{\eta}_n : \mathbf{x} \mapsto \int_{\mathcal{Y}} h_n(\mathbf{x}, \mathbf{y})\, d\hat{\nu}_n(\mathbf{y})$ and $\eta : \mathbf{x} \mapsto \int_{\mathcal{Y}} h(\mathbf{x}, \mathbf{y})\, d\nu(\mathbf{y})$

Since $f_n^{\star} \to f^{\star}$ on $\mathcal{X}$ and $g_n^{\star} \to g^{\star}$ on $\mathcal{Y}$ in sup-norm, $h_n \to h$ in sup-norm on $\mathcal{X} \times \mathcal{Y}$. Indeed, for $(\mathbf{x}, \mathbf{y}) \in (\mathcal{X} \times \mathcal{Y})$, one has:

$$
\begin{aligned}
&|h_n(\mathbf{x}, \mathbf{y}) - h(\mathbf{x}, \mathbf{y})| \\
&= \left| \exp\left(\frac{f_n^{\star}(\mathbf{x}) + g_n^{\star}(\mathbf{y}) - c(\mathbf{x}, \mathbf{y})}{\varepsilon}\right) - \exp\left(\frac{f^{\star}(\mathbf{x}) + g^{\star}(\mathbf{y}) - c(\mathbf{x}, \mathbf{y})}{\varepsilon}\right) \right| \\
&= \exp\left(\frac{c(\mathbf{x}, \mathbf{y})}{\varepsilon}\right) \left| \exp\left(\frac{f_n^{\star}(\mathbf{x}) + g_n^{\star}(\mathbf{y})}{\varepsilon}\right) - \exp\left(\frac{f^{\star}(\mathbf{x}) + g^{\star}(\mathbf{y})}{\varepsilon}\right) \right| \\
&\leq M_{c,\varepsilon} \left| \exp\left(\frac{f_n^{\star}(\mathbf{x}) + g_n^{\star}(\mathbf{y})}{\varepsilon}\right) - \exp\left(\frac{f^{\star}(\mathbf{x}) + g^{\star}(\mathbf{y})}{\varepsilon}\right) \right|
\end{aligned}
\tag{29}
$$

with $M_{c,\varepsilon} = \sup_{(\mathbf{x}, \mathbf{y}) \in \mathcal{X} \times \mathcal{Y}} \exp\left(\frac{c(\mathbf{x}, \mathbf{y})}{\varepsilon}\right) < +\infty$, since the cost is continuous on the compact $\mathcal{X} \times \mathcal{Y}$, so is $(\mathbf{x}, \mathbf{y}) \mapsto \exp\left(\frac{c(\mathbf{x}, \mathbf{y})}{\varepsilon}\right)$. Then,

$$
\begin{aligned}
&\left| \exp\left(\frac{f_n^{\star}(\mathbf{x}) + g_n^{\star}(\mathbf{y})}{\varepsilon}\right) - \exp\left(\frac{f^{\star}(\mathbf{x}) + g^{\star}(\mathbf{y})}{\varepsilon}\right) \right| \\
&\leq \left| \exp\left(\frac{f_n^{\star}(\mathbf{x})}{\varepsilon}\right) \exp\left(\frac{g_n^{\star}(\mathbf{x})}{\varepsilon}\right) - \exp\left(\frac{f_n^{\star}(\mathbf{x})}{\varepsilon}\right) \exp\left(\frac{g^{\star}(\mathbf{x})}{\varepsilon}\right) \right| \\
&\quad + \left| \exp\left(\frac{f_n^{\star}(\mathbf{x})}{\varepsilon}\right) \exp\left(\frac{g^{\star}(\mathbf{x})}{\varepsilon}\right) - \exp\left(\frac{f^{\star}(\mathbf{x})}{\varepsilon}\right) \exp\left(\frac{g^{\star}(\mathbf{x})}{\varepsilon}\right) \right|
\end{aligned}
\tag{30}
$$

For the first term, one has:

$$
\begin{aligned}
&\left| \exp\left(\frac{f_n^\star(\mathbf{x})}{\varepsilon}\right) \exp\left(\frac{g_n^\star(\mathbf{x})}{\varepsilon}\right) - \exp\left(\frac{f_n^\star(\mathbf{x})}{\varepsilon}\right) \exp\left(\frac{g^\star(\mathbf{x})}{\varepsilon}\right) \right| \\
&= \exp\left(\frac{f_n^\star(\mathbf{x})}{\varepsilon}\right) \left| \exp\left(\frac{g_n^\star(\mathbf{x})}{\varepsilon}\right) - \exp\left(\frac{g^\star(\mathbf{x})}{\varepsilon}\right) \right| \\
&\leq \exp\left(\frac{\|f_n^\star\|_\infty}{\varepsilon}\right) \left| \exp\left(\frac{g_n^\star(\mathbf{x})}{\varepsilon}\right) - \exp\left(\frac{g^\star(\mathbf{x})}{\varepsilon}\right) \right|
\end{aligned}
\tag{31}
$$

First, we can bound uniformly $\exp(\|f_n^\star\|_\infty/\varepsilon)$ since $f_n$ converges in sup-norm, so $(\|f_n^\star\|_\infty)_{n\geq 0}$ is bounded. Then, since $g_n^\star$ convergences in sup-norm, it is uniformly bounded, and since $g^\star$ is continuous on the compact $\mathcal{X}$, it is bounded. Therefore, we can find a compact $K \subset \mathbb{R}$ s.t. $g^\star(\mathcal{X}) \subset K$ and for each $n$, $g_n^\star(\mathcal{X}) \subset K$. Then, applying the mean value theorem to the $C_1$ function $\mathbf{x} \mapsto \exp(\mathbf{x}/\varepsilon)$ on $K$, we can bound:

$$
\left| \exp\left(\frac{g_n^\star(\mathbf{x})}{\varepsilon}\right) - \exp\left(\frac{g^\star(\mathbf{x})}{\varepsilon}\right) \right| \leq \sup_{\mathbf{z} \in K} \frac{1}{\varepsilon} \exp(\tfrac{1}{\varepsilon}\mathbf{z}) \left| g_n^\star(\mathbf{x}) - g^\star(\mathbf{x}) \right|
\tag{32}
$$

Finally, this yields the existence of a constant $M_1 > 0$ s.t.

$$
\left| \exp\left(\frac{f_n^\star(\mathbf{x})}{\varepsilon}\right) \exp\left(\frac{g_n^\star(\mathbf{x})}{\varepsilon}\right) - \exp\left(\frac{f_n^\star(\mathbf{x})}{\varepsilon}\right) \exp\left(\frac{g^\star(\mathbf{x})}{\varepsilon}\right) \right| \leq M_1 \|g_n^\star - g^\star\|_\infty
\tag{33}
$$

Using the same strategy, we get the existence of a constant $M_2 > 0$ s.t.

$$
\left| \exp\left(\frac{f_n^\star(\mathbf{x})}{\varepsilon}\right) \exp\left(\frac{g^\star(\mathbf{x})}{\varepsilon}\right) - \exp\left(\frac{f^\star(\mathbf{x})}{\varepsilon}\right) \exp\left(\frac{g^\star(\mathbf{x})}{\varepsilon}\right) \right| \leq M_2 \|f_n^\star - f^\star\|_\infty
\tag{34}
$$

Combining (33) and (33) with (B.1) and (29), we get that:

$$
|h_n(\mathbf{x}) - h(\mathbf{x})| \leq M_{c,\varepsilon} \left( M_1 \|g_n^\star - g^\star\|_\infty + M_2 \|f_n^\star - f^\star\|_\infty \right)
\tag{35}
$$

from which we can deduce that $h_n \to h$ in sup-norm, from the convergence of $f_n \to f$ and $g_n \to g$ in sup-norm.

Now, we can show the pointwise convergence of $\hat{\eta}_n$. For any $\mathbf{x} \in \mathcal{X}$, one has:

$$
\begin{aligned}
&|\hat{\eta}_n(\mathbf{x}) - \eta(\mathbf{x})| \\
&= \left| \int h_n(\mathbf{x}, \mathbf{y}) \, \mathrm{d}\hat{\nu}_n(\mathbf{y}) - \int h(\mathbf{x}, \mathbf{y}) \, \mathrm{d}\nu(\mathbf{y}) \right| \\
&\leq \left| \int h_n(\mathbf{x}, \mathbf{y}) \, \mathrm{d}\hat{\nu}_n(\mathbf{y}) - \int h(\mathbf{x}, \mathbf{y}) \, \mathrm{d}\hat{\nu}_n(\mathbf{y}) \right| + \left| \int h(\mathbf{x}, \mathbf{y}) \, \mathrm{d}\hat{\nu}_n(\mathbf{y}) - \int h(\mathbf{x}, \mathbf{y}) \, \mathrm{d}\nu(\mathbf{y}) \right| \\
&\leq \int \|h_n - h\|_\infty \, \mathrm{d}\hat{\nu}_n(\mathbf{y}) + \left| \int h(\mathbf{x}, \mathbf{y}) \, \mathrm{d}\hat{\nu}_n(\mathbf{y}) - \int h(\mathbf{x}, \mathbf{y}) \, \mathrm{d}\nu(\mathbf{y}) \right| \\
&= \|h_n - h\|_\infty + \left| \int h(\mathbf{x}, \mathbf{y}) \, \mathrm{d}\hat{\nu}_n(\mathbf{y}) - \int h(\mathbf{x}, \mathbf{y}) \, \mathrm{d}\nu(\mathbf{y}) \right|
\end{aligned}
\tag{36}
$$

Therefore, it almost surely holds that $\hat{\eta}_n(\mathbf{x}) \to \eta(\mathbf{x})$. Indeed, $\|h_n - h\|_\infty \to 0$ since we have shown that $h_n \to h$ in sup-norm. Then, $h$ is continuous on the compact $\mathcal{X} \times \mathcal{Y}$, so it is bounded, so since $\mu_n \rightharpoonup \mu$, we get $\int h \, \mathrm{d}\hat{\nu}_n \to \int h \, \mathrm{d}\nu$. Next, we show similarly that, almost surely, $\hat{\xi}_n \to \xi$ pointwise. This finally yields that $\hat{\eta}_n(\mathbf{x}_i) \to \eta(\mathbf{x}_i)$ and $\hat{\xi}_n(\mathbf{x}_i) \to \xi(\mathbf{y}_i)$

## C  METRICS

We start with introducing general metrics in § C.1, some of which will be used in the metrics introduced in the context of experiments on single-cell data in § C.2.

## C.1 GENERAL METRICS

In the following, we discuss a way how to classify predictions in a generative model. We start with the setting where each mapped sample is to be assigned to a category based on labelled data in the target distribution. We then continue with the case where there are also labels for samples in the source distribution, and this way define a classifier $f_{\text{class}}$ between labels in the source distribution and labels in the target distribution. Building upon this, we assign the classifier $f_{\text{class}}$ an uncertainty score for each prediction. Finally, we define a calibration score assessing the quality of a given uncertainty score.

**Turning a generative model into a classifier** In the following, consider a finite set of samples in the target domain $\mathbf{y}_1, \ldots, \mathbf{y}_M \in \mathcal{Y}$. Assume $\{\mathbf{y}_m\}_{m=1}^M$ allows for a partition $\{\mathbf{y}_m\}_{m=1}^M = \sqcup_{k \in K} \mathcal{T}_k$. Hence, each sample belongs to exactly one class, which we interchangeably refer to as the sample being labelled. Let $T : \mathcal{X} \to \mathcal{Y}$ be a map (deterministic or stochastic), and let $f_{\text{1-NN}} : \mathcal{Y} \to \{\mathcal{T}_k\}_{k=1}^K$ be the 1-nearest neighbor classifier. We obtain a map $g$ from $\mathcal{X}$ to $\{\mathcal{T}_k\}_{k=1}^K \subset \mathcal{Y}$ by the concatenation of $f_{\text{1-NN}}$ and $T$. This map $g$ proves useful in settings when mapped cells are to be categorized, e.g. to assign mapped cells to a cell type.

**A metric to assess the accuracy of a generative model** In the following, assume that the set of samples in the source domain $\mathbf{x}_1, \ldots \mathbf{x}_N$ allows for a partition $\{\mathbf{x}_n\}_{n=1}^N = \sqcup_{k \in K} \mathcal{S}_k$. Note that the number of elements in the partition of both the source and the target domain is set to $K$. We want to construct a classifier $f_{\text{class}}$ assigning each category in the source distribution $\{\mathcal{S}_k\}_{k=1}^K$ probabilistically to a category in the target distribution $\{\mathcal{T}_k\}_{k=1}^K$. Define $f_{\text{class}} : \{\mathcal{S}_k\}_{k=1}^K \to \mathbb{N}^K$ via $(f_{\text{class}}(\mathcal{S}_k))_j = \sum_{\mathbf{x}_n \in \mathcal{S}_k} 1_{\{g(\mathbf{x}_n) = \mathcal{T}_j\}}$ where $g : \mathcal{X} \to \{\mathcal{T}_k\}_{k=1}^K$ was defined above.

Assume that there exists a known one-to-one match between elements in $\{\mathcal{S}_k\}_{k=1}^K$ and elements in $\{\mathcal{T}_k\}_{k=1}^K$. Then we can define a confusion matrix $\mathcal{A}$ with entries $\mathcal{A}_{ij} := \sum_{\mathbf{x}_n \in \mathcal{T}_i} 1_{\{g(\mathbf{x}_n) = \mathcal{S}_j\}}$. In the context of entropic OT the confusion matrix is element-wise defined as

$$\mathcal{A}_{ij} := \sum_{\mathbf{x}_n \in \mathcal{T}_i} 1_{\{f_{\text{1-NN}}(T(\mathbf{x}_n)) = \mathcal{S}_j\}} \tag{37}$$

This way we obtain an accuracy score of the classifier $f_{\text{class}}$ mapping a partition of one set of samples to a partition of another set of samples.

**Calibration score** To assess the meaningfulness of an uncertainty score, we introduce the following calibration score. Assume we have a classifier which yields predictions along with uncertainty estimations. Let $\mathbf{u} \in \mathbb{R}^K$ be a vector containing an uncertainty estimation for each element in $\{\mathcal{S}_k\}_{k=1}^K$. Moreover, let $\mathbf{a} \in \mathbb{R}^K$ be a vector containing the accuracy for each element in $\{\mathcal{S}_k\}_{k=1}^K$. We then define our calibration score to be the Spearman rank correlation coefficient between $u$ and $\mathbf{1}_K - a$, where $\mathbf{1}_K$ denotes the $K-$dimensional vector containing 1 in every entry. In effect, the calibration score is close to 1 if the model assigns high uncertainty to wrong predictions and low uncertainty to true predictions, while the calibration score is close to $-1$ if the model assigns high uncertainty to correct predictions and low uncertainty to wrong predictions.

In the following, we consider a stochastic map $T$. Let $\mathbf{y}_1, \ldots, \mathbf{y}_L \sim \hat{\pi}_\varepsilon(\cdot|\mathbf{x})$ obtained from $T$. To obtain a calibration score for $f_{\text{class}}$ we estimate a statistic $V(\hat{\pi}_\varepsilon(\cdot|\mathbf{x}))$ from the samples $\mathbf{y}_1, \ldots, \mathbf{y}_L$, reflecting an estimation of uncertainty. Then, we let the uncertainty of the prediction of $f_{\text{class}}$ for category $\mathcal{S}_i$ be the mean uncertainty statistic, i.e. $\sum_{\mathbf{x} \in \mathcal{S}_i} \frac{V(\hat{\pi}_\varepsilon(\cdot|\mathbf{x}))}{|\mathcal{S}_i|}$. In effect, for each prediction $f_{\text{class}}(\mathcal{S}_i)$ we get the uncertainty score

$$u_i = \sum_{\mathbf{x} \in \mathcal{S}_i} \frac{V(\hat{\pi}_\varepsilon(\cdot|\mathbf{x}))}{|\mathcal{S}_i|}. \tag{38}$$

**Assessing the uncertainty with the *cos-var* metric** Gayoso et al. (2022) introduce a statistic to assess the uncertainty of deep generative RNA velocity methods from samples of the posterior distribution, which we adapt to the OT paradigm to obtain

$$\text{cos-var}(\hat{\pi}_\varepsilon(\cdot|\mathbf{x})) = \text{Var}_{Y \sim \hat{\pi}_\varepsilon(\cdot|\mathbf{x})}[\text{cos-sim}(Y, \mathbb{E}_{Y \sim \hat{\pi}_\varepsilon(\cdot|\mathbf{x})}[Y])], \tag{39}$$

where cos-sim denotes the cosine similarity. We refer to this metric as cos-var, as it computes the variance of the cosine similarity of samples following the conditional distribution and the conditional mean. We use 30 samples from the conditional distribution to compute this metric.

## C.2 SINGLE-CELL SPECIFIC METRICS

**Cell type / cell lineage transition scores**   As in most single-cell tasks there is no ground truth of matches between cells, we rely on labels of clusters of the data, i.e. on cell types. We then assess the accuracy of a generative model by considering the accuracy of the corresponding classifier $f_{\text{class}}$ as described above. The correct matches between classes have to be considered task-specifically. In the following, we discuss the choice of the labels $\{\mathcal{S}_k\}_{k=1}^K$ and $\{\mathcal{T}_k\}_{k=1}^K$ for different tasks.

- U-GENOT-K for pancreas development: Cells of the developing mouse pancreas (at the time points we consider, i.e. embryonic day 14.5, corresponding to the source distribution, and 15.5, corresponding to the target distribution) can be classified into two lineages (Bastidas-Ponce et al., 2019), which both originate from a *Multipotent* cell population. These lineages are the Acinar (A) lineage, and the ED lineage containing endocrine and dutcal cells. Thus, we define $\{\mathcal{S}_k\}_{k=1}^K = \{\mathcal{T}_k\}_{k=1}^K = \{A, ED\}$ as we know that cells in the A lineage won't develop into cells belonging to the ED lineage, and vice versa.
- U-GENOT-K for perturbation prediction: Each drug was applied to cells belonging to three different cell types/cell lines, namely A549, K562, and MCF7. Hence, we can define $\{\mathcal{S}_k\}_{k=1}^K = \{\mathcal{T}_k\}_{k=1}^K = \{A549, K562, MCF7\}$ as for each perturbed cell we know the cell type at the time of injecting the drug.

**FOSCTTM score**   In the following, we consider a setting where the true match between *samples* is known. The FOSCTTM score ("Fraction of Samples Closer than True Match") measures the fraction of cells which are closer to the true match than the predicted cell. Hence, a random match has a FOSCTTM score of $0.5$, while a perfect match has a FOSCTTM score of $0.0$. In the following we only consider discrete distributions. To define the FOSCTTM score for a map $T : \mathcal{X} \to \mathcal{Y}$, let $\mathbf{x}_1, \ldots \mathbf{x}_K \in \mathcal{X}$ be samples from the source distribution and $\mathbf{y}_1, \ldots \mathbf{y}_K \in \mathcal{Y}$ be samples from the target distribution, such that $\mathbf{x}_k$ and $\mathbf{y}_k$ form a true match. Moreover, let $\hat{\mathbf{y}}_k = T(\mathbf{x}_k)$. Let

$$p_j = \frac{\sum_{k \in K} \mathbf{1}_{\|\mathbf{y}_k - \hat{\mathbf{y}}_j\|_2^2 \leq \|\mathbf{y}_j - \hat{\mathbf{y}}_j\|_2^2}}{|K|} \tag{40}$$

and

$$q_j = \frac{\sum_{k \in K} \mathbf{1}_{\|\mathbf{y}_j - \hat{\mathbf{y}}_k\|_2^2 \leq \|\mathbf{y}_j - \hat{\mathbf{y}}_k\|_2^2}}{|K|} \tag{41}$$

Then, the FOSCTTM score between the predicted target $\{\hat{\mathbf{y}}_k\}_{k \in K}$ and the target $\{\mathbf{y}_k\}_{k \in K}$ is obtained as

$$\text{FOSCTMM}(\{\hat{\mathbf{y}}_k\}_{k \in K}, \{\mathbf{y}_k\}_{k \in K}) = \sum_{k \in K} \frac{p_j + q_j}{2}. \tag{42}$$

## D   DATASETS

### D.1   PANCREAS SINGLE-CELL DATASET

The dataset of the developing mouse pancreas was published in Bastidas-Ponce et al. (2019) and can be downloaded following the guidelines on `https://www.ncbi.nlm.nih.gov/geo/query/acc.cgi?acc=GSE132188`. The full dataset contains measurements of embryonic days 12.5, 13.5, 14.5, and 15.5, while we only consider time points 14.5 and 15.5.

**Benchmark & uncertainty evaluation**   For the benchmark against competing methods we filter the dataset such that we only keep cells belonging to the cell types of the endocrine branch to ensure that learnt transitions are biologically plausible. Moreover, cells annotated as *Ngn3 high cycling* were removed due to its unknown stage in the developmental process (Bastidas-Ponce et al., 2019). The removal is justified by the small number of cells belonging to this cell type and its outlying position in gene expression space. For the uncertainty analysis presented in 2 we use the same

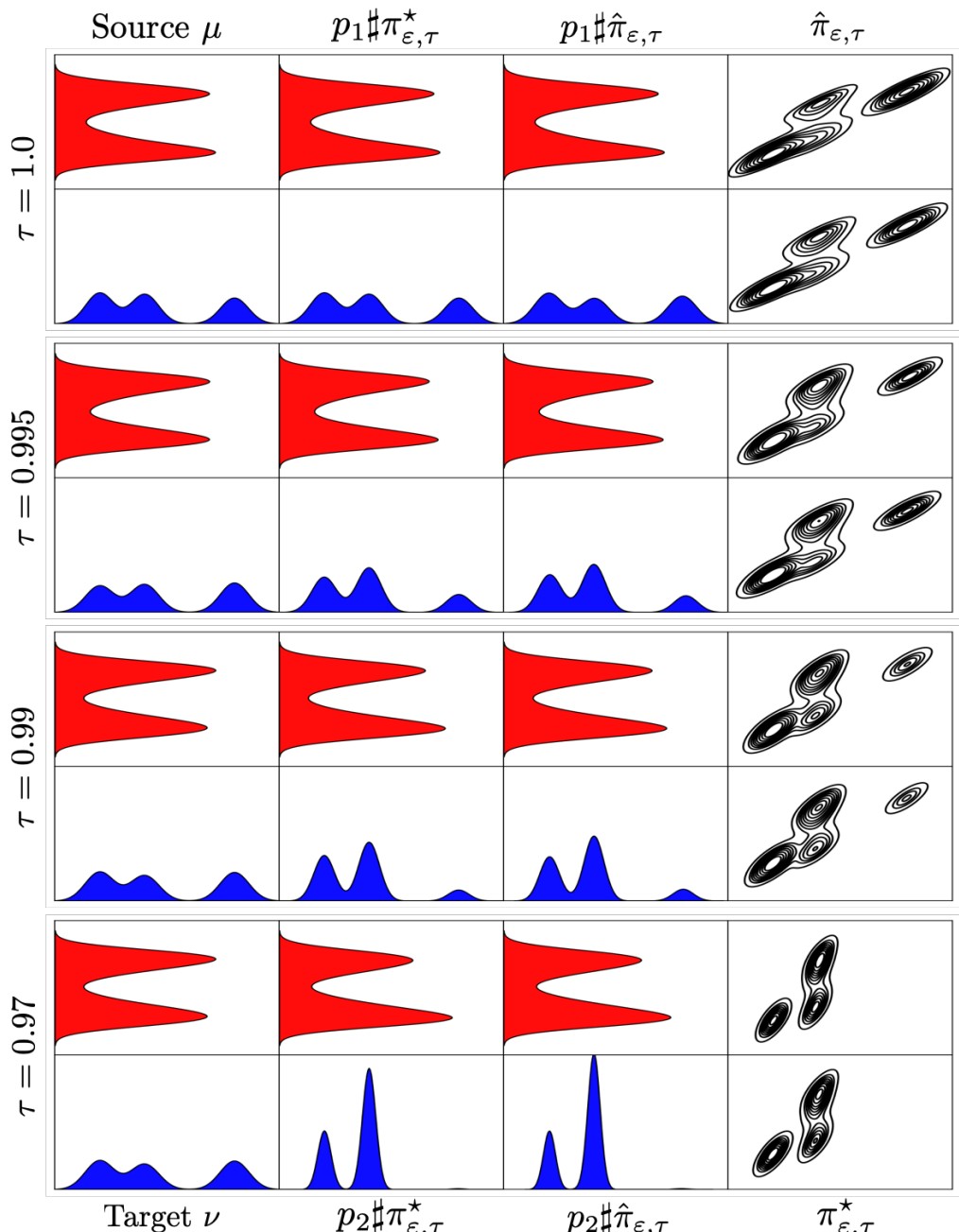

Figure 7: Unbalanced entropic neural optimal transport plan with $\varepsilon = 0.05$ and varying unbalanced-ness parameter $\tau = \tau_1 = \tau_2$. Figure 1 shows the results for $\tau = 0.98$.

dataset. The final list of cell types included can hence be found in figure 14. The benchmark was performed on 30-dimensional PCA space of log-transformed gene expression counts.

**Application in the unbalanced case**  For demonstrating the capabilities of UGENOT-K, we use all cells belonging to E14.5 or E15.5 except for Multipotent cells as these cells can develop into either of the considered cell lineages ED or A, hence the cell type transition score as defined in appendix C.1 could not be computed.

## D.2 DRUG PERTURBATION SINGLE-CELL DATASET

The dataset was published in (Srivatsan et al., 2020). We download the dataset following the instructions detailed on `https://github.com/bunnech/cellot/tree/main`.

For all analyses (figure 3) we computed PCA embeddings on the filtered dataset including the control cells and the corresponding drug only. This ensures the capturing of relevant distributional shifts and hence prevents the model from near-constant predictions as the effect of numerous drugs is weak.

## D.3 HUMAN BONE MARROW SINGLE-CELL DATASET FOR MODALITY TRANSLATION

This dataset contains paired measurements of single-cell RNA-seq readouts and single-nucleus ATAC-seq measurements (Luecken et al., 2021). This means that we have a ground truth one-to-one matching for each cell. We use the processed data provided in moscot (Klein et al., 2023), which can be downloaded following the instructions on `https://moscot.readthedocs.io/en/latest/genapi/moscot.datasets.bone_marrow.html#moscot.datasets.bone_marrow`. This version of the dataset additionally contains a shared embedding for both the RNA and the ATAC data, which we use in the fused term. This embedding was created using a variational autoencoder (scVI (Lopez et al., 2018b)) by integrating the RNA counts of the gene expression dataset and gene activity (Stuart et al., 2021) derived from the ATAC data, a commonly used approximation for gene expression estimation from ATAC data (Heumos et al., 2023).

In RNA space we use the PCA embedding (the dimension of which is detailed in the corresponding experiments), while the embedding used in ATAC space is the given LSI (latent semantic indexing) embedding, followed by a feature-wise L2-normalization as proposed in Demetci et al. (2022).

## E   ADDITIONAL INFORMATION AND RESULTS FOR EXPERIMENTS

If not stated otherwise, the GENOT model configuration follows the setup described in appendix G.

## E.1   1D SIMULATED DATA

While figure 1 shows results for $\tau = \tau_1 = \tau_2 = 0.98$, figure 7 visualizes the influence of $\tau$. While $\tau = 1.0$ corresponds to the fully balanced case, setting $\tau = 0.97$ results in a complete discardment of one mode in the target distribution. The ground truth is computed with a discrete entropy-regularized OT solver (Cuturi et al., 2022).

## E.2   GENOT-K BENCHMARKS

**Benchmark on entropic optimal transport between Gaussians**   In this paragraph, we investigate GENOT's ability to recover EOT couplings that are known analytically, especially in high dimension. We benchmark GENOT against baselines on the task of estimating the EOT plan $\pi_\varepsilon^\star$ for the squared Euclidean cost between multivariate Gaussians, which is known in closed form (Janati et al., 2020), for various dimension $d \in \{2, 8, 32, 128, 256\}$ and entropic regularization stregth $\varepsilon \in \{0.1, 1, 10\}$. We detail the experimental setup below.

- **Data generation:** For various dimension $d \in \{2, 8, 32, 128, 256\}$, we generate a pair of source and target centered Gaussian, by generating their covariance from a Wishart distribution with $d$ degree of freedom and scaling matrix $\Sigma = 0.01 \cdot \mathrm{Id}$.
- **Training details:** We train the models using $30,000$ training samples of each Gaussian. We run GENOT-K as described in appendix G, but iterations were increased to $20,000$. Details on competing methods can be found in appendix F.
- **Evaluation details:** We evaluate the model by computing the Sinkhorn divergence (Feydy et al., 2019a) $S_{\varepsilon_0}(\hat{\pi}_\varepsilon, \pi_\varepsilon^\star)$, for the squared Euclidean cost, between the estimated plan $\hat{\pi}_\varepsilon$ and the actual known plan $\pi_\varepsilon^\star$ on $30,000$ (unseen) testing samples. We stress that $\varepsilon_0 = 0.1$ is fixed across the experiments and does not vary like $\varepsilon \in \{0.1, 1, 10\}$ which is the entropy regularization we

use to estimate the $\hat{\pi}_\varepsilon$ plan. We only use $S_{\varepsilon_0}$ as a valid divergence between probability distributions Feydy et al. (2019a, Theorem 1).

Results are shown in Figures 8, 9 and 10. We can see that GENOT is very competitive, especially in large dimensions $d$, where it slightly outperforms all the baselines, for each $\varepsilon$ value. We can note that DSBM (Shi et al., 2023) does not outperform the other baselines in large dimension, despite not relying on mini-batch OT. This echoes the discussion led on mini-batches and biases in § 3.

$$\varepsilon = 10$$

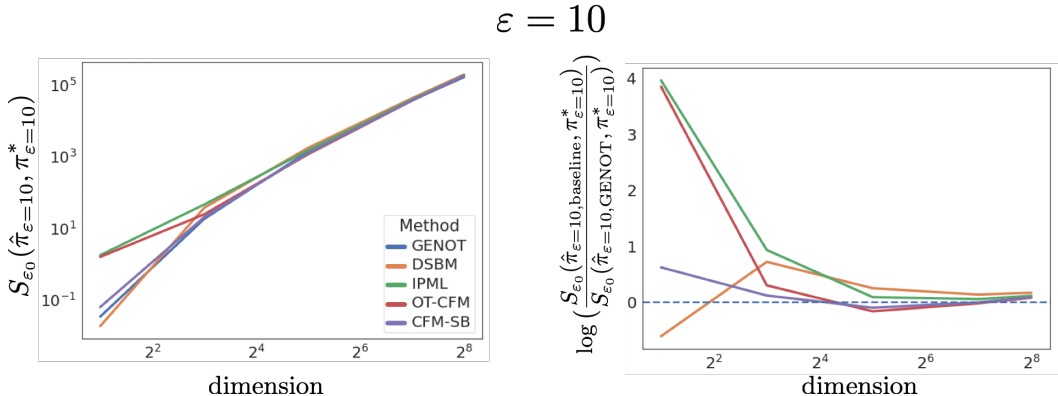

Figure 8: Benchmark on learning the EOT plan with $\varepsilon = 10$ between two Gaussian distributions in different dimensions $d \in \{2, 8, 32, 128, 256\}$. Left: Sinkhorn divergence $S_{\varepsilon_0}$(with $\varepsilon_0 = 0.1$) between 30,000 samples from the estimated plan and 30,000 samples from the true plan. Right: Log-ratio between Sinkhorn divergence of the baseline method and the Sinkhorn divergence of GENOT.

$$\varepsilon = 1$$

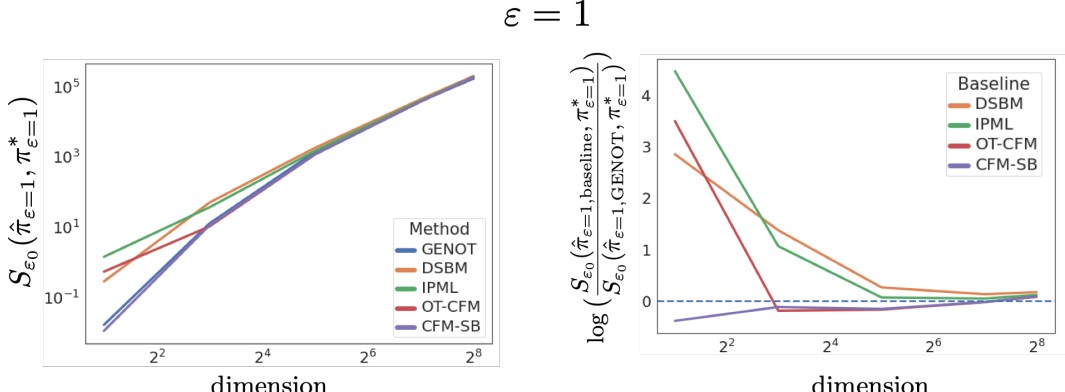

Figure 9: Benchmark on learning the EOT plan with $\varepsilon = 1$ between two Gaussian distributions in different dimensions $d \in \{2, 8, 32, 128, 256\}$. Left: Sinkhorn divergence $S_{\varepsilon_0}$(with $\varepsilon_0 = 0.1$) between 30,000 samples from the estimated plan and 30,000 samples from the true plan. Right: Log-ratio between Sinkhorn divergence of the baseline method and the Sinkhorn divergence of GENOT.

**Influence of batch size**  In this paragraph, we study the influence of the batch size on the performance of GENOT-K. In dimension $d = 32$, and for $\varepsilon \in \{0.1, 1, 10\}$, we evaluate GENOT's ability to estimate the known EOT plan between Gausian using various batch sizes $n \in \{4, 16, 64, 128, 512, 1024, 2048\}$. The number of iterations $n_{\text{iter}}$ depends on the batch size such that the total number of seen samples during training is constant. To avoid *very* long training, we set $n_{\text{iter}} = \min(100, 000, 1, 024 \cdot 10, 000/n)$, where $n$ denotes the batch size. The training is performed on 30,000 samples and evaluation is performed on a test set of the same size. Figure 11 shows the expected behavior. The Sinkhorn divergence between samples from $30, 000$ (computed with entropy regularisation parameter $\varepsilon = 0.01$) decreases with increasing batch size.

**GENOT-K benchmark on the developing mouse pancreas dataset**  The benchmark is performed on the dataset capturing the development of the mouse pancreas (appendix D.1) by trans-

$$\varepsilon = 0.1$$

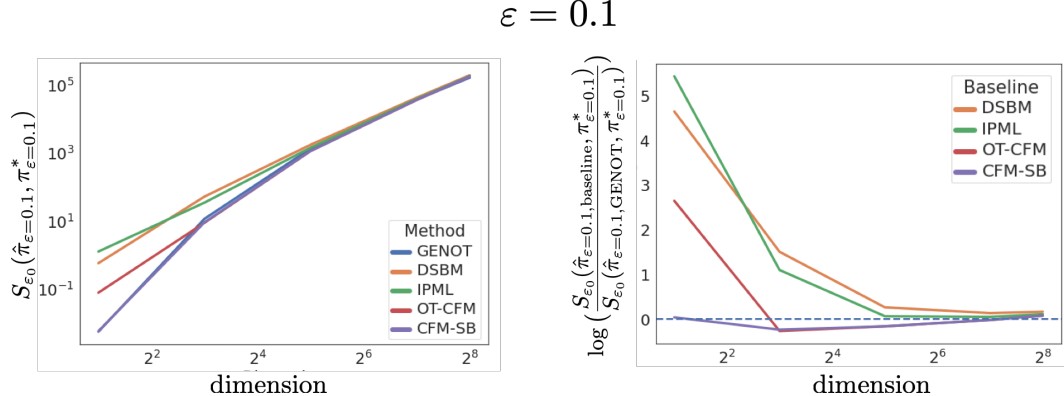

Figure 10: Benchmark on learning the EOT plan with $\varepsilon = 0.1$ between two Gaussian distributions in different dimensions $d \in \{2, 8, 32, 128, 256\}$. Left: Sinkhorn divergence $S_{\varepsilon_0}$ (with $\varepsilon_0 = 0.1$) between 30,000 samples from the estimated plan and 30,000 samples from the true plan. Right: Log-ratio between Sinkhorn divergence of the baseline method and the Sinkhorn divergence of GENOT.

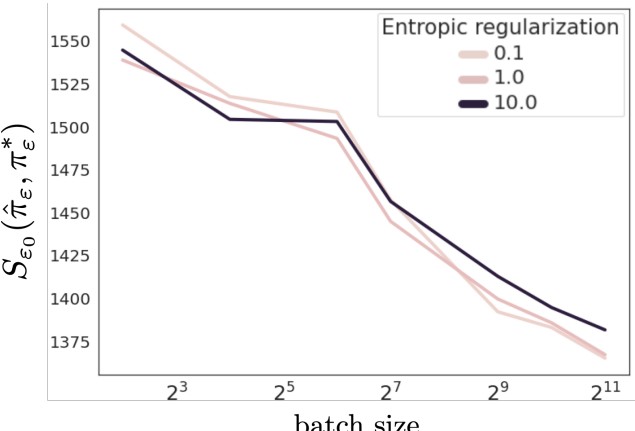

Figure 11: Sinkhorn divergence (with parameter $\varepsilon_0 = 0.1$) between samples from the ground truth plan and samples of the estimated plan depending on the batch size.

porting cells from the early time point to the later time point. For the benchmark the data was standard-normalized to prevent performance issues of models which are not built for data ranges attained by PCA space of the processed single-cell RNA-seq data. The dataset was randomly divided into a 60/40 split of training and test set. As there is no ground truth match between cells in the source and the target distribution, we assess the performance of each model by measuring the Sinkhorn divergence (Feydy et al., 2019b) with regularization parameter $\varepsilon = 1e - 3$ between the test target dataset and the predicted test target dataset, i.e. the pushforward of the test source dataset. GENOT-K was run as described in appendix G, but iterations were increased to $20\_000$. Details on competing methods can be found in appendix F.

We benchmark the models with different entropy regularisation parameters $\varepsilon$. We also report the conditional mean across 30 samples of the pushforward of GENOT-K (denoted by GENOT-K CM) as it is shown to prove useful in many real-world scenarios (see e.g. section 5.2). Figure 12 shows the superior performance of GENOT-K across all entropy regularisation parameters. While GENOT-K CM performs even better, we would like to highlight that the conditional mean of the pushforward is not expected to follow the target distribution. For Scones (Daniels et al., 2021), we observed that training diverged for $\varepsilon \in \{0.1, 0.01, 0.001\}$.

**Interpreting the conditional distribution in mouse pancreas development** Obtaining samples from the conditional distribution allows for an assessment of the uncertainty of the trajectory of a cell. We use the metric $\text{Var}_{Y \sim \hat{\pi}_\varepsilon(\cdot|\mathbf{x})}[\text{cos-sim}(Y, \mathbb{E}_{Y \sim \hat{\pi}_\varepsilon(\cdot|\mathbf{x})}[Y])]$ suggested in Gayoso et al. (2022)

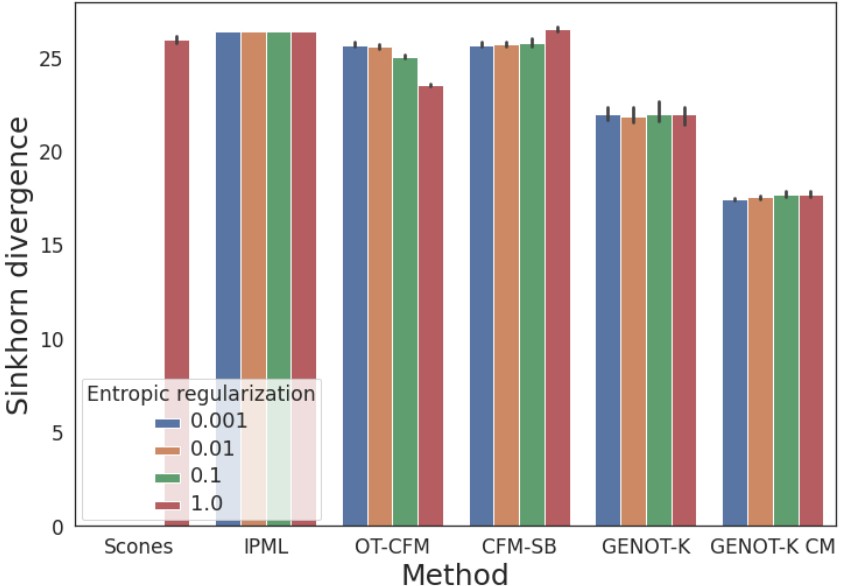

Figure 12: Mean and standard deviation of the Sinkhorn divergence (Feydy et al., 2019b) between test target distribution and pushforward of the test source distribution across three runs on the developing mouse pancreas dataset D.1.

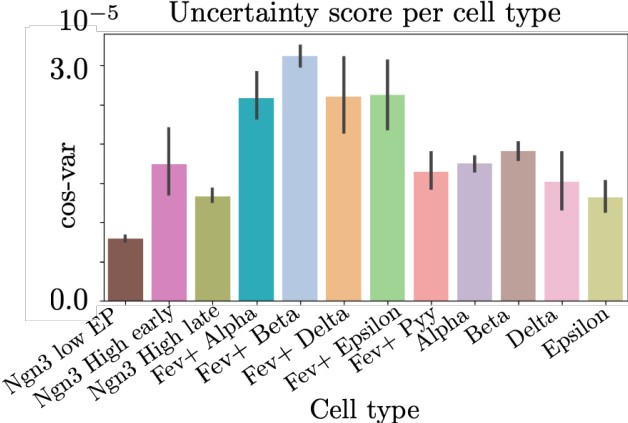

Figure 13: Uncertainty score (appendix C.1) as displayed in figure 2 aggregated to cell type level.

for generative RNA velocity models (appendix C.1). Therefore, we use 30 samples from the conditional distribution.

A cell is expected to have an uncertain trajectory when it awaits a lineage decision. In contrast, cells are expected to have a less uncertain trajectory when their descending population is homogeneous or they belong to a terminal cell state, and hence have committed to a certain lineage.

GENOT-K produces meaningful uncertainty assessments as can be seen from figure 2 and figure 13. Indeed, the Ngn3 low EP population has low variance as all of these cells are expected to transition to the Ngn3 high EP population (Bastidas-Ponce et al., 2019; Klein et al., 2023). In the Ngn3 High early population cells undergo a lineage decision towards the Alpha/Beta or the Delta/Epsilon lineage, hence the uncertainty is higher. Afterwards, cells in the Ngn3 High late or in any Fev+ cell population await fate decisions, while cells in the mature cell types Alpha, Beta, Delta, and Epsilon have committed to a cell type, and hence their trajectory is less uncertain.

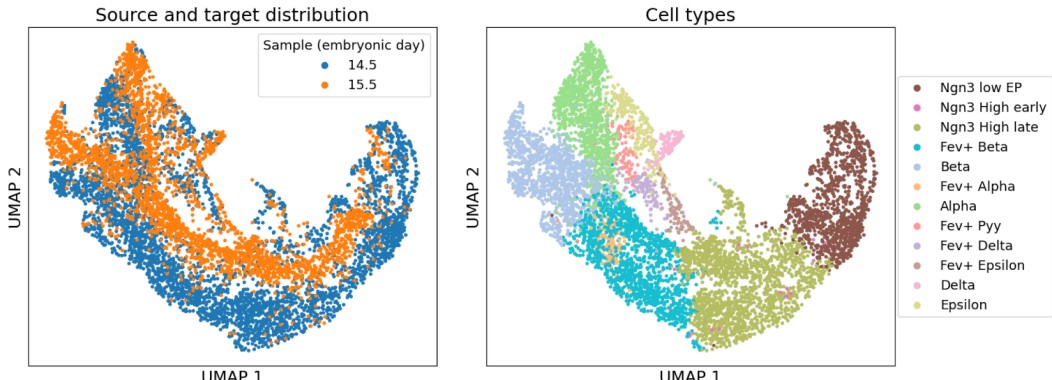

Figure 14: Left: UMAP of the mouse pancreas development dataset colored by sample. We transport samples from embryonic day 14.5 to embryonic day 15.5. Right: UMAP colored by cell type.

| $\tau_1 = \tau_2$ | 1.0 | 0.999 | 0.99 | 0.95 | 0.90 | 0.80 |
|---|---|---|---|---|---|---|
| CTS | 0.660 | 0.677 | 0.714 | 0.786 | 0.815 | 0.86 |
| Sinkhorn divergence | 19.85 | 20.15 | 19.63 | 21.23 | 21.78 | 21.80 |

Table 1: Cell type transition score (CTS, C.2) for U-GENOT-K and Sinkhorn divergence (Feydy et al., 2019b) between target and predicted target. The results reported are the mean across three runs, see table 2 for the variance.

**Unbalancedness in mouse pancreas development** Due to different rates in proliferation (cell birth) and apoptosis (cell death), as well as sampling biases (e.g. due to cell sorting, also referred to as Fluorescence-activated cell sorting), the incorporation of unbalancedness is crucial for numerous datasets in single-cell biology.

We demonstrate this necessity on the mouse pancreas development dataset. The dataset captures two major lineages originating from the multipotent cell population (hence we drop multipotent cells). One lineage (A) develops into Acinar cells and comprises, additionally to Acinar cells, their progenitor population of Tip cells. On the other hand, we have the endocrine/ductal lineage (ED), comprising all remaining cell types. We use a random 60/40 split into train and test data, and run each U-GENOT-K three times with different seeds. Table 1 shows that U-GENOT-K is able to compensate for the undesired distributional shift to a large extent. Table 2 reports the corresponding variance across three runs.

Figure 15 and figure 16 visualize the mean and standard error of the learnt left and right rescaling functions, grouped by evaluations on the training and test set. We expect the rescaling functions within one cell type to attain similar values, hence this is a way to validate whether the learnt rescaling function is meaningful also on the test set. Indeed, we can see that the mean and standard deviation of the learnt left and right rescaling functions are very similar for evaluations on the training and the test dataset.

**Perturbation modeling with GENOT-K and U-GENOT-K** For each drug, we project the single-cell RNA-seq readout of the unperturbed and perturbed cells to a 50-dimensional PCA embedding. Subsequently, we split the data randomly to obtain a train and test set with a ratio of 60%/40%. This preprocessing step holds for both the calibration score experiments and the experiments conducted with U-GENOT-K to assess the influence of unbalancedness to the accuracy score.

The uncertainty score for the calibration study is computed based on 30 samples from the conditional distribution, see appendix C.2.

In figure 17 and figure 18, we visualize the influence of unbalancedness for perturbation modeling with Dacinostat and Tanespimycin, respectively. These experiments were conducted on the full dataset for visualization reasons. While the fitting property seems to be little affected by incorporat-

| $\tau_1 = \tau_2$ | 1.0 | 0.999 | 0.99 | 0.95 | 0.90 | 0.80 |
|---|---|---|---|---|---|---|
| CTS | $4.8 \times 10^{-5}$ | $6.0 \times 10^{-5}$ | $6.5 \times 10^{-4}$ | $1.7 \times 10^{-5}$ | $4.2 \times 10^{-5}$ | $2.5 \times 10^{-5}$ |
| S. div. | $2.9 \times 10^{-2}$ | $5.1 \times 10^{-2}$ | $4.8 \times 10^{-1}$ | $1.4 \times 10^{-2}$ | $5.6 \times 10^{-3}$ | $8.8 \times 10^{-2}$ |

Table 2: Variance of cell type transition score (CTS) and Sinkhorn divergence (S. div.) between target and predicted target of U-GENOT-K across three runs. Means are reported in table 1.

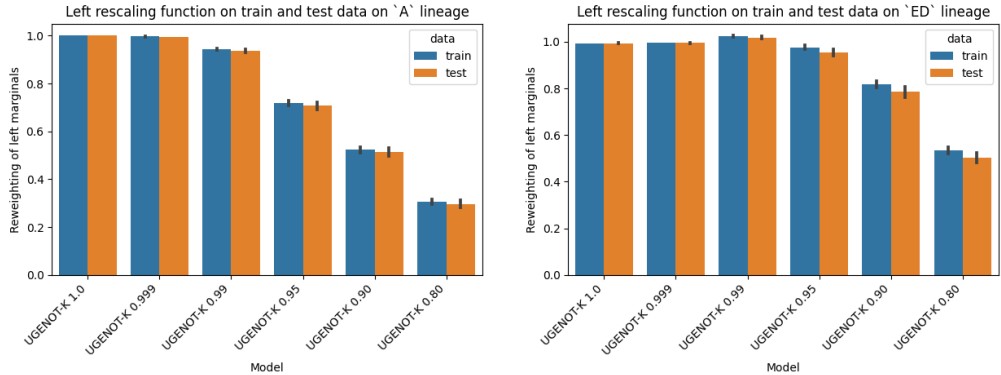

Figure 15: Mean and variance of the predictions of the left rescaling functions on the train and test dataset for different parameters $\tau = \tau_1 = \tau_2$ (denoted as U-GENOT-K $\tau$) across three runs on the full pancreas development dataset. On the left, results are reported for the A lineage, on the right for the ED lineage.

ing unbalancedness (top rows), the cell type clusters are better separated for U-GENOT-K transport plans than for GENOT-K transport plans.

### E.3    GENOT-GW & GENOT-FGW

**GENOT-GW on toy data**    Here, we explicitly visualize the dependence of the conditional distribution on the entropy regularization parameter $\varepsilon$.

**Modality translation with GENOT-GW**    For all experiments, we perform a random 60-40 split for training and test data. All results are reported on the test dataset. The cost matrices of all models were scaled by its mean and the entropy regularization parameter $\varepsilon$ was set to $0.001$. Moreover, the models were trained for 5,000 iterations.

**Modality translation with GENOT-FGW**    For all experiments, we perform a random 60-40 split for training and test data. All results are reported on the test dataset. The cost matrices of all models were scaled by its mean and the entropy regularization parameter $\varepsilon$ was set to $0.001$. Moreover, the models were trained for 20,000 iterations.

Figure 20 reports results of the GENOT-FGW model with interpolation parameter $\alpha = 0.7$. While the Sinkhorn divergences are not comparable with results of the GENOT-GW model due to the respective target distributions living in different spaces, we can compare GENOT-GW with GENOT-FGW with the FOSCTTM score. Figure 20 shows that GENOT-FGW strikingly outperforms GENOT-GW, hence the incorporation of the fused term is crucial for a good performance. At the same time, it is important to mention that the GW terms add valuable information to the problem setting, which can be derived from the results for GENOT-FGW with $\alpha = 0.3$ presented in figure 20. Here, the higher influence of the fused term causes the model to perform overall worse. Interestingly, the geodesic cost approximation performs significantly worse than the squared Euclidean cost with respect to the FOSCTTM score. We note that the construction of the geodesic cost involves multiple hyperparameters, which we did not optimize for. Yet, the fitting term, measured with the Sinkhorn divergence, does not suffer significantly from the performance loss with respect to the FOSCTTM score.

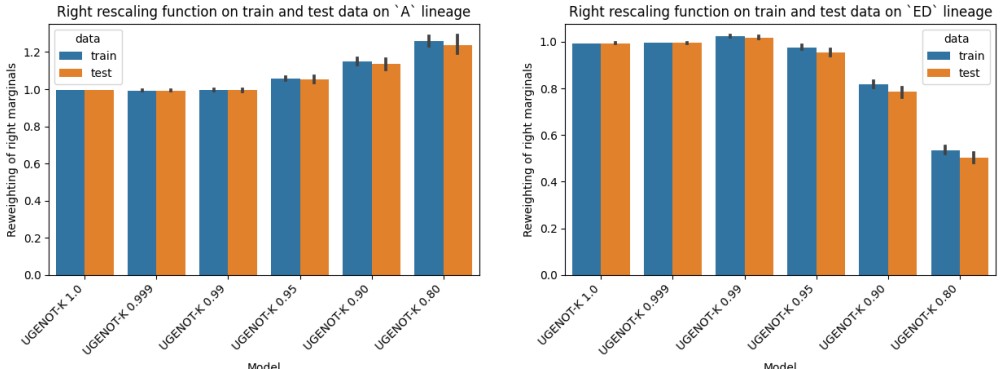

Figure 16: Mean and variance of the predictions of the right rescaling functions on the train and test dataset for different parameters $\tau = \tau_1 = \tau_2$ (denoted as U-GENOT-K $\tau$) across three runs on the full pancreas development dataset. On the left, results are reported for the A lineage, on the right for the ED lineage.

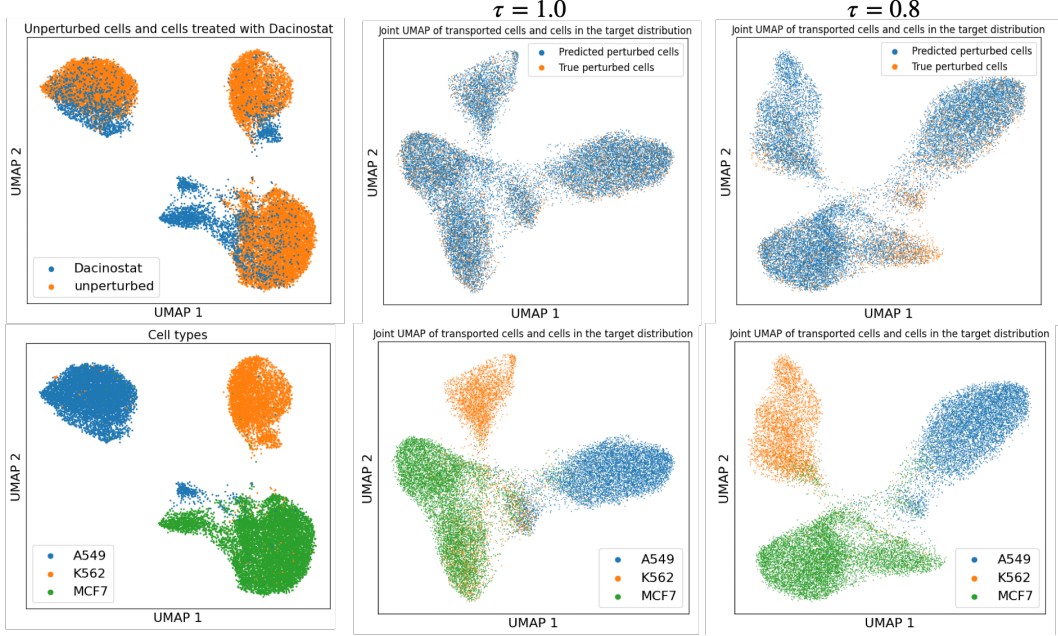

Figure 17: Visual assessment of the influence of unbalancedness in modeling cellular predictions to the cancer drug Dacinostat. In the left column, the source and target distribution are jointly plotted with cells colored by whether they belong to the source (unperturbed) or the target (perturbed) distribution (top), and which cell type they belong to (bottom). In the center column, we plot a UMAP embedding of target and predicted target distribution. The top plot colors cells according to whether a cell belongs to the target distribution or the predicted target distribution. The bottom plot is colored by cell type. The cell type of the predicted target distribution is the cell type of the pre-image of the predicted cell. The right column visualizes the same results, but this time obtained from U-GENOT-K with unbalancedness parameters $\tau = \tau_1 = \tau_2 = 0.8$.

Moreover, we can visualize the optimality and fitting term in a UMAP embedding (McInnes et al., 2018). To demonstrate the robustness of our model, we train a GENOT-FGW model with $\varepsilon = 0.01$, $\alpha = 0.5$ and the Euclidean distance on 60% of the dataset (38 dimensions for the ATAC LSI embedding, 50 dimensions for the RNA PCA embedding, and 28 dimensions for the VAE embedding in the fused term) and evaluate the learnt transport plan visually. Figure 5 shows the joint UMAP embedding of predicted target and target, the full legend of cell types can be found

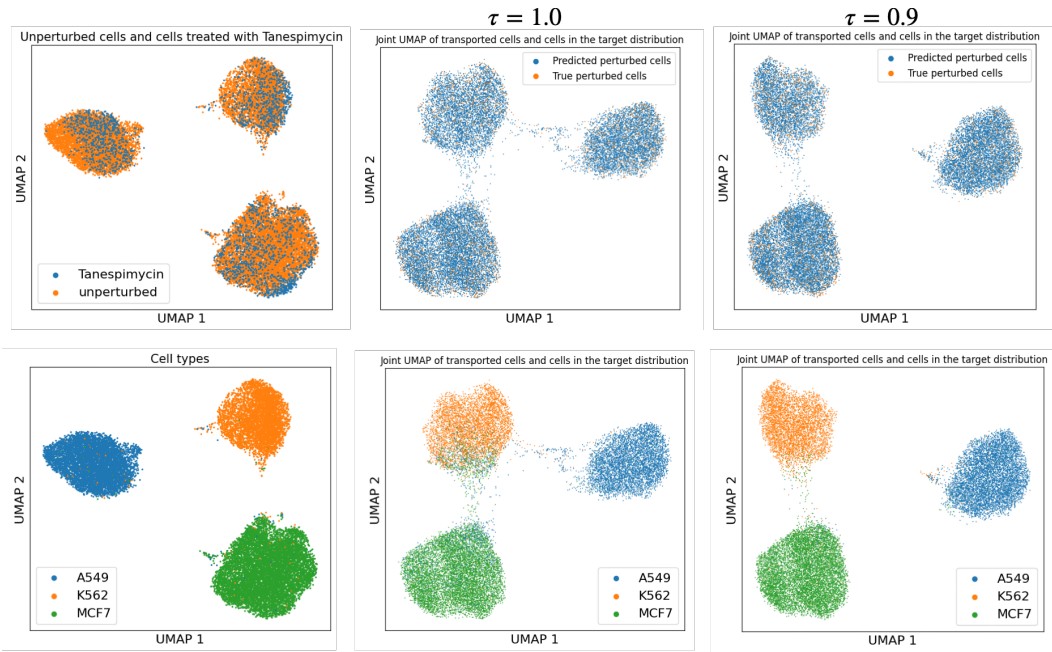

Figure 18: Visual assessment of the influence of unbalancedness in modeling cellular predictions to the cancer drug Tanespimycin. In the left column, the source and target distribution are jointly plotted with cells colored by whether they belong to the source (unperturbed) or the target (perturbed) distribution (top), and which cell type they belong to (bottom). In the center column, we plot a UMAP embedding of target and predicted target distribution. The top plot colors cells according to whether a cell belongs to the target distribution or the predicted target distribution. The bottom plot is colored by cell type. The cell type of the predicted target distribution is the cell type of the pre-image of the predicted cell. The right column visualizes the same results, but this time obtained from U-GENOT-K with unbalancedness parameters $\tau = \tau_1 = \tau_2 = 0.9$.

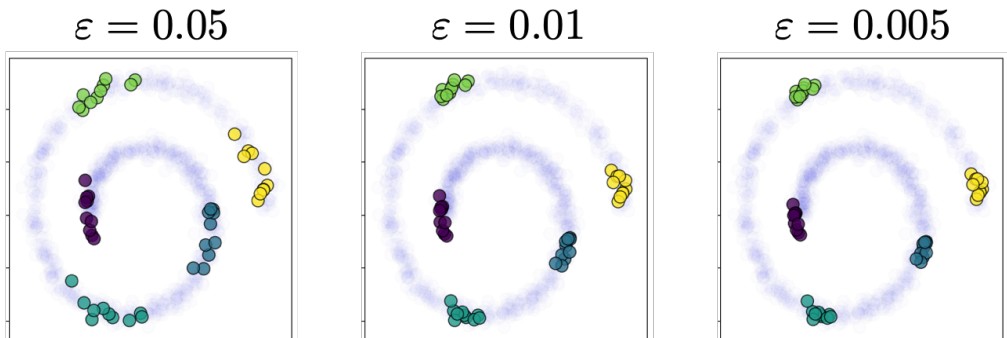

Figure 19: Conditional distribution $\hat{\pi}_\varepsilon(\cdot|\mathbf{x})$ for GENOT-GW models trained with different entropy regularization parameter $\varepsilon$. The setup is the same as in figure 4, in effect we transport a three-dimensional Swiss roll to a two-dimensional spiral, which is colored in blue (with high transparency). The source distribution as well as the data points which are conditioned on are visualized in figure 4.

in figure 22. Qualitatively, a good mix between data points of the predicted target and the target distribution suggests a good fitting term. Optimality of the mapping can be visually assessed by considering to what extent cell types are mixed (low optimality) or separated from other cluster (high optimality). Similarly, figure 23 and figure 24 show the results based on a UMAP embedding created on the fused space (space corresponding to the fused term) only and on a UMAP embedding created from the GW space (term corresponding to the GW target term) only, respectively. Note that these UMAP embeddings were created based on a subspace of the space the FGW problem lives

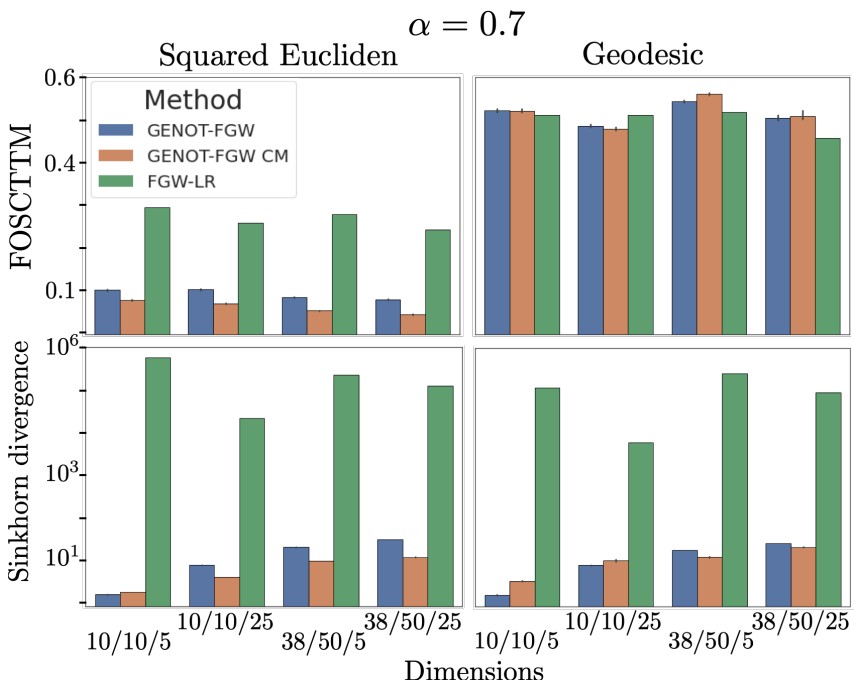

Figure 20: Mean and standard deviation (across three runs) of the FOSCTTM score (top) and the Sinkhorn divergence (bottom) of GENOT-FGW and discrete FGW with linear regression for out-of-sample estimation. Experiments are categorized by the numbers $d_1/d_2/d_3$, where $d_1$ is the dimension of the space corresponding to the GW of the source distribution, $d_2$ is the dimension of the space corresponding to the GW of the target distribution, and $d_3$ is the dimension of the shared space. Results are reported for the interpolation parameter $\alpha = 0.7$. The best performing configuration is GENOT-FGW CM on the embeddings of dimension (38/50/25) with a mean FOSCTTM score of $0.048$.

on, in particular we do *not* train a GENOT-K or GENOT-GW model. We can see that the target distribution is well matched in both of these spaces separately.

We observed that taking the conditional mean improves results on the FOSCTTM score, but can impair the fitting property. Indeed, the mixing rate between data points belonging to the target and data points belonging to the predicted target seems to be slightly worse when considered in the joint embedding as well as when considering only the fused space and only considering the quadratic space (24) .

**Modality translation with U-GENOT-FGW**   To simulate a setting where there is not a match for certain cells in the gene expression dataset, we choose to drop the cells labelled as Proerythroblasts, Erythroblasts, and Normoblasts as these cells form a lineage, developing into mature Reticulocytes (not present in the dataset). Thus, they are similar in their cellular profile while being clearly distinguishable from the remaining cells.

While we keep the right marginals constant, as we have a true match for each cell in the target distribution, we introduce unbalancedness in the source marginals. It is important to note that the influence of the unbalancedness parameters are affected by the number of samples, as well as the entropy regularization parameter $\epsilon$. To demonstrate the robustness of GENOT-FGW with respect to hyperparameters, we still choose $\alpha = 0.7$, but this time set $\varepsilon = 5 \cdot 10^{-3}$. We use 50-dimensional PCA-space for the Gromov term in the RNA space, 38-dimensional LSI-space for the Gromov term in the ATAC space, and a 30-dimensional VAE-embedding for the shared space.

The computation of the growth rates for the discrete setting is described in appendix F.2. We perform a random 60-40 split to divide the data into training and test set. The FOSCTTM score only considers those cells which have a true match, i.e. cells in the source distribution belonging to the

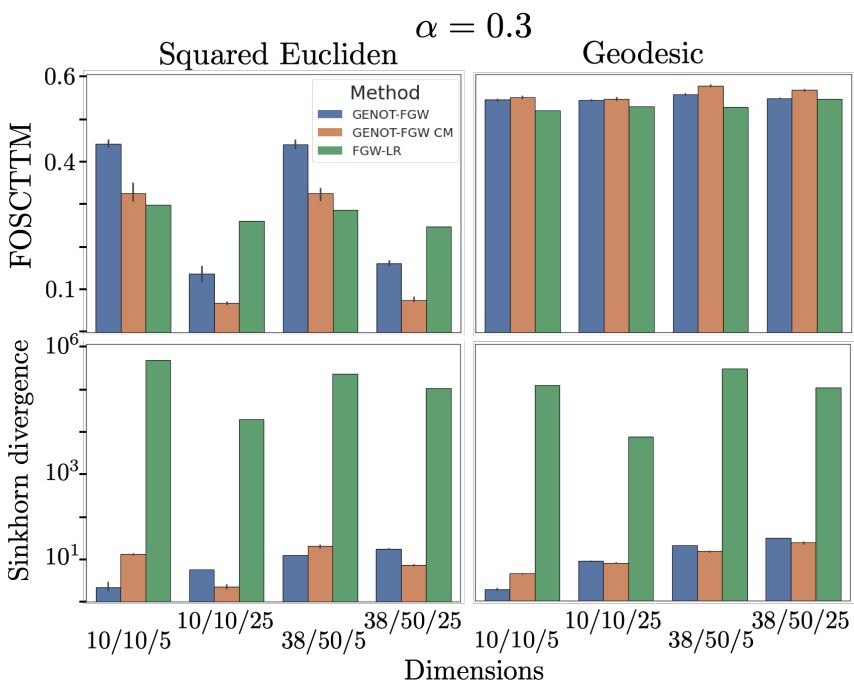

Figure 21: Mean and standard deviation (across three runs) of the FOSCTTM score (top) and the Sinkhorn divergence (bottom) of GENOT-FGW and discrete FGW with linear regression for out-of-sample estimation. Experiments are categorized by the numbers $d_1/d_2/d_3$, where $d_1$ is the dimension of the space corresponding to the GW of the source distribution, $d_2$ is the dimension of the space corresponding to the GW of the target distribution, and $d_3$ is the dimension of the shared space. Results are reported for the interpolation parameter $\alpha = 0.3$. The best performing configuration is GENOT-FGW CM on the embeddings of dimension (10/10/25) with a mean FOSCTTM score of 0.068.

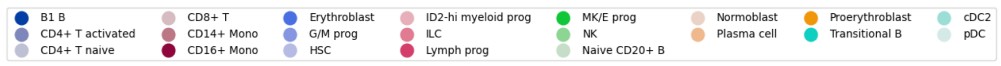

Figure 22: Complete legend of cell types for figures 5, 23, and 24.

Normoblast, Erythroblast, and Proerythroblast cell types are not taken into account as their true match was removed from the target distribution.

We assess the performance w.r.t. the FOSCTTM score to ensure that the model still learns meaningful results, and consider the average reweighting function $\hat{\eta}$ per cell type (appendix E.3). We consider two values ($\tau_1 = 0.8$ and $\tau_1 = 0.3$) of the left unbalancedness parameter, while $\tau_2 = 1.0$ as for every cell in the target distribution there exists the true match in the source distribution. Table 3 shows that U-GENOT-FGW learns more meaningful reweighting functions than discrete UFGW as the average rescaling function on the left-out cell types is closer to 0, while the mean value of the rescaling function on all remaining cell types ("other") is closer to 1. At the same time, U-GENOT-FGW yields lower FOSCTMM scores and hence learns more optimal couplings.

Table 5 shows the variance across three runs, demonstrating the stability of both the learnt rescaling function as well as the performance with respect to the FOSCTTM score.

Figure 25 and figure 26 show the mean and the standard deviation of the learnt growth rates per cell type. First, it is interesting to see that Normoblasts have the lowest mean of rescaling function evaluations (for both discrete UFGW and U-GENOT-FGW), which is due to them being most mature among the left out cell types and hence being furthest away in gene expression space / ATAC space from the common origin of all cells, the HSC cluster. Moreover, it is obvious that the standard deviation of the reweighting function (across cells in one cell type) is much smaller for U-GENOT-

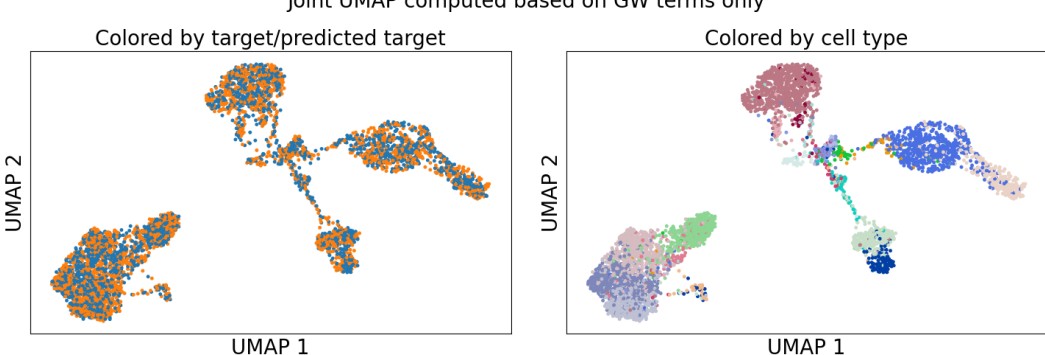

a) UMAP embedding created on the space corresponding to the fused term only.

b) UMAP embedding created on the space corresponding to the target GW term only.

Figure 23: UMAP embeddings of predicted target and target. Left panels: Cells are colored based on whether they belong to the target distribution or the predicted target distribution. Right: Cells are colored according to their cell type. For cells which belong to the predicted target distribution, the cell type is defined as the cell type of the preimage. Results are shown on the test data set, corresponding to 40% of the full dataset.

Table 3: Mean value of the rescaling function per cluster for U-GENOT-FGW and discrete unbalanced FGW together with the FOSCTTM scores across three runs. Table 5 reports the variances for the GENOT-GW models.

| model ($\tau_1$) | Normoblast | Erythroblast | Proerythroblast | other | FOSCTTM |
|---|---|---|---|---|---|
| Discrete UFGW (0.8) | 0.788 | 0.820 | **0.842** | **0.945** | 0.258 |
| U-GENOT-FGW (0.8) | **0.622** | **0.733** | 0.894 | 1.077 | **0.131** |
| Discrete UFGW (0.3) | 0.591 | 0.586 | 0.734 | 0.761 | 0.311 |
| U-GENOT-FGW (0.3) | **0.295** | **0.430** | **0.554** | **1.186** | **0.162** |

FGW than for discrete UFGW. This is desirable as cells within one cell type are very similar in their ATAC profile.

## F    COMPETING METHODS

In the following, we discuss the setup of the competing methods. In particular, we discuss the setup for the benchmark on the pancreas dataset in section F.1 and discuss linear regression-based out of sample estimation for discrete Gromov in section F.2.

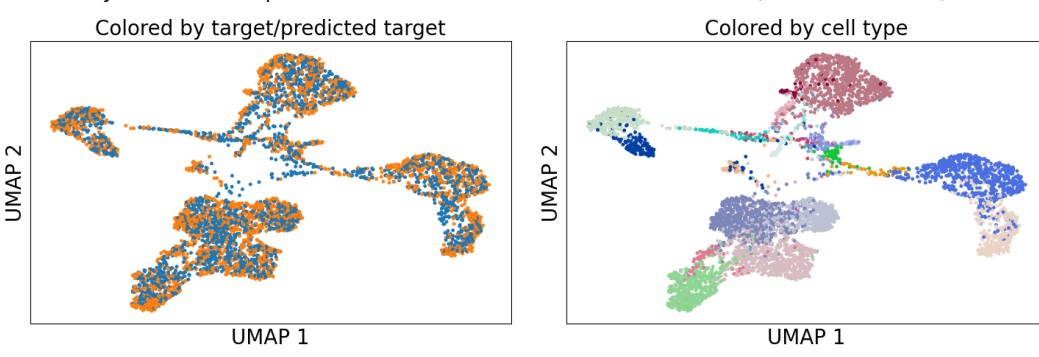

a) UMAP embedding.

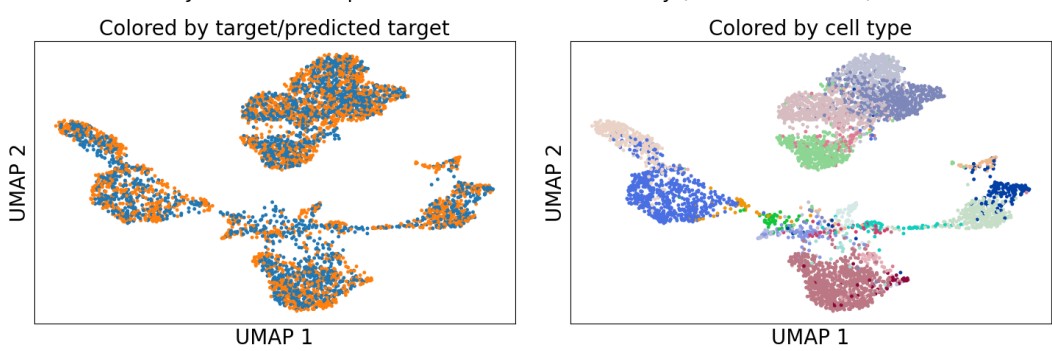

b) UMAP embedding created on the space corresponding to the fused term only.

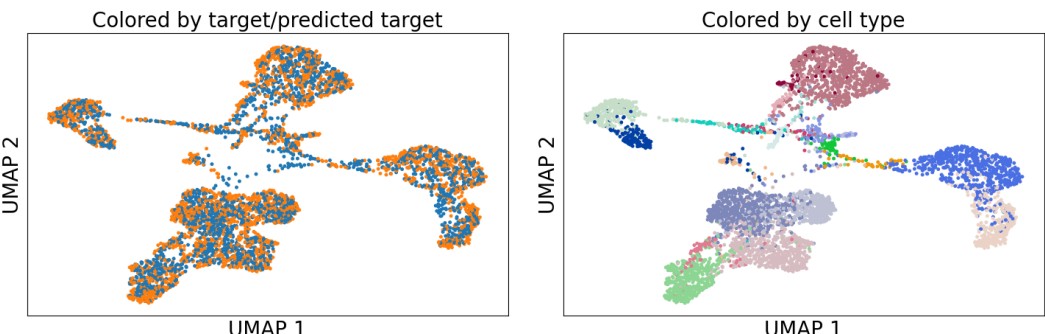

c) UMAP embedding created on the space corresponding to the target GW term only.

Figure 24: UMAP embeddings of predicted target and target for GENOT-FGW CM. Left panels: Cells are colored based on whether they belong to the target distribution or the predicted target distribution. Right: Cells are colored according to their cell type. For cells which belong to the predicted target distribution, the cell type is defined as the cell type of the preimage. Results are shown on the test data set, corresponding to 40% of the full dataset.

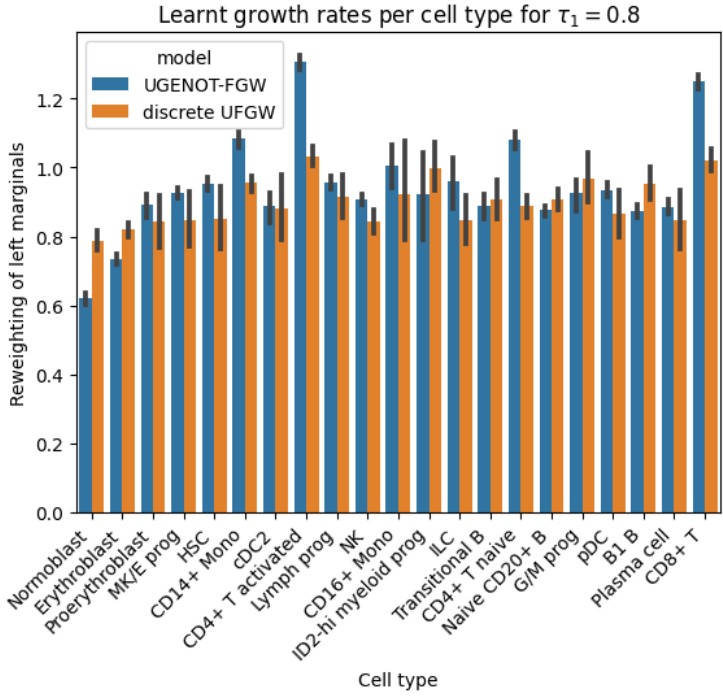

Figure 25: Comparison of learnt growth rates of discrete UFGW and U-GENOT-FGW aggregated to cell type level for unbalancedness parameters $\tau_1 = 0.8$ and $\tau_2 = 1$.

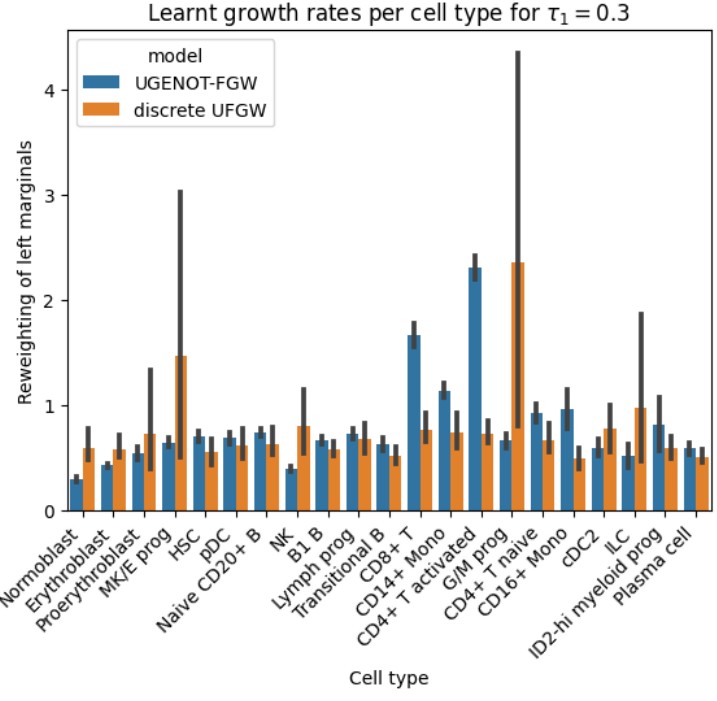

Figure 26: Comparison of learnt growth rates of discrete UFGW and U-GENOT-FGW aggregated to cell type level for unbalancedness parameters $\tau_1 = 0.3$ and $\tau_2 = 1$.

Table 4: Comparison of reweighting functions learnt by U-GENOT-FGW and discrete unbalanced FGW

| model ($\tau_1$) | Normoblast | Erythroblast | Proerythroblast | other | FOSCTTM |
|---|---|---|---|---|---|
| U-GENOT-FGW (0.8) | $3 \cdot 10^{-6}$ | $2 \cdot 10^{-5}$ | $1 \cdot 10^{-4}$ | $5 \cdot 10^{-5}$ | $3 \cdot 10^{-5}$ |
| U-GENOT-FGW (0.3) | $2 \cdot 10^{-6}$ | $9 \cdot 10^{-6}$ | $5 \cdot 10^{-4}$ | $8 \cdot 10^{-4}$ | $9 \cdot 10^{-5}$ |

Table 5: Variance of the mean of the learnt rescaling function per cell type for U-GENOT-FGW across three different seeds.

## F.1 ENTROPIC NEURAL OT METHODS

As we choose the same architecture and hyperparameters for the GENOT-models as described in appendix G across experiments and problem settings, we do *not* optimize the hyperparameters of competing methods. In the following, we outline the code based on which we run the benchmarks.

**Daniels et al. (2021)** We use the configuration provided in `https://github.com/mdnls/scones-synthetic` for the Gaussian data.

**Tong et al. (2023b)** We use the configuration of the notebook for the single-cell experiments in `https://github.com/atong01/conditional-flow-matching/blob/main/examples/notebooks/single-cell_example.ipynb`.

**Tong et al. (2023a)** We use the configuration of the notebook for the single-cell experiments in `https://github.com/atong01/conditional-flow-matching/blob/main/examples/notebooks/single-cell_example.ipynb`.

**Vargas et al. (2021)** We follow the code provided at `https://github.com/AforAnonyMeta/IPML-2548`, specifically the configuration for the single-cell experiments (`https://github.com/AforAnonyMeta/IPML-2548/blob/main/script/EB_Dataset.py`). We set the *gp_prior* to *None* as the model runs out of memory otherwise (for the given datasets).

**Shi et al. (2023)** We follow the code provided at `https://github.com/yuyang-shi/dsbm-pytorch`, and adapt the configuration provided for the Gaussian experiments (`https://github.com/yuyang-shi/dsbm-pytorch/blob/main/conf/gaussian.yaml`).

## F.2 REGRESSION FOR OUT-OF-SAMPLE DATA POINTS

Out-of-sample prediction for GW has been considered in Alvarez-Melis & Jaakkola (2018). Yet, their methods rely on an orthogonal projection, which only works if both the sample size and the feature dimensions are the same in both spaces. Hence, we rely on a barycentric projection for in-sample data points. For out-of-sample data points we project a data point onto the training set and apply the barycentric projection to the linear combination of points in the in-sample distribution. Let $\mathbf{X} \in \mathbb{R}^{n \times d}$ be the matrix containing $n$ in-sample data points.

Then, for a data point in the source distribution $\mathbf{x} \in \mathbb{R}^d$, let

$$\hat{\beta}_{\mathbf{x}} = \arg\min_{\beta \in \mathbb{R}^n} \|\hat{\mathbf{x}} - \mathbf{X}^T \beta\|_2^2 \tag{43}$$

where the sum is taken over the $n$ in-sample data points. Moreover, let $p_i = \sum_{j=1}^m \Pi_{ij}$. Then, the barycentric projection of a point in the source distribution is given as

$$\hat{\mathbf{y}} = \sum_{i=1}^n \frac{\hat{\beta}_i}{p_i} \sum_{j=1}^m \Pi_{ij} \mathbf{y}_j \in \mathcal{Y}. \tag{44}$$

Similarly, we can apply this procedure to estimate rescaling factors in the unbalanced setting. To ensure non-negativity of the rescaling function, we perform regressino with non-negative weights:

$$\hat{\alpha}_{\mathbf{x}} = \arg\min_{\alpha \in \mathbb{R}^n_{\geq 0}} \|\hat{\mathbf{x}} - \mathbf{X}^T \alpha\|_2^2 \tag{45}$$

To estimate the rescaling function for a data point $\hat{x}$, the estimated left rescaling function is given as

$$\hat{\eta} = \sum_{i=1}^{n} \hat{\alpha}_i \eta_i \in \mathbb{R} \tag{46}$$

where $\{\eta_i\}_{i=1}^{n}$ is the set of reweighting function evaluations of in-sample data points.

## G  IMPLEMENTATION

The GENOT framework is implemented in JAX Bradbury et al. (2018). Discrete OT solvers are provided by OTT-JAX Cuturi et al. (2022).

### G.1  PARAMETERIZATION OF THE VECTOR FIELD

The vector field is parameterized with a feed-forward neural network which takes as input the time, the condition (i.e. the samples from the source distribution) and the latent noise. Each of these input vectors are independently embedded by one block of layers before the embeddings are concatenated and applied to another block of layers, followed by one output layer. If not stated otherwise, one block of layers consists of 8 layers of width 256 with *silu* activation functions.

### G.2  PARAMETERIZATION OF THE RESCALING FUNCTIONS

Rescaling functions are parameterized as feed-forward neural networks with 5 layers of width 128, followed with a final *softplus* activation function to ensure non-negativity.

### G.3  TRAINING DETAILS

In the following, we report default values for different parameters of the GENOT models. If not stated otherwise in the corresponding experiments section, these parameters are used:

- number of training iterations: 10,000
- optimizer: AdamW with learning rate $1e-4$ and weight decay $1e-10$ (also for learning the rescaling functions)
- entropy regularisation parameter $\varepsilon = 1e-2$
- by default, we do not scale the cost matrix for the computation of the discrete OT solver
- cost function: squared Euclidean distance (we always use the same cost for all terms, even if it would be possible to choose different costs in separate spaces in the GW and FGW settings)
- batch size: 1024
- number of samples from the conditional distribution: 1

When using the graph distance, we construct a k-nearest neighbor graph with $batch\_size + 1$ number of edges. For the approximation of the heat kernel, we use the default parameters provided by the implementation in OTT-JAX (Cuturi et al., 2022).

### G.4  CODE

In the following, we provide python files containing the implementation of GENOT, and two workflows to reproduce results in the paper.

**The GENOT model**    The following code displays the GENOT model.

```python
import collections
import types
from functools import partial
from typing import Any, Callable, Dict, Literal, Optional, Tuple, Type, Union
```

```python
import diffrax
import optax
import tensorflow as tf
import tensorflow_datasets as tfds
from flax.training.train_state import TrainState
from tqdm import tqdm

import jax
import jax.numpy as jnp
from ott.geometry import costs, geometry, graph, pointcloud
from ott.problems.linear import linear_problem
from ott.problems.quadratic import quadratic_problem
from ott.solvers import was_solver
from ott.solvers.linear import sinkhorn
from ott.solvers.nn.models import ModelBase
from ott.solvers.quadratic import gromov_wasserstein

Match_fn_T = Callable[
    [jax.random.PRNGKeyArray, jnp.array, jnp.array], Tuple[jnp.array, jnp.array, jnp.array, jnp.array]
]
Match_latent_fn_T = Callable[[jax.random.PRNGKeyArray, jnp.array, jnp.array], Tuple[jnp.array, jnp.array]]

def sample_conditional_indices_from_tmap(
    key: jax.random.PRNGKeyArray,
    tmat: jnp.ndarray,
    k_samples_per_x: Union[int, jnp.ndarray],
    left_marginals: Optional[jnp.ndarray],
    *,
    is_balanced: bool,
) -> Tuple[jnp.array, jnp.array]:
    if not is_balanced:
        key, key2 = jax.random.split(key, 2)
        indices = jax.random.choice(
            key=key2, a=jnp.arange(len(left_marginals)), p=left_marginals, shape=(len(left_marginals),)
        )
    else:
        indices = jnp.arange(tmat.shape[0])
    tmat_adapted = tmat[indices]
    indices_per_row = jax.vmap(
        lambda tmat_adapted: jax.random.choice(
            key=key, a=jnp.arange(tmat.shape[1]), p=tmat_adapted, shape=(k_samples_per_x,)
        ),
        in_axes=0,
        out_axes=0,
    )(tmat_adapted)

    return jnp.repeat(indices, k_samples_per_x), indices_per_row % tmat.shape[1]

class GENOT:
    def __init__(
        self,
        neural_net: Union[Type[ModelBase], Tuple[Type[ModelBase], Type[ModelBase]]],
        input_dim: int,
        output_dim: int,
        iterations: int,
        ot_solver: Type[was_solver.WassersteinSolver],
        optimizer: Optional[Any] = None,
        k_noise_per_x: int = 1,
        t_offset: float = 1e-5,
        epsilon: float = 1e-2,
        cost_fn: Union[costs.CostFn, Literal["graph"]] = costs.SqEuclidean(),
        solver_latent_to_data: Optional[Type[was_solver.WassersteinSolver]] = None,
        latent_to_data_epsilon: float = 1e-2,
        latent_to_data_scale_cost: Any = 1.0,
        scale_cost: Any = 1.0,
        graph_kwargs: Dict[str, Any] = types.MappingProxyType({}),
        fused_penalty: float = 0.0,
        split_dim: int = 0,
        mlp_eta: Callable[[jnp.ndarray], float] = None,
        mlp_xi: Callable[[jnp.ndarray], float] = None,
        tau_a: float = 1.0,
        tau_b: float = 1.0,
        callback: Optional[Callable[[jnp.ndarray, jnp.ndarray, jnp.ndarray], Any]] = None,
        callback_kwargs: Dict[str, Any] = {},
        callback_iters: int = 10,
        seed: int = 0,
        **kwargs: Any,
    ) -> None:
        """
        The GENOT training class.

        Parameters
        ----------
        neural_net
            Neural vector field
        input_dim
            Dimension of the source distribution
        output_dim
            Dimension of the target distribution
        iterations
```

```
        Number of iterations to train
    ot_solver
        Solver to match samples from the source to the target distribution
    optimizer
        Optimizer for the neural vector field
    k_noise_per_x
        Number of samples to draw from the conditional distribution
    t_offset
        Offset for sampling from the time t
    epsilon
        Entropy regularization parameter for the discrete solver
    cost_fn
        Cost function to use for the discrete OT solver
    solver_latent_to_data
        Linear OT solver to match samples from the noise to the conditional distribution
    latent_to_data_epsilon
        Entropy regularization term for `solver_latent_to_data`
    latent_to_data_scale_cost
        How to scale the cost matrix for the `solver_latent_to_data` solver
    scale_cost
        How to scale the cost matrix in each discrete OT problem
    graph_kwargs
        Keyword arguments for the graph cost computation in case `cost="graph"`
    fused_penalty
        Penalisation term for the linear term in a Fused GW setting
    split_dim
        Dimension to split the data into fused term and purely quadratic term in the FGW setting
    mlp_eta
        Neural network to learn the left rescaling function
    mlp_xi
        Neural network to learn the right rescaling function
    tau_a
        Left unbalancedness parameter
    tau_b
        Right unbalancedness parameter
    callback
        Callback function
    callback_kwargs
        Keyword arguments to the callback function
    callback_iters
        Number of iterations after which to evaluate callback function
    seed
        Random seed
    kwargs
        Keyword arguments passed to `setup`, e.g. custom choice of optimizers for learning rescaling functions
"""
if isinstance(ot_solver, gromov_wasserstein.GromovWasserstein) and epsilon is not None:
    raise ValueError(
        "If `ot_solver` is `GromovWasserstein`, `epsilon` must be `None`. This check is performed "
        "to ensure that in the (fused) Gromov case the `epsilon` parameter is passed via the `ot_solver`."
    )

# setup parameters
self.rng = jax.random.PRNGKey(seed)
self.seed = seed
self.iterations = iterations
self.metrics = {"loss": [], "loss_eta": [], "loss_xi": []}

# neural parameters
self.neural_net = neural_net
self.state_neural_net: Optional[TrainState] = None
self.optimizer = optax.adamw(learning_rate=1e-4, weight_decay=1e-10) if optimizer is None else optimizer
self.noise_fn = self.noise_fn = jax.tree_util.Partial(
    jax.random.multivariate_normal, mean=jnp.zeros((output_dim,)), cov=jnp.diag(jnp.ones((output_dim,)))
)
self.input_dim = input_dim
self.output_dim = output_dim
self.k_noise_per_x = k_noise_per_x
self.t_offset = t_offset

# OT data-data matching parameters
self.ot_solver = ot_solver
self.epsilon = epsilon
self.cost_fn = cost_fn
self.scale_cost = scale_cost
self.graph_kwargs = graph_kwargs  # "k_neighbors", kwargs for graph.Graph.from_graph()
if fused_penalty != 0 and split_dim == 0:
    raise ValueError("Missing 'split_dim' for FGW.")
self.fused_penalty = fused_penalty
self.split_dim = split_dim

# OT latent-data matching parameters
self.solver_latent_to_data = solver_latent_to_data
self.latent_to_data_epsilon = latent_to_data_epsilon
self.latent_to_data_scale_cost = latent_to_data_scale_cost

# unbalancedness parameters
self.mlp_eta = mlp_eta
self.mlp_xi = mlp_xi
self.state_eta: Optional[TrainState] = None
self.state_xi: Optional[TrainState] = None
self.tau_a: float = tau_a
```

```python
        self.tau_b: float = tau_b

        # callback parameteres
        self.callback = callback
        self.callback_kwargs = callback_kwargs
        self.callback_iters = callback_iters

        self.setup(**kwargs)

    def setup(self, **kwargs: Any) -> None:
        """
        Set up the model.

        Parameters
        ----------
        kwargs
            Keyword arguments for the setup function
        """
        self.state_neural_net = self.neural_net.create_train_state(self.rng, self.optimizer, self.input_dim)
        self.step_fn = self._get_step_fn()
        if self.solver_latent_to_data is not None:
            self.match_latent_to_data_fn = self._get_match_latent_fn(
                self.solver_latent_to_data, self.latent_to_data_epsilon, self.latent_to_data_scale_cost
            )
        else:
            self.match_latent_to_data_fn = lambda key, x, y, **_: (x, y)

        if isinstance(self.ot_solver, sinkhorn.Sinkhorn):
            problem_type = "linear"
        else:
            problem_type = "fused" if self.fused_penalty > 0 else "quadratic"

        if self.cost_fn == "graph":
            self.match_fn = self._get_match_fn_graph(
                problem_type=problem_type,
                ot_solver=self.ot_solver,
                epsilon=self.epsilon,
                tau_a=self.tau_a,
                tau_b=self.tau_b,
                fused_penalty=self.fused_penalty,
                split_dim=self.split_dim,
                k_samples_per_x=self.k_noise_per_x,
                scale_cost=self.scale_cost,
                **self.graph_kwargs,
            )
        else:
            if problem_type == "linear":
                self.match_fn = self._get_sinkhorn_match_fn(
                    self.ot_solver,
                    epsilon=self.epsilon,
                    cost_fn=self.cost_fn,
                    tau_a=self.tau_a,
                    tau_b=self.tau_b,
                    scale_cost=self.scale_cost,
                    k_samples_per_x=self.k_noise_per_x,
                )
            else:
                self.match_fn = self._get_gromov_match_fn(
                    self.ot_solver,
                    cost_fn=self.cost_fn,
                    tau_a=self.tau_a,
                    tau_b=self.tau_b,
                    scale_cost=self.scale_cost,
                    split_dim=self.split_dim,
                    fused_penalty=self.fused_penalty,
                    k_samples_per_x=self.k_noise_per_x,
                )

        if self.mlp_eta is not None:
            opt_eta = kwargs["opt_eta"] if "opt_eta" in kwargs else optax.adamw(learning_rate=1e-4, weight_decay=1e-10)
            self.state_eta = self.mlp_eta.create_train_state(self.rng, opt_eta, self.input_dim)
        if self.mlp_xi is not None:
            opt_xi = kwargs["opt_xi"] if "opt_xi" in kwargs else optax.adamw(learning_rate=1e-4, weight_decay=1e-10)
            self.state_xi = self.mlp_xi.create_train_state(self.rng, opt_xi, self.output_dim)

    def __call__(
        self,
        x: Union[jnp.array, collections.abc.Iterable],
        y: Union[jnp.array, collections.abc.Iterable],
        batch_size_source: Optional[int] = None,
        batch_size_target: Optional[int] = None,
    ) -> None:
        """
        Train GENOT.

        Parameters
        ----------
        x
            Source data as an iterator or data array
        y
            Target data as an iterator or data array
        batch_size_source
```

```python
        Batch size for the source distribution.
    batch_size_target
        Batch size for the target distribution.
    """
    # prepare data
    if not hasattr(x, "shape"):
        x_loader = x
        x_load_fn = lambda x: x
    else:
        assert batch_size_source is not None, "'batch_size_source' must be specified when 'x' is an array."
        x_loader = iter(
            tf.data.Dataset.from_tensor_slices(x)
            .repeat()
            .shuffle(buffer_size=10_000, seed=self.seed)
            .batch(batch_size_source)
        )
        x_load_fn = tfds.as_numpy
    if not hasattr(y, "shape"):
        y_loader = y
        y_load_fn = lambda x: x
    else:
        assert batch_size_target is not None, "'batch_size_target' must be specified when 'y' is an array."
        y_loader = iter(
            tf.data.Dataset.from_tensor_slices(y)
            .repeat()
            .shuffle(buffer_size=10_000, seed=self.seed)
            .batch(batch_size_target)
        )
        y_load_fn = tfds.as_numpy

    batch: Dict[str, jnp.array] = {}
    for step in tqdm(range(self.iterations)):
        source_batch, target_batch = x_load_fn(next(x_loader)), y_load_fn(next(y_loader))
        self.rng, rng_time, rng_noise, rng_step_fn = jax.random.split(self.rng, 4)
        batch["source"] = source_batch
        batch["target"] = target_batch
        n_samples = len(source_batch) * self.k_noise_per_x
        t = (jax.random.uniform(rng_time, (1,)) + jnp.arange(n_samples) / n_samples) % (1 - self.t_offset)
        batch["time"] = t[:, None]
        batch["noise"] = self.noise_fn(rng_noise, shape=(len(source_batch), self.k_noise_per_x))
        (
            metrics,
            self.state_neural_net,
            self.state_eta,
            self.state_xi,
            eta_predictions,
            xi_predictions,
        ) = self.step_fn(
            rng_step_fn,
            self.state_neural_net,
            batch,
            self.state_eta,
            self.state_xi,
        )
        for key, value in metrics.items():
            self.metrics[key].append(value)

        if self.callback is not None and step % self.callback_iters == 0:
            self.callback(
                source=batch["source"],
                target=batch["target"],
                eta_predictions=eta_predictions,
                xi_predictions=xi_predictions,
                **self.callback_kwargs,
            )

def _get_sinkhorn_match_fn(
    self,
    ot_solver: Any,
    epsilon: float,
    cost_fn: str,
    tau_a: float,
    tau_b: float,
    scale_cost: Any,
    k_samples_per_x: int,
) -> Callable:
    @partial(
        jax.jit,
        static_argnames=["ot_solver", "epsilon", "cost_fn", "scale_cost", "tau_a", "tau_b", "k_samples_per_x"],
    )
    def match_pairs(
        key: jax.random.PRNGKeyArray,
        x: jnp.ndarray,
        y: jnp.ndarray,
        ot_solver: Any,
        tau_a: float,
        tau_b: float,
        epsilon: float,
        cost_fn: str,
        scale_cost: Any,
        k_samples_per_x: int,
    ) -> Tuple[jnp.ndarray, jnp.ndarray, jnp.ndarray, jnp.ndarray]:
```

```python
            geom = pointcloud.PointCloud(x, y, epsilon=epsilon, scale_cost=scale_cost, cost_fn=cost_fn)
            out = ot_solver(linear_problem.LinearProblem(geom, tau_a=tau_a, tau_b=tau_b))
            a, b = out.matrix.sum(axis=1), out.matrix.sum(axis=0)
            inds_source, inds_target = sample_conditional_indices_from_tmap(
                key=key,
                tmat=out.matrix,
                k_samples_per_x=k_samples_per_x,
                left_marginals=a,
                is_balanced=(tau_a == 1.0) and (tau_b == 1.0),
            )
            return x[inds_source], y[inds_target], a, b

        return jax.tree_util.Partial(
            match_pairs,
            ot_solver=ot_solver,
            epsilon=epsilon,
            cost_fn=cost_fn,
            tau_a=tau_a,
            tau_b=tau_b,
            scale_cost=scale_cost,
            k_samples_per_x=k_samples_per_x,
        )

    def _get_gromov_match_fn(
        self,
        ot_solver: Any,
        cost_fn: str,
        tau_a: float,
        tau_b: float,
        scale_cost: Any,
        split_dim: int,
        fused_penalty: float,
        k_samples_per_x: int,
    ) -> Callable:
        @partial(
            jax.jit,
            static_argnames=[
                "ot_solver",
                "cost_fn",
                "scale_cost",
                "fused_penalty",
                "split_dim",
                "tau_a",
                "tau_b",
                "k_samples_per_x",
            ],
        )
        def match_pairs(
            key: jax.random.PRNGKeyArray,
            x: Tuple[jnp.ndarray, jnp.ndarray],
            y: Tuple[jnp.ndarray, jnp.ndarray],
            ot_solver: Any,
            tau_a: float,
            tau_b: float,
            cost_fn: str,
            scale_cost,
            fused_penalty: float,
            k_samples_per_x: int,
            split_dim: int = 0,
        ) -> Tuple[jnp.array, jnp.array]:
            geom_xx = pointcloud.PointCloud(
                x=x[..., split_dim:], y=x[..., split_dim:], cost_fn=cost_fn, scale_cost=scale_cost
            )
            geom_yy = pointcloud.PointCloud(
                x=y[..., split_dim:], y=y[..., split_dim:], cost_fn=cost_fn, scale_cost=scale_cost
            )
            if split_dim > 0:
                geom_xy = pointcloud.PointCloud(
                    x=x[..., :split_dim], y=y[..., :split_dim], cost_fn=cost_fn, scale_cost=scale_cost
                )
            else:
                geom_xy = None
            prob = quadratic_problem.QuadraticProblem(
                geom_xx, geom_yy, geom_xy, fused_penalty=fused_penalty, tau_a=tau_a, tau_b=tau_b
            )
            out = ot_solver(prob)
            a, b = out.matrix.sum(axis=1), out.matrix.sum(axis=0)
            inds_source, inds_target = sample_conditional_indices_from_tmap(
                key=key,
                tmat=out.matrix,
                k_samples_per_x=k_samples_per_x,
                left_marginals=a,
                is_balanced=(tau_a == 1.0) and (tau_b == 1.0),
            )
            return x[inds_source], y[inds_target], a, b

        return jax.tree_util.Partial(
            match_pairs,
            ot_solver=ot_solver,
            cost_fn=cost_fn,
            tau_a=tau_a,
            tau_b=tau_b,
```

```python
            scale_cost=scale_cost,
            split_dim=split_dim,
            fused_penalty=fused_penalty,
            k_samples_per_x=k_samples_per_x,
        )

    def _get_match_fn_graph(
        self,
        problem_type: Literal["linear", "quadratic", "fused"],
        ot_solver: Any,
        epsilon: float,
        k_neighbors: int,
        tau_a: float,
        tau_b: float,
        scale_cost: Any,
        fused_penalty: float,
        split_dim: int,
        k_samples_per_x: int,
        **kwargs,
    ) -> Callable:
        def get_nearest_neighbors(
            X: jnp.ndarray, Y: jnp.ndarray, k: int = 30  # type: ignore[name-defined]
        ) -> Tuple[jnp.ndarray, jnp.ndarray]:  # type: ignore[name-defined]
            concat = jnp.concatenate((X, Y), axis=0)
            pairwise_euclidean_distances = pointcloud.PointCloud(concat, concat).cost_matrix
            distances, indices = jax.lax.approx_min_k(
                pairwise_euclidean_distances, k=k, recall_target=0.95, aggregate_to_topk=True
            )
            return distances, indices

        def create_cost_matrix(X: jnp.array, Y: jnp.array, k_neighbors: int, **kwargs: Any) -> jnp.array:
            distances, indices = get_nearest_neighbors(X, Y, k_neighbors)
            a = jnp.zeros((len(X) + len(Y), len(X) + len(Y)))
            adj_matrix = a.at[
                jnp.repeat(jnp.arange(len(X) + len(Y)), repeats=k_neighbors).flatten(), indices.flatten()
            ].set(distances.flatten())
            return graph.Graph.from_graph(adj_matrix, normalize=kwargs.pop("normalize", True), **kwargs).cost_matrix[
                : len(X), len(X) :
            ]

        @partial(
            jax.jit,
            static_argnames=[
                "ot_solver",
                "problem_type",
                "epsilon",
                "k_neighbors",
                "tau_a",
                "tau_b",
                "k_samples_per_x",
                "fused_penalty",
                "split_dim",
            ],
        )
        def match_pairs(
            key: jax.random.PRNGKeyArray,
            x: jnp.ndarray,
            y: jnp.ndarray,
            ot_solver: Any,
            problem_type: Literal["linear", "quadratic", "fused"],
            epsilon: float,
            tau_a: float,
            tau_b: float,
            fused_penalty: float,
            split_dim: int,
            k_neighbors: int,
            k_samples_per_x: int,
            **kwargs,
        ) -> Tuple[jnp.array, jnp.array, jnp.ndarray, jnp.ndarray]:
            if problem_type == "linear":
                cm = create_cost_matrix(x, y, k_neighbors, **kwargs)
                geom = geometry.Geometry(cost_matrix=cm, epsilon=epsilon, scale_cost=scale_cost)
                out = ot_solver(linear_problem.LinearProblem(geom, tau_a=tau_a, tau_b=tau_b))
            else:
                cm_xx = create_cost_matrix(x[..., split_dim:], x[..., split_dim:], k_neighbors, **kwargs)
                cm_yy = create_cost_matrix(y[..., split_dim:], y[..., split_dim:], k_neighbors, **kwargs)
                geom_xx = geometry.Geometry(cost_matrix=cm_xx, epsilon=epsilon, scale_cost=scale_cost)
                geom_yy = geometry.Geometry(cost_matrix=cm_yy, epsilon=epsilon, scale_cost=scale_cost)
                if problem_type == "quadratic":
                    geom_xy = None
                else:
                    cm_xy = create_cost_matrix(x[..., :split_dim], y[..., :split_dim], k_neighbors, **kwargs)
                    geom_xy = geometry.Geometry(cost_matrix=cm_xy, epsilon=epsilon, scale_cost=scale_cost)
                prob = quadratic_problem.QuadraticProblem(
                    geom_xx, geom_yy, geom_xy, fused_penalty=fused_penalty, tau_a=tau_a, tau_b=tau_b
                )
                out = ot_solver(prob)
            a, b = out.matrix.sum(axis=0), out.matrix.sum(axis=1)
            inds_source, inds_target = sample_conditional_indices_from_tmap(
                key=key,
                tmat=out.matrix,
                k_samples_per_x=k_samples_per_x,
```

```python
                left_marginals=a,
                is_balanced=(tau_a == 1.0) and (tau_b == 1.0),
            )
            return x[inds_source], y[inds_target], a, b

        return jax.tree_util.Partial(
            match_pairs,
            problem_type=problem_type,
            ot_solver=ot_solver,
            epsilon=epsilon,
            k_neighbors=k_neighbors,
            tau_a=tau_a,
            tau_b=tau_b,
            k_samples_per_x=k_samples_per_x,
            fused_penalty=fused_penalty,
            split_dim=split_dim,
            **kwargs,
        )

    def _get_match_latent_fn(self, ot_solver: Type[sinkhorn.Sinkhorn], epsilon: float, scale_cost: Any) -> Callable:
        @partial(jax.jit, static_argnames=["ot_solver", "epsilon", "scale_cost"])
        def match_latent_to_data(
            key: jax.random.PRNGKeyArray,
            ot_solver: Type[was_solver.WassersteinSolver],
            x: jnp.ndarray,
            y: jnp.ndarray,
            epsilon: float,
            scale_cost: Any,
        ) -> Tuple[jnp.ndarray, jnp.ndarray]:
            geom = pointcloud.PointCloud(x, y, epsilon=epsilon, scale_cost=scale_cost)
            out = ot_solver(linear_problem.LinearProblem(geom))
            inds_source, inds_target = sample_conditional_indices_from_tmap(key, out.matrix, 1, None, is_balanced=True)
            return x[inds_source], y[inds_target]

        return jax.tree_util.Partial(match_latent_to_data, ot_solver=ot_solver, epsilon=epsilon, scale_cost=scale_cost)

    def _get_step_fn(self) -> Callable:
        def loss_fn(
            params_mlp: jnp.array,
            apply_fn_mlp: Callable,
            batch: Dict[str, jnp.array],
        ):
            def phi_t(x_0: jnp.ndarray, x_1: jnp.ndarray, t: jnp.ndarray) -> jnp.ndarray:
                return (1 - t) * x_0 + t * x_1

            def u_t(x_0: jnp.ndarray, x_1: jnp.ndarray) -> jnp.ndarray:
                return x_1 - x_0

            phi_t_eval = phi_t(batch["noise"], batch["target"], batch["time"])
            mlp_pred = apply_fn_mlp(
                {"params": params_mlp}, t=batch["time"], latent=phi_t_eval, condition=batch["source"]
            )
            d_psi = u_t(batch["noise"], batch["target"])

            return jnp.mean(optax.l2_loss(mlp_pred, d_psi))

        def loss_a_fn(
            params_eta: Optional[jnp.ndarray],
            apply_fn_eta: Optional[Callable],
            x: jnp.ndarray,
            a: jnp.ndarray,
            expectation_reweighting: float,
        ) -> float:
            eta_predictions = apply_fn_eta({"params": params_eta}, x)
            return (
                optax.l2_loss(eta_predictions[:, 0], a).mean()
                + optax.l2_loss(jnp.mean(eta_predictions) - expectation_reweighting),
                eta_predictions,
            )

        def loss_b_fn(
            params_xi: Optional[jnp.ndarray],
            apply_fn_xi: Optional[Callable],
            x: jnp.ndarray,
            b: jnp.ndarray,
            expectation_reweighting: float,
        ) -> float:
            xi_predictions = apply_fn_xi({"params": params_xi}, x)
            return (
                optax.l2_loss(xi_predictions, b).mean()
                + optax.l2_loss(jnp.mean(xi_predictions) - expectation_reweighting),
                xi_predictions,
            )

        @jax.jit
        def step_fn(
            key: jax.random.PRNGKeyArray,
            state_neural_net: TrainState,
            batch: Dict[str, jnp.array],
            state_eta: Optional[TrainState] = None,
            state_xi: Optional[TrainState] = None,
        ):
```

```python
            rng_match, rng_noise = jax.random.split(key, 2)
            original_source_batch = batch["source"]
            original_target_batch = batch["target"]
            source_batch, target_batch, a, b = self.match_fn(rng_match, batch["source"], batch["target"])
            rng_noise = jax.random.split(rng_noise, (len(target_batch)))

            noise_matched, conditional_target = jax.vmap(self.match_latent_to_data_fn, 0, 0)(
                key=rng_noise, x=batch["noise"], y=target_batch
            )

            batch["source"] = jnp.reshape(source_batch, (len(source_batch), -1))
            batch["target"] = jnp.reshape(conditional_target, (len(source_batch), -1))
            batch["noise"] = jnp.reshape(noise_matched, (len(source_batch), -1))

            grad_fn = jax.value_and_grad(loss_fn, has_aux=False)
            loss, grads_mlp = grad_fn(
                state_neural_net.params,
                state_neural_net.apply_fn,
                batch,
            )
            metrics = {}
            metrics["loss"] = loss

            integration_eta = jnp.sum(a)
            integration_xi = jnp.sum(b)

            if state_eta is not None:
                grad_a_fn = jax.value_and_grad(loss_a_fn, argnums=0, has_aux=True)
                (loss_a, eta_predictions), grads_eta = grad_a_fn(
                    state_eta.params,
                    state_eta.apply_fn,
                    original_source_batch[:,],
                    a * len(original_source_batch),
                    integration_xi,
                )

                new_state_eta = state_eta.apply_gradients(grads=grads_eta)
                metrics["loss_eta"] = loss_a

            else:
                new_state_eta = eta_predictions = None
            if state_xi is not None:
                grad_b_fn = jax.value_and_grad(loss_b_fn, argnums=0, has_aux=True)
                (loss_b, xi_predictions), grads_xi = grad_b_fn(
                    state_xi.params,
                    state_xi.apply_fn,
                    original_target_batch[:,],
                    (b * len(original_target_batch))[:, None],
                    integration_eta,
                )
                new_state_xi = state_xi.apply_gradients(grads=grads_xi)
                metrics["loss_xi"] = loss_b
            else:
                new_state_xi = xi_predictions = None

            return (
                metrics,
                state_neural_net.apply_gradients(grads=grads_mlp),
                new_state_eta,
                new_state_xi,
                eta_predictions,
                xi_predictions,
            )

        return step_fn

    def transport(
        self, source: jnp.array, seed: int = 0, diffeqsolve_kwargs: Dict[str, Any] = types.MappingProxyType({})
    ) -> Union[jnp.array, diffrax.Solution, Optional[jnp.ndarray]]:
        """
        Transport the distribution.

        Parameters
        ----------
        source
            Source distribution to transport
        seed
            Random seed for sampling from the latent distribution
        diffeqsolve_kwargs
            Keyword arguments for the ODE solver.

        Returns
        -------
            The transported samples, the solution of the neural ODE, and the rescaling factor.
        """
        diffeqsolve_kwargs = dict(diffeqsolve_kwargs)
        rng = jax.random.PRNGKey(seed)
        latent_shape = (len(source),)
        latent_batch = self.noise_fn(rng, shape=latent_shape)
        apply_fn_partial = partial(self.state_neural_net.apply_fn, condition=source)
        solution = diffrax.diffeqsolve(
            diffrax.ODETerm(
```

```python
            lambda t, y, *args: apply_fn_partial({"params": self.state_neural_net.params}, t=t, latent=y)
            ),
            diffeqsolve_kwargs.pop("solver", diffrax.Tsit5()),
            t0=0,
            t1=1,
            dt0=diffeqsolve_kwargs.pop("dt0", None),
            y0=latent_batch,
            stepsize_controller=diffeqsolve_kwargs.pop(
                "stepsize_controller", diffrax.PIDController(rtol=1e-3, atol=1e-6)
            ),
            **diffeqsolve_kwargs,
        )
        if self.state_eta is not None:
            weight_factors = self.state_eta.apply_fn({"params": self.state_eta.params}, x=source)
        else:
            weight_factors = jnp.ones(source.shape)
        return solution.ys, solution, weight_factors
```

**The network architectures** The following code contains the neural network architectures.

```python
from typing import Callable, Optional

import flax.linen as nn
import optax
from flax.training import train_state

import jax
import jax.numpy as jnp
from ott.solvers.nn.models import ModelBase, NeuralTrainState

class Block(nn.Module):
    """
    Block of a neural network.

    Parameters
    ----------
    dim
        Input dimension.
    out_dim
        Output dimension.
    num_layers
        Number of layers.
    act_fn
        Activation function.
    """

    dim: int = 128
    out_dim: int = 32
    num_layers: int = 3
    act_fn: Callable[[jnp.ndarray], jnp.ndarray] = nn.silu

    @nn.compact
    def __call__(self, x):
        for i in range(self.num_layers):
            x = nn.Dense(self.dim, name=f"fc{i}")(x)
            x = self.act_fn(x)
        return nn.Dense(self.out_dim, name="fc_final")(x)

class MLP_vector_field(ModelBase):
    """
    Neural vector field.

    Parameters
    ----------
    output_dim
        Output dimension.
    latent_embed_dim
        Latent embedding dimension.
    condition_embed_dim
        Condition embedding dimension.
    t_embed_dim
        Time embedding dimension.
    joint_hidden_dim
        Joint hidden dimension.
    num_layers
        Number of layers per block.
    act_fn
        Activation function.
    n_frequencies
        Number of frequencies for time embedding.
    """

    output_dim: int
    latent_embed_dim: int
    condition_embed_dim: Optional[int] = None
    t_embed_dim: Optional[int] = None
    joint_hidden_dim: Optional[int] = None
    num_layers: int = 3
    act_fn: Callable[[jnp.ndarray], jnp.ndarray] = nn.silu
```

```python
    n_frequencies: int = 1

    def time_encoder(self, t: jnp.array) -> jnp.array:
        freq = 2 * jnp.arange(self.n_frequencies) * jnp.pi
        t = freq * t
        return jnp.concatenate((jnp.cos(t), jnp.sin(t)), axis=-1)

    def __post_init__(self):
        if self.condition_embed_dim is None:
            self.condition_embed_dim = self.latent_embed_dim
        if self.t_embed_dim is None:
            self.t_embed_dim = self.latent_embed_dim

        concat_embed_dim = self.latent_embed_dim + self.condition_embed_dim + self.t_embed_dim
        if self.joint_hidden_dim is not None:
            assert self.joint_hidden_dim >= concat_embed_dim, (
                "joint_hidden_dim must be greater than or equal to the sum of " "all embedded dimensions. "
            )
            self.joint_hidden_dim = self.latent_embed_dim
        else:
            self.joint_hidden_dim = concat_embed_dim
        super().__post_init__()

    @property
    def is_potential(self) -> bool:
        return self.output_dim == 1

    @nn.compact
    def __call__(self, t: float, condition: jnp.ndarray, latent: jnp.ndarray) -> jnp.ndarray:
        condition, latent = jnp.atleast_2d(condition, latent)

        t = jnp.full(shape=(len(condition), 1), fill_value=t)
        t = self.time_encoder(t)
        t = Block(
            dim=self.t_embed_dim,
            out_dim=self.t_embed_dim,
            num_layers=self.num_layers,
            act_fn=self.act_fn,
        )(t)

        condition = Block(
            dim=self.condition_embed_dim,
            out_dim=self.condition_embed_dim,
            num_layers=self.num_layers,
            act_fn=self.act_fn,
        )(condition)

        latent = Block(
            dim=self.latent_embed_dim, out_dim=self.latent_embed_dim, num_layers=self.num_layers, act_fn=self.act_fn
        )(latent)

        concat_embed = jnp.concatenate((t, condition, latent), axis=-1)
        out = Block(
            dim=self.joint_hidden_dim,
            out_dim=self.joint_hidden_dim,
            num_layers=self.num_layers,
            act_fn=self.act_fn,
        )(concat_embed)

        return nn.Dense(self.output_dim, use_bias=True, name="final_layer")(out)

    def create_train_state(
        self,
        rng: jax.random.PRNGKeyArray,
        optimizer: optax.OptState,
        input_dim: int,
    ) -> NeuralTrainState:
        params = self.init(rng, jnp.ones((1, 1)), jnp.ones((1, input_dim)), jnp.ones((1, self.output_dim)))["params"]
        return train_state.TrainState.create(apply_fn=self.apply, params=params, tx=optimizer)

class MLP_marginal(ModelBase):
    """
    Neural network parameterizing a reweighting function.

    Parameters
    ----------
    hidden_dim
        Hidden dimension.
    num_layers
        Number of layers.
    act_fn
        Activation function.
    """

    hidden_dim: int
    num_layers: int = 3
    act_fn: Callable[[jnp.ndarray], jnp.ndarray] = nn.silu

    @property
    def is_potential(self) -> bool:
        return True
```

```python
    @nn.compact
    def __call__(self, x: jnp.ndarray) -> jnp.ndarray:
        z = x
        z = Block(dim=self.hidden_dim, out_dim=1, num_layers=self.num_layers, act_fn=self.act_fn)(z)
        return nn.softplus(z)
```

**Example workflow for recovering lineage branching events in the pancreas dataset** The following code reproduces the computations needed for figure 2.

```python
# import GENOT
from genot.models.model import GENOT
from genot.nets.nets import MLP_vector_field

from typing import Iterable, Tuple
from joblib import Parallel, delayed
import jax
import jax.numpy as jnp
import numpy as np
import pandas as pd
from ott.solvers.linear import sinkhorn
import scanpy as sc

# load data into current directory. The data can be downloaded from https://www.ncbi.nlm.nih.gov/geo/query/acc.cgi?acc=GSE132188
adata = sc.read("../../data/adata_pancreas_2019_endocrine.h5ad")
sc.pp.pca(adata, n_comps=30)

source_train = adata.obsm["X_pca"]
target_train = adata.obsm["X_pca"]

# Train the GENOT-K model
ot_solver = sinkhorn.Sinkhorn()
neural_net = MLP_vector_field(target_train.shape[1], latent_embed_dim=256, num_layers=8, n_frequencies=128)

genot_k = GENOT(
    neural_net,
    ot_solver=ot_solver,
    epsilon=1e-2,
    scale_cost="mean",
    input_dim=30,
    output_dim=30,
    iterations=10_000,
    k_noise_per_x=1,
)
genot_k(source_train, target_train, 1024, 1024)

# Sample from the conditional distribution
push_source = [None] * 30

for i in range(30):
    push_source[i] = genot_k.transport(source_train, seed=i)[0][0, ...]

# The following code snippets are needed for computing the uncertainty statistics and are taken from
# https://github.com/YosefLab/velovi/blob/a8090d63396e2695bc8695f2fc69c34b39d62dc4/velovi/_model.py

def sample_velocities_from_transport_matrix(
    transport_matrix: np.ndarray, target_distribution: np.ndarray, n_samples: int
) -> np.ndarray:
    res = [None] * n_samples
    for i in range(n_samples):
        res[i] = target_distribution[
            jax.vmap(lambda x: jax.random.categorical(jax.random.PRNGKey(i), x))(jnp.log(transport_matrix))
        ]
    return jnp.asarray(res)

def compute_statistics_from_samples(
    samples: np.ndarray, split_to_k_batches: int, cell_names: Iterable[str]
) -> pd.DataFrame:
    batch_size = samples.shape[0] // split_to_k_batches
    assert samples.shape[0] % split_to_k_batches == 0
    # samples is of shape (n_samples, n_cells, dim_velocity_vector)
    df = pd.DataFrame(index=cell_names)

    for i in range(split_to_k_batches):
        df[f"var_{i}"] = np.sum(np.var(samples[i * batch_size : (i + 1) * batch_size, ...], axis=0), axis=1)
        var_cols = [f"var_{i}" for i in range(split_to_k_batches)]

    df["var_of_variance"] = df[var_cols].var(axis=1)
    df["mean_of_variance"] = df[var_cols].mean(axis=1)
    return df

def compute_directional_statistics_from_samples(
    samples: np.ndarray, n_jobs: int, cell_names: Iterable[str]
) -> pd.DataFrame:
    samples = np.asarray(samples)
    n_cells = len(cell_names)
    df = pd.DataFrame(index=cell_names)
```

```python
        df["directional_variance"] = np.nan
        df["directional_difference"] = np.nan
        df["directional_cosine_sim_variance"] = np.nan
        df["directional_cosine_sim_difference"] = np.nan
        df["directional_cosine_sim_mean"] = np.nan
        results = Parallel(n_jobs=n_jobs, verbose=3)(
            delayed(_directional_statistics_per_cell)(samples[:, cell_index, :]) for cell_index in range(n_cells)
        )
        # cells by samples
        cosine_sims = np.stack([results[i][0] for i in range(n_cells)])
        df.loc[:, "directional_cosine_sim_variance"] = [results[i][1] for i in range(n_cells)]
        df.loc[:, "directional_cosine_sim_difference"] = [results[i][2] for i in range(n_cells)]
        df.loc[:, "directional_variance"] = [results[i][3] for i in range(n_cells)]
        df.loc[:, "directional_difference"] = [results[i][4] for i in range(n_cells)]
        df.loc[:, "directional_cosine_sim_mean"] = [results[i][5] for i in range(n_cells)]

        return df, cosine_sims

def _cosine_sim(v1: np.ndarray, v2: np.ndarray) -> np.ndarray:
    """Returns cosine similarity of the vectors."""
    v1_u = _centered_unit_vector(v1)
    v2_u = _centered_unit_vector(v2)
    return np.clip(np.dot(v1_u, v2_u), -1.0, 1.0)

def _directional_statistics_per_cell(
    tensor: np.ndarray,
) -> Tuple[np.ndarray, np.ndarray, np.ndarray, np.ndarray, np.ndarray]:
    """Internal function for parallelization.

    Parameters
    ----------
    tensor
        Shape of samples by genes for a given cell.
    """
    n_samples = tensor.shape[0]
    # over samples axis
    mean_velocity_of_cell = tensor.mean(0)
    cosine_sims = [_cosine_sim(tensor[i, :], mean_velocity_of_cell) for i in range(n_samples)]
    angle_samples = [np.arccos(el) for el in cosine_sims]
    return (
        cosine_sims,
        np.var(cosine_sims),
        np.percentile(cosine_sims, 95) - np.percentile(cosine_sims, 5),
        np.var(angle_samples),
        np.percentile(angle_samples, 95) - np.percentile(angle_samples, 5),
        np.mean(cosine_sims),
    )

def _centered_unit_vector(vector: np.ndarray) -> np.ndarray:
    """Returns the centered unit vector of the vector."""
    vector = vector - np.mean(vector)
    return vector / np.linalg.norm(vector)

# In the following, we compute the statistics
cosine_vars_source = compute_directional_statistics_from_samples(
    np.array(push_source), 2, adata[adata.obs["day"] == "14.5"].obs_names
)
adata.obs["directional_cosine_sim_variance"] = cosine_vars_source[0]["directional_cosine_sim_variance"]
```

**Example workflow for modality translation in the bone marrow dataset**  The following code reproduces the computations needed for figure 5.

```python
# import GENOT
from genot.models.model import GENOT
from genot.nets.nets import MLP_vector_field

import jax
import jax.numpy as jnp
import numpy as np
import scipy
import sklearn.preprocessing as pp
from ott.geometry.pointcloud import PointCloud
from ott.solvers.linear import acceleration, sinkhorn
from ott.solvers.quadratic import gromov_wasserstein
from ott.tools.sinkhorn_divergence import sinkhorn_divergence
import scanpy as sc
from moscot import datasets

# Perform modality translation with GW and graph costs
adata_atac = datasets.bone_marrow(rna=False)
adata_rna = datasets.bone_marrow(rna=True)

# Preprocess data and split into train and test split
adata_source = adata_atac.copy()
adata_target = adata_rna.copy()
```

```python
n_cells_source = len(adata_atac)

n_samples_train = int(n_cells_source * 0.6)
n_samples_test = n_cells_source - n_cells_source

inds_train = np.asarray(jax.random.choice(jax.random.PRNGKey(0), n_cells_source, (n_samples_train,), replace=False))
inds_test = list(set(list(range(n_cells_source))) - set(np.asarray(inds_train)))

fused = np.concatenate((adata_atac.obsm["geneactivity_scvi"], adata_rna.obsm["geneactivity_scvi"]), axis=0)
fused = sc.pp.pca(fused, n_comps=25)

source_fused = fused[: len(adata_source), :]
target_fused = fused[len(adata_target) :, :]

source_q = pp.normalize(adata_source.obsm["ATAC_lsi_red"], norm="l2")
target_q = adata_target.obsm["GEX_X_pca"]

source_train_q = source_q[inds_train, :]
source_test_q = source_q[inds_test, :]
target_train_q = target_q[inds_train, :]
target_test_q = target_q[inds_test, :]
source_train_fused = source_fused[inds_train, :]
source_test_fused = source_fused[inds_test, :]
target_train_fused = target_fused[inds_train, :]
target_test_fused = target_fused[inds_test, :]

source_train = np.concatenate((source_train_fused, source_train_q), axis=1)
source_test = np.concatenate((source_test_fused, source_test_q), axis=1)
target_train = np.concatenate((target_train_fused, target_train_q), axis=1)
target_test = np.concatenate((target_test_fused, target_test_q), axis=1)

# Train the GENOT-FGW model
neural_net = MLP_vector_field(target_train.shape[1], latent_embed_dim=256, num_layers=8, n_frequencies=128)
linear_ot_solver = sinkhorn.Sinkhorn(momentum=acceleration.Momentum(value=1.0, start=25))
solver = gromov_wasserstein.GromovWasserstein(epsilon=0.01, linear_ot_solver=linear_ot_solver)

genot_fgw = GENOT(
    neural_net,
    epsilon=None,
    scale_cost="mean",
    input_dim=source_train.shape[1],
    output_dim=target_train.shape[1],
    iterations=5_000,
    ot_solver=solver,
    k_noise_per_x=1,
    fused_penalty=1.0,
    split_dim=fused.shape[1],
)
genot_fgw(source_train, target_train, 1024, 1024)

# Sample 30 times from the conditional distribution
res_test = [None] * 30

for i in range(30):
    res_test[i] = genot_fgw.transport(source_test, seed=i)[0][0, ...]

# Compute the conditional mean
cond_mean_test = jnp.mean(jnp.asarray(res_test), axis=0)

# define the FOSCTTM score (taken from SCOT, Demetci et al., 2022)
def foscttm(
    x: np.ndarray,
    y: np.ndarray,
) -> float:
    d = scipy.spatial.distance_matrix(x, y)
    foscttm_x = (d < np.expand_dims(np.diag(d), axis=1)).mean(axis=1)
    foscttm_y = (d < np.expand_dims(np.diag(d), axis=0)).mean(axis=0)
    fracs = []
    for i in range(len(foscttm_x)):
        fracs.append((foscttm_x[i] + foscttm_y[i]) / 2)
    return np.mean(fracs).round(4)

# Evaluate optimality with the FOSCTTM score
foscttm_one_sample = foscttm(res_test[0], target_test)
foscttm_cond_mean = foscttm(cond_mean_test, target_test)

# Evaluate fitting property with the sinkhorn divergence
sinkhorn_div_one_sample = sinkhorn_divergence(PointCloud, res_test[0], target_test, epsilon=1e-2).divergence
sinkhorn_div_cond_mean = sinkhorn_divergence(PointCloud, cond_mean_test, target_test, epsilon=1e-2).divergence
```

