# OpenReview forum: "Generative Entropic Neural Optimal Transport To Map Within and Across Space"
_ICLR.cc/2024/Conference — Submitted to ICLR 2024_

### Official Review · Reviewer_fozM · 2023-10-23

**Soundness:** 4 excellent
**Presentation:** 4 excellent
**Contribution:** 2 fair
**Rating:** 5
**Confidence:** 3

**Summary:**

This work proposes a general framework called GENOT that solves entropy regularized balanced/unbalanced OT and Gromov-Wasserstein extensions. The proposed framework is flexible with any ground cost, offers a way to sample from the optimal conditional plan, and can map across incomparable spaces. The key idea is to use solutions from a discrete OT solver as supervision to train conditional flow-matching models. Experiments on toy data and single-cell data are done to illustrate the proposed method's effectiveness compared to other methods.

**Strengths:**

* The presentation of the paper is really good. The writing is concise and to the point, without lacking in rigor.
* The proposed method is simple and effective, and the authors have adequately demonstrated its flexibility in various scenarios in the experiments section.
* The way the paper handles unbalanced OT by learning a reweighting function is novel and I haven't seen it before.
* Experiments seem solid and cover a good range of applications.

**Weaknesses:**

* The main weakness in my opinion is that the proposed framework is not actually solving OT; rather it relies on a discrete solver to compute approximate discrete transport plans and then fit a neural network (in this case conditional flow matching) to follow the the discrete solutions. This point is not obvious until I read to the end of Section 3.1 after describing the main method. I think the authors need to mention this reliance on a discrete solver in their abstract and introduction. Otherwise, I think the paper is overselling, since by reading the abstract and the introduction, I was very excited because I thought the paper has found a unified way to solve all EOT problems, without relying on a discrete solver.
* Since the method is essentially regressing discrete OT solutions from a black-box discrete OT solver, very little of the proposed method is related to optimal transport. For example, if we have any task of mapping one measure to another (not necessarily in an optimal way) and if we can solve a discretized version of the task, then we could use the exact formulation (7) without any changes.
* The proposed algorithm relies on the discrete OT solver to do the heavy lifting and hence must inherit the limitation of discretization. For instance, it might be difficult to apply to the image domain where using a small batch of images would not capture the whole distribution accurately. This is (to some degree) not a problem of other neural solvers like [De Bortoli 2021], [Korotin et al. 2022a;b]. This limitation should be discussed more.
* While in Proposition 3.1, the authors proved minimizing (7) results in the correct optimal plan, in practice in Algorithm 1,2, a **biased** loss is being optimized due to finite batch sizes. By biased, what I mean is that the expectation of the discrete estimator in Algorithm 1,2 is not necessarily the same as (7). One could expect the biasedness will decrease as the batch size increases. However, the effect of batch sizes is not discussed at all; it is taken to be 1024 throughout. I view this point as a central limitation of the current work. If theoretical analysis is difficult, empirical evidence to show how the results degrade as a function of batch size could be important. For instance, using a batch size of 1 would likely break the algorithm.

**Questions:**

1. In "Noise Outsourcing" paragraph in Section 3.1, I think the condition $x \sim \mu$ can be replaced with "for all x in the support of $\mu$" to improve the clarity of the sentence "More precisely ..."
2. Above (5), it says "instead of directly modeling $T_\theta(\cdot |x)$ as a neural network." What is the advantage of using flow matching versus a pushforward map?
3. $\hat \pi$ under (7) is undefined.
4. It would be great if the authors could comment on the biasedness issue I mentioned in the weakness section. Similarly, while Proposition 3.3 shows the consistency of the estimator of $\hat{\eta_n}$, it does not address the biasedness issue of this estimator.
5. Is there any reason that the authors have not applied the current framework to image domains? Is it because the bottleneck is due to finite batch sizes as mentioned above in the weakness section?

---

> ### Author Response · Authors · 2023-11-17
> **Many thanks for your feedback**
>
> > **_The presentation of the paper is really good. The writing is concise and to the point, without lacking in rigor._**
>
> ➤ Thanks! We have tried to improve a few things in the updated pdf.
>
> > **_“The main weakness in my opinion is that the proposed framework is not actually solving OT; rather it relies on a discrete solver to compute approximate discrete transport plans and then fit a neural network (in this case conditional flow matching) to follow the the discrete solutions.”_**
>
> ➤ We would like to introduce an important nuance to your statement: our aim is to replicate, with our architecture, _entropic_ OT, not (unregularized) OT.
>
> > **_“Since the method is essentially regressing discrete OT solutions from a black-box discrete OT solver, very little of the proposed method is related to optimal transport. For example, if we have any task of mapping one measure to another (not necessarily in an optimal way) and if we can solve a discretized version of the task, then we could use the exact formulation (7) without any changes.”_**
>
> ➤ Our goal is to learn _entropic, unbalanced_ OT; therefore, we use discrete _entropic_ solvers to regress, as you mention, a continuous model on these values. It seems natural that the type of continuous (conditional) coupling we learn is tied to the coupling we feed to the model. Because we do not know other approaches to estimate couplings (aside from the trivial independent coupling) than OT, we focus on OT solutions.
>
> We do agree with you, however, on the fact that another method would arise when _changing_ the OT solver. For instance, plugging a _low-rank_ OT solver (as in [Scetbon+21, Scetbon+22]) could result in a completely different "neural low-rank OT" model
>
> > **_“The proposed algorithm relies on the discrete OT solver to do the heavy lifting and hence must inherit the limitation of discretization. For instance, it might be difficult to apply to the image domain where using a small batch of images would not capture the whole distribution accurately. This is (to some degree) not a problem of other neural solvers like [De Bortoli 2021], [Korotin et al. 2022a;b]. This limitation should be discussed more.”_**
>
> ➤ We agree that “our” mini-batch limitation is not the _same_ problem encountered by the solvers you mention, which rely instead on perfect convergence of intermediate projection steps, or which can suffer from biases resulting from min-max saddled points.
>
> We agree that the mini-batch limitation is not discussed enough, and we have started to add a discussion around Proposition 3.1. This limitation is shared by all OT-guided approaches, including e.g. [Genevay+2018], [Salimans+2018] early work, as well as, more recently, [Uscidda+23,Pooladian+23,Tong+23 ab]’s approaches.
>
> We argue, however, that we naturally suffer _less_ from that limitation, because our “ground truth” is _precisely_ the _entropic_ coupling, which is known to have better recovery rates than “true OT”. The aforementioned works typically set their theoretical golden standard to be $\pi_0^\star$, and use $\pi_\varepsilon^\star$ for computational convenience. We target **directly** from $\pi_\varepsilon^\star$. This estimation suffers, intuitively, less from mini-batch bias, because of the machinery developed in the recent works of [Genevay+19, Mena+19, Rigollet+22].
>
> With this, we do not mean that the limitation does not exist, but we believe this is an ongoing debate in the community, and that we might, because our target is set on entropic couplings from the start, be less sensitive to it.
>
> > **_“While in Proposition 3.1, the authors proved minimizing (7) results in the correct optimal plan, in practice in Algorithm 1,2, a biased loss is being optimized due to finite batch sizes. By biased, what I mean is that the expectation of the discrete estimator in Algorithm 1,2 is not necessarily the same as (7).”_**
>
> ➤ We have clarified in the updated pdf that Proposition 3.1 is given as a way to theoretically justify our loss, not as a proper justification of our algorithm. We have substantially changed the language in that part, and introduced a discussion on why, while not being a panacea, we believe that our mini-batch approach is particularly adapted to our goal of estimating entropic couplings (and *not* the elusive “cursed-by-dimension-true-OT-map”).
>
> > **_“One could expect the biasedness will decrease as the batch size increases. However, the effect of batch sizes is not discussed at all; it is taken to be 1024 throughout. I view this point as a central limitation of the current work. If theoretical analysis is difficult, empirical evidence to show how the results degrade as a function of batch size could be important. For instance, using a batch size of 1 would likely break the algorithm”_**
>
> ➤  We added an analysis of the influence of the batch size in appendix E2. Conceptually it seems that a batch size of 1 would, in the balanced case, fall back on the independent sampling of flow matching.

---

> ### Author Response · Authors · 2023-11-17
> **Second part of answers**
>
> > ***“In "Noise Outsourcing" paragraph in Section 3.1, I think the condition can be replaced with "for all x in the support of " to improve the clarity of the sentence "More precisely ..." “***
>
> ➤ We have corrected the paper following your suggestion, in the precise area you highlighted, but elsewhere, many thanks for this remark.
>
> > **_“Above (5), it says "instead of directly modeling as a neural network." What is the advantage of using flow matching versus a pushforward map?”_**
>
> ➤ We make this choice following the observation that CFM methods are easier to train for complex distributions [Lipman+'2023], [Pooladian+'2023], [Tong+'2023;a;b] than direct maps, because they parameterize the map from one distribution to another implicitly as the solution of an ODE. Learning a map through this **progressive refinement** (as opposed to a _static_, single step jump for OT map parameterization) is also one of the origins of the success of diffusion models (and can be traced back to past developments, e.g. in the sampling literature, with SMC vs. MCMC)
>
> > **_"It would be great if the authors could comment on the biasedness issue I mentioned in the weakness section. Similarly, while Proposition 3.3 shows the consistency of the estimator of \eta_n, it does not address the biasedness issue of this estimator."_**
>
> ➤ We agree. As detailed in the proof of Prop 3.3, the estimation of the rescaling factors $\eta$, $\xi$ involves the estimation of the dual entropic Sinkorn potentials $(f^\star_\varepsilon, g^\star_\varepsilon)$. Estimating the re-weighting functions should not be harder than estimating the EOT plan $\pi^\star_\varepsilon$.
>
> > **_"Is there any reason that the authors have not applied the current framework to image domains? Is it because the bottleneck is due to finite batch sizes as mentioned above in the weakness section?"_**
>
> ➤ This is a good point. We have focused on genomics task because this is where our expertise lies, and because of the exciting FGW applications  [Nitzan+19, Zeira+22, Klein+23, Lange+23]. We believe this is a promising and high-stakes scientific endeavor. Switching to images, as done previously by other authors, would be doable but will very likely require additional compute resources. Batch size will likely be an issue but can be navigated, as done e.g. by [Pooladian+23] or [Tong+23]. We argue, however, that the batch-size issue will likely be less of an issue in our case because we operate in a large $\varepsilon$ regime.
>
> > **_“\hat{Pi} under (7) is undefined”_**
>
> ➤ You mean $\hat{\pi}_\varepsilon$. This is a discrete coupling (i.e. a rescaled $n\times n$ bistochastic matrix) obtained from two families of $n$ samples. We will change notations to highlight more clearly this is a matrix, and not a coupling density on the whole space.

---

> > ### Comment · Reviewer_fozM · 2023-11-18
> > **Reply to authors**
> >
> > Thank you for your thorough rebuttal. I appreciate the two new experiments in Appendix E.2 on EOT between Gaussians and an ablation study of the batch size. The improved texts also give better clarity. On the other hand, I agree with reviewer DBjq on the limited novelty of the current work. As such, I would like to keep my current score.

---

> ### Author Response · Authors · 2023-11-20
> **Many thanks for your comments**
>
> We are very grateful for your time, and thankful for your encouragements regarding new experiments.
>
> We respect your impression that our work has limited novelty following reviewer **DBjq** comments.
>
> We kindly ask, should you have the time to do so, that you read our answer to the first question raised by Reviewer **DBjq**, in which we did our best to clarify this.
>
> While we do, of course, build on such methods, we believe there are significant departures from [Tong+23] (and [Pooladian+23]). We have re-parameterized velocities in **target space**, on **Gaussian noise** to accommodate both spaces of different dimension as well as stochasticity. We would like to emphasize that neither [Tong+23] nor [Pooladian+23] can handle that level of modelling freedom. This opens the door for far more advanced tools (e.g. using FGW solvers). We have added other contributions, e.g. unbalanced marginals. Taken together, we believe the gap is quite large w.r.t those methods.
>
> We agree with you and other reviewers that this was not emphasized sufficiently in the "related works" section, and have started editing this part.

---

### Official Review · Reviewer_NfWc · 2023-10-25

**Soundness:** 3 good
**Presentation:** 3 good
**Contribution:** 3 good
**Rating:** 6
**Confidence:** 3

**Summary:**

This paper introduces a general method for computing neural OT couplings, called GENOT. GENOT can cover any cost function, can incorporate randomness using conditional generative models, can map points across incomparable spaces, and can be applied to the unbalanced problems. GENOT employs Conditional Flow Matching approach to fit the conditional distribution of couplings. The authors evaluated GENOT in various single-cell biology problems.

**Strengths:**

-	The proposed GENOT is a flexible approach	that can address various OT problems.
-	This work is overall well-written.
-	GENOT is well-motivated (Sec 3).

**Weaknesses:**

-	**W1.** This paper lacks a quantitative evaluation regarding whether GENOT is a reasonable approach for estimating OT couplings (Only qualitative samples in Fig 1 and 4).
-	**W2.** This paper lacks a comparative evaluation in Sec 5 with previous approaches for various single-cell biology problems, such as [1, 2] (presented in Introduction Section).

**Questions:**

-	**Q1.** What is the advantages of employing Conditional Flow Matching for modeling conditional generators $T(\cdot | x )$? We could consider directly modeling $T(\cdot | x )$ with neural networks as in [3].
-	**Q2.** Proposition 3.1 assumes $Y \sim \pi_{\epsilon}^{\star} (\cdot | x)$ in Eq 7. In practice, Algorithm 1 adopts the mini-batch estimate for $\hat{\pi}_{\epsilon}$. Can GENOT still recovers Optimal Conditional Generators by taking expectations over mini-batch estimate, e.g. [4]? In this respect, I am curious about the quantitative results for the OT coupling estimates from GENOT for Fig 1 and Fig 4.
-	**Q3.** Which algorithms are used for this mini-batch estimate?


**Reference**

[1] Schiebinger, Geoffrey, et al. "Optimal-transport analysis of single-cell gene expression identifies developmental trajectories in reprogramming." Cell 176.4 (2019): 928-943.
[2] Demetci, Pinar, et al. "SCOT: single-cell multi-omics alignment with optimal transport." Journal of Computational Biology 29.1 (2022): 3-18.
[3] Korotin, Alexander, Daniil Selikhanovych, and Evgeny Burnaev. "Neural optimal transport." ICLR, 2023.
[4] Fatras, Kilian, et al. "Unbalanced minibatch optimal transport; applications to domain adaptation." International Conference on Machine Learning. PMLR, 2021.

---

> ### Author Response · Authors · 2023-11-17
> **Many thanks for your feedback**
>
> > **_“This paper lacks a quantitative evaluation regarding whether GENOT is a reasonable approach for estimating OT couplings (Only qualitative samples in Fig 1 and 4).”_**
>
> ➤ We believe the new experiments that we have added in Appendix E.2, paragraph 2, testing GENOT's ability to learn the known EOT coupling between Gaussians, for several dimensions $d$ and $\varepsilon$ values can provide such an answer.
>
> > **_“This paper lacks a comparative evaluation in Sec 5 with previous approaches for various single-cell biology problems, such as [1, 2] (presented in Introduction Section).”_**
>
> Regarding comparison with **[Schiebinger+2019]** (WOT, i.e. discrete unbalanced OT): because we perform a train-test split for the experiments, and hence it is not possible to directly compare with WOT, as WOT does not allow for out-of-sample estimation (WOT only outputs an in-sample coupling matrix, not a "Monge" map). Another contribution of WOT is the estimation of growth rates of cells, to adjust the marginal distribution. It would be straightforward to include this in the GENOT framework, but not really needed in this paper.
>
> Regarding the comparison with **[Demetci+2022]** (SCOT, i.e. discrete Gromov-Wasserstein OT): **this is indeed the competing method we compare with**, and is denoted by GW-LR, which stands for “Gromov Wasserstein with Linear Regression”. Since SCOT is discrete, as WOT, and does not use parameterized maps, it is exclusively “in-sample”, and cannot be used “out-of-sample”. Therefore, we extend SCOT out-of-sample using simply a vector-valued linear regression. For the fused setting, which was not treated in SCOT, we do the same and denote this baseline by FGW-LR (“Fused Gromov Wasserstein with Linear Regression”) accordingly.
>
> > **_“What is the advantages of employing Conditional Flow Matching for modeling conditional generators ? We could consider directly modeling with neural networks as in [3].”_**
>
> ➤ We model the conditional generators $T(\cdot | \mathbf{x})$ via CFM as this method has proven powerful for fitting very complex distributions while being simulation-free and very easy to train [Lipman+'2023], [Pooladian+'2023], [Tong+'2023;a;b].
> CFM parametrizes the map from one distribution to another implicitly as the solution to an ODE. Inference requires solving this ODE, and samples of the fitted distribution are produced through an iterative and progressive procedure that allows the model to learn complex data distributions effectively. This concept of _progressive refinement_ (as opposed to a _static_, single step GAN-like jumps) likely explains the success of diffusion models.
>
> If we understand your suggestion correctly, fitting each conditional generator $T(\cdot | \mathbf{x})$ using directly a feed-forward neural network, would mean producing a "one-step" procedure to map the noise $\rho$ onto each $\pi_\varepsilon^\star(\cdot | \mathbf{x})$. If one were to remove time dependence, it seems this would be similar to a conditional GAN.
>
> > **_“Proposition 3.1 assumes  in Eq 7. In practice, Algorithm 1 adopts the mini-batch estimate for. Can GENOT still recovers Optimal Conditional Generators by taking expectations“?_**
>
> ➤ We have changed the statement of our Proposition 3.1 in the paper, to make it clear that Proposition 31. is a theoretical justification for the loss and not supporting using mini-batch training.  We have also added a discussion of the use of mini-batches, and mentioned to other reviewers the fact that our goal to learn _entropic_ OT couplings (as opposed to unregularized) makes us fall in the better statistical regimes of entropic OT.
>
> > **_“Which algorithms are used for this mini-batch estimate?”_**
>
> ➤ We briefly introduced all these solvers in the Background Sec. 2.  All our solvers are implemented in OTT-JAX, (https://ott-jax.readthedocs.io/en/latest/index.html).
>
> **GENOT-K** and **U-GENOT-K** use the entropic Kantorovich problem and its unbalanced counterpart, using the Sinkhorn algorithm [Cuturi+’2013] and its unbalanced generalized [Séjourné+’2023].
>
>  **GENOT-(F)GW** and **U-GENOT-(F)GW** use entropic (Fused) Gromov-Wasserstein problem and its unbalanced counterpart, using [Peyré+’2016], and its unbalanced generalized [Séjourné+2022], with the trick outlined by [Scetbon+22] to reduce it to quadratic complexity. We use Anderson acceleration to speed up evaluations in the inner loops of Sinkhorn.

---

> ### Comment · Reviewer_NfWc · 2023-11-19
>
> Thank you for the rebuttal, especially the additional experiments evaluating the estimated OT couplings in Appendix E.2. However, I agree with other reviewers that the novelty of the proposed method is incremental. Particularly, [1] already suggested a method for training Flow Matching model based on the minibatch EOT couplings. I suggest that the authors include [1] in the Related Work Section. Therefore, I would like to keep my current rating.
>
> $ $
>
> [1] Pooladian, Aram-Alexandre, et al. "Multisample flow matching: Straightening flows with minibatch couplings." ICML 2023.

---

> > ### Author Response · Authors · 2023-11-20
> > **Many thanks for your response**
> >
> > We are very grateful for your appreciation of our additional experiments, and for taking the time to answer our rebuttal. On the two points that you raised:
> >
> > ➤ The omission of [Pooladian+'23] in the original draft, as well as in our updated pdf, was an involuntary mistake on our end. The reference was cut out at some point in writing, when driven mostly by space constraints. This important and very recent reference will be added back shortly, and this will be reflected in the updated pdf very soon.
> >
> > ➤ With respect to the incrementality our work, we have addresed that point as well as we could, in our answer to the first question raised by Reviewer **DBjq**. We believe that re-parameterizing the velocities **in target space**, on **Gaussian noise**, can accommodate **both** spaces of different dimension and stochasticity. This provides a lot of modelling freedom and more advanced tools (e.g. using FGW solvers) that _neither [Tong+23] nor [Pooladian+23]_ can handle at the moment. This is a _fundamental_ difference, and we do not know, at this moment, of any other competing method that is able to be stochastic across spaces. In addition, we have also explored flexibility in defining both **cost** function and **unbalanced** properties, making GENOT a practical solution for the applications we target.
> >
> > We thank you again for your time.

---

### Official Review · Reviewer_DBjq · 2023-10-31

**Soundness:** 1 poor
**Presentation:** 3 good
**Contribution:** 2 fair
**Rating:** 5
**Confidence:** 5

**Summary:**

---- REVIEW UPDATE ----

I am negatively surprised that the authors have not added the requested numbers of metric baselines (this should have been very quick and simple). I regret that the authors have failed to dismiss my concerns about this evaluation in the Gaussian case. Overall, I am not confident that the evaluation is done correctly.

Given the fact that there is no real understanding of method's performance in learning entropy-regularized transport and the rest evaluation is done mostly in some biological data which is far from my expertise (and I can not clearly assess the results), I can not vote for the acceptance of the paper. I keep my initial score and increase my confidence.

---- INITIAL SUMMARY ----

In this paper, the authors propose the method to compute entropic optimal transport couplings via a flow-based generative model. The proposed approach addresses different practical challenges met by typical optimal transport methods – it is applicable for arbitrary cost function, allows randomness by considering a stochastic version of optimal transport, can be used to map points across incompatible spaces considering the Gromov-Wasserstein problem and adapts unbalanced optimal transport formulations. The authors propose a two-step algorithm for estimating the continuous conditional entropy-regularized couplings $\pi_{\theta}$ defined as the set of conditional generators $T_{\theta}(\cdot|x)$. The generators are parametrized implicitly as the conditional flow matching model induced by a neural vector field. At each step of the algorithm, the authors estimate discrete conditional couplings between empirical samples of the source and target distributions, and then train a conditional matching model between samples from the source distribution and calculated discrete couplings. The algorithm is generalized for unbalanced settings and fused-GW setup. The evaluation is performed in the synthetic setups and problems from the field of single-cell biology, i.e., modeling development of cells, testing cells' response to different drugs, learning translation between different cells’ modalities.

**Strengths:**

The proposed approach addresses different challenges hampering the application of OT-based approaches in practical tasks. It is evaluated on several tasks in the field of single-cell biology where these challenges appear. The authors show that the proposed approach improves the metrics in comparison to other methods evaluated on the biological task.

**Weaknesses:**

I want to point out important aspects of evaluating this paper. First, the experiments mainly involve specific biology tasks that might require some biological background to fully understand (which I don't have). Second, a lot of experiments and important findings are put in the Appendix, which is about 20+ pages long. Based on my experience reviewing papers in similar conferences, I think these Appendix parts might not get a thorough review (at least from me). So, I can't tell if a big part of the paper contains truly important experiments with real-world applications or if these results are just complex experiments presented as being biologically significant. I hope another reviewer (or the area chair) can look into this part of the paper because it's important for deciding if it should be published. Therefore, I'll focus on evaluating the methodological aspects from a machine learning perspective.

As a whole, the algorithm presented in the paper seems incremental to some extent as it just aggregates the ideas which were already implemented in other papers. Indeed, the idea to distillate the discrete OT solutions using the flow matching was initially proposed by the authors of a conditional flow matching model (Tong et al., 2023). The authors claim that one of their contributions lies in using CFM for solving the GW problem, however, switching between classic OT and GW (Fused-GW) setups in their Algorithm 1 leads only to insignificant changes in computing discrete plans. Another contribution stated by the authors consists in extending the algorithm for solving the unbalanced OT (GW) problems. However, the provided scheme for re-weighting the source and target distributions has something in common with the algorithm provided in a paper (Lübeck et al, 2022).

The algorithm provided by the authors solves the continuous entropy-regularized OT (GW) problem by distilling the solutions of the discrete one. This raises my major concern about the paper because plans estimated on empirical samples are known to be bad approximations of the continuous ones which was also mentioned by the authors (in Section 5.2). The provided theoretical results do not clarify this. Furthermore, their Proposition 1 is a little bit misleading. It states that achieving zero value in the loss (7) one recovers the desired conditional plan. However, it is true when the loss is calculated using the samples from the ground truth conditional plan and not the discrete one as in the proposed algorithm. Thus, even when the loss in this algorithm decreases to zero, it is not clear what is actually being learned. This concern extends to the unbalanced variant of the algorithm since it is based on the balanced one. Here additional questions arise about the discrete approximation of scaling factors. The situation is getting even more complex in the case of a GW problem, since it is non-convex and, as such, may have multiple local minimums. Thus, the optimization objective may notably change at every step depending on the calculated discrete OT plans, which may make the result of the training even more unreliable.

The provided algorithm is assessed in simulated low-dimensional 2D (3D) experiments and several real-world single-cell tasks. These experiments do not demonstrate the nature of the learned solutions. Indeed, the comparison with the ground truth plans is performed only in low dimensions, in biological one the comparison is reduced to marginal distributions. Thus, taking into account my comments from the paragraphs above, it is not evident what it learns in high dimensions. This might impose severe risks in its application to biological tasks (again, I am not an expert in these applications, but this my comment seems fair). In order to show that the proposed algorithm indeed learns an entropic plan in high dimensions, I suggest the authors, for example, evaluate their algorithm on a recent benchmark (Gushchin et al., 2023) which seems to be relevant to the current study at least for the non-GW case.

The authors do not provide code for their method which is important since the provided biological setup seems to be not easy to reproduce.

**Short summary:**
- No clear assessment of the errors caused by discrete approximations of OT in high dimensions is provided (my main concern);
- No code in the supplementary material to reproduce the experiments.
- Not clear significance from the biological point of view (at least for me because I do not have a background in biology).

**Questions:**

- Could you please compare with other neural unbalanced OT methods, e.g., (Yang & Uhler, 2018) and (Lübeck et al., 2022)? Maybe it is already present somewhere and I just missed this part.
- Could you please evaluate your algorithm and compare with other entropy-regularized approaches in high dimensions using a recent benchmark (Gushchin et al., 2023)?
- Could you please mention additional papers on neural OT for solving GW, e.g., (Bunne et al., 2019), an elaborate if they are relevant or not?

**Minor:**

In section 5.1, 5.2 links to the Figures 5.1, 5.2 are incorrect;
The notation GW-LR on Figure 5 should be introduced in the main text.

**References:**

Lübeck, F., Bunne, C., Gut, G., del Castillo, J. S., Pelkmans, L., & Alvarez-Melis, D. (2022, October). Neural Unbalanced Optimal Transport via Cycle-Consistent Semi-Couplings. In NeurIPS 2022 AI for Science: Progress and Promises.

Yang, K. D., & Uhler, C. (2018, September). Scalable Unbalanced Optimal Transport using Generative Adversarial Networks. In International Conference on Learning Representations.

Gushchin, N., Kolesov, A., Mokrov, P., Karpikova, P., Spiridonov, A., Burnaev, E., & Korotin, A. (2023). Building the Bridge of Schr\"odinger: A Continuous Entropic Optimal Transport Benchmark. In Advances in Neural Information Processing Systems, 2023

Bunne, C., Alvarez-Melis, D., Krause, A., & Jegelka, S. (2019, May). Learning generative models across incomparable spaces. In International conference on machine learning (pp. 851-861). PMLR.

---

> ### Author Response · Authors · 2023-11-17
> **Many thanks for your feedback**
>
> We thank the reviewer for their thorough review.
>
> > **_“Indeed, the idea to distillate the discrete OT solutions using the flow matching was initially proposed by the authors of a conditional flow matching model (Tong et al., 2023). The authors claim that one of their contributions lies in using CFM for solving the GW problem, however, switching between classic OT and GW (Fused-GW) setups in their Algorithm 1 leads only to insignificant changes in computing discrete plans.”_**
>
> ➤ It seems you imply that our contribution w.r.t. [Tong+23a;b] is mostly on adding the flexibility to sample from other couplings, i.e. FGW or GW. We believe this misses an important aspect of our contribution: GENOT learns a flow *in the target space*  between noise (sampled in that same space) and observations of the target distribution, _conditionally_ to an instance in the source distribution.
>
> Indeed, in [Tong+23a]’s approach, the vector field $v$ is a parameterized map $t, x \in [0,1]\times \mathbb{R}^p \rightarrow \mathbb{R}^p$,
>
> In our approach, in the most general FGW case, we have *two* dimensions $p$ and $q$, and $v$ is then a map $t, z, x \in [0,1]\times \mathbb{R}^q \times \mathbb{R}^p \rightarrow \mathbb{R}^q$ (we’re simply rephrasing the sentence above Equation (5) in our original submission). We agree that this should be made more clear.
>
> For this reason, as you can see, [Tong+23a,b] _cannot_ be extended _directly_ to the (Fused) GW setting, by simply replacing Sinkhorn couplings to those obtained with (F)GW (e.g. in step 3 of our Algorithm 1). This would simply be a problem of differing dimensions.
>
> We have added details in Section 4, and agree this need to be emphasized further in our paper.
>
> > **_“Another contribution stated by the authors consists in extending the algorithm for solving the unbalanced OT (GW) problems. However, the provided scheme for re-weighting the source and target distributions has something in common with the algorithm provided in a paper (Lübeck et al, 2022).”_**
>
> ➤ Thanks for this great suggestion. All existing unbalanced OT (neural) formulations that we are aware of share the similar idea of penalizing deviations to the original marginal weights. However, we feel there are many important differences with [Lübeck+22] and [Yang+19]: they both rely on a Monge formulation; [Lübeck+22] uses ICNNs, and compute _two_ unbalanced (semi) couplings, while [Yang+19]'s formulation is asymmetric and min-max driven, with no links to unbalanced solvers. We have clarified this in the first paragraph of Section 3.2.
>
> > **_“The algorithm provided by the authors solves the continuous entropy-regularized OT (GW) problem by distilling the solutions of the discrete one. This raises my major concern about the paper because plans estimated on empirical samples are known to be bad approximations of the continuous ones which was also mentioned by the authors (in Section 5.2). The provided theoretical results do not clarify this.”_**
>
> ➤ We agree. We believe, however, that two things should be considered:
>
> First, the alternative is to use independent sampling of the marginals (corresponding to $\varepsilon = \infty$). We believe that even in high-dimensions and limited $n$, it is worth extracting more meaningful couplings from discrete solvers.
>
> Second, the efficiency of discrete solvers is improving, notably for (F)GW solvers. Up to a few years ago, they were widely deemed cubic. There are now reliable off-the-shelf implementations, with suitable costs, that are quadratic. From a purely practical perspective, we don’t even think they should be run to convergence, since we mostly aim at recovering some stochasticity from them, in our sampling step.
>
> We continue this discussion in the updated pdf (below Prop. 3.1)
>
> > **_“Furthermore, their Proposition 1 is a little bit misleading. It states that achieving zero value in the loss (7) one recovers the desired conditional plan. However, it is true when the loss is calculated using the samples from the ground truth conditional plan and not the discrete one as in the proposed algorithm.”_**
>
> ➤ We agree that Proposition 3.1 was not clear, and have rephrased its wording and its context, please see the updated pdf.
>
> > **_“Thus, even when the loss in this algorithm decreases to zero, it is not clear what is actually being learned. This concern extends to the unbalanced variant of the algorithm since it is based on the balanced one. Here additional questions arise about the discrete approximation of scaling factors.”_**
>
> ➤ We agree that recovering the unbalanced scalings is also yet another hurdle. Our intuition, from the Proof of Prop. 3.3, however, is that recovering a “good” scaling factor is not harder than recovering the “good” coupling.

---

> ### Author Response · Authors · 2023-11-17
> **second part of answers**
>
> > **“The situation is getting even more complex in the case of a GW problem, since it is non-convex and, as such, may have multiple local minimums. Thus, the optimization objective may notably change at every step depending on the calculated discrete OT plans, which may make the result of the training even more unreliable.”**
>
> ➤ We agree. We would like to add an important comment: recent theoretical works (e.g. [Rioux+23], added in the updated pdf) point out that the _larger_ entropic regularization, the “more convex” GW problems can become (to the extent that it does become convex above a certain $\varepsilon$).
>
> We therefore think that your intuition that GW couplings cannot be used to guide FM should be nuanced by the fact that we target the _entropic_ GW coupling. In short, recovering, in the FGW case, a coupling $\pi_\varepsilon^\star$ is likely much easier and stable throughout iterations than $\pi_0^\star$, for $\varepsilon$ large enough.
>
> > **_“The provided algorithm is assessed in simulated low-dimensional 2D (3D) experiments and several real-world single-cell tasks. These experiments do not demonstrate the nature of the learned solutions. Indeed, the comparison with the ground truth plans is performed only in low dimensions, in biological one the comparison is reduced to marginal distributions.”_**
>
> ➤ We understand the concerns about the metrics in the single-cell data benchmarks. For **GENOT-K**, we follow the setup of previous neural OT papers which benchmark on single-cell data [Bunne+2022], [Uscidda+2023] which assess the learned Monge map only with respect to the fitting term.
>
> Yet, we would like to highlight that for **GENOT-GW** and **GENOT-FGW**, we _**do assess**_ the optimality of the learned transport plan using the FOSCTTM score (see Appendix C.2 for the definition of that metric), which involves the joint law of the coupling and not just its marginals.
>
> > **_“The authors do not provide code for their method which is important since the provided biological setup seems to be not easy to reproduce.”_**
>
> ➤ We have added code in the appendix. We are aware this is not ideal. We commit to implementing a readily usable GENOT implementation in a leading OT toolbox shortly, and will release code to reproduce experiments in a different, independent, repo.
>
> > **_“No clear assessment of the errors caused by discrete approximations of OT in high dimensions is provided (my main concern);”_**
>
> ➤ We feel this is a problem that all papers that use guidance from discrete solvers to explore a continuous formulation suffer from, including e.g. Genevay / Salimans early work, as well as, more recently, [Uscidda+23,Pooladian+23,Tong+23 a;b]’s approaches.
>
> In fact, we argue in the updated pdf, below Prop. 3.1, that we suffer _less_ from that problem, because our “ground truth” is _precisely_ the _entropic_ coupling. All works above typically start their theoretical justification from $\pi_0^\star$, and use $\pi_\varepsilon^\star$ in practice. Our starting point is $\pi_\varepsilon^\star$.
>
> > ***“Not clear significance from the biological point of view (at least for me because I do not have a background in biology).”***
>
> ➤ We understand that the biological significance is hard to assess without a background in biology. Yet, an entire community in bioinformatics has been pushing for generative and foundation models in general, and neural OT, in particular. These tools are having a very large impact on very important tasks in single cell genomics, as documented by the many references we have included.
>
> To help readers parse this, we have added additional references in the updated pdf, towards reviews or roadmap papers, in Section 5.1 and 5.2, to help the reader parse each of our experiments.
>
> We are open to adding more content if you feel this would help readability.
>
> > **_“Could you please compare with other neural unbalanced OT methods, e.g., (Yang & Uhler, 2018) and (Lübeck et al., 2022)? Maybe it is already present somewhere and I just missed this part.”_**
>
> ➤ We agree, we did not describe in details differences w.r.t [Yang&Uhler+19] and [Lübeck+22]. We have added a discussion to that end in Section 3.2.
>
> > **_“Could you please evaluate your algorithm and compare with other entropy-regularized approaches in high dimensions using a recent benchmark (Gushchin et al., 2023)?”_**
>
> ➤ We agree with the reviewer that it is relevant to measure GENOT's ability to recover EOT couplings that are known analytically. Due to time constraints, we are not able to evaluate GENOT on the very recent [Guschin+’2023;a]’s benchmark. Instead, we evaluate GENOT's ability to learn the known EOT coupling between Gaussians [Janati+’2020], whose original code we use. We considered several dimensions $d$ and $\varepsilon$ values, as done in [Mokrov+2023], [Guschin+’2023;b], or [Shi’+2023]. These results paint GENOT under a favorable light.
>
> We have **added these results to Appendix E2, paragraph**.

---

> ### Author Response · Authors · 2023-11-17
> **third part of answers**
>
> > **_“Could you please mention additional papers on neural OT for solving GW, e.g., (Bunne et al., 2019), an elaborate if they are relevant or not?”_**
>
> ➤ [Bunne+2019] is an important reference, but it does not solve neural GW. Much like the original Wasserstein GAN paper, which does _not_ output a neural OT map, it “only” uses GW as a _distributional_ loss between heterogeneous distributions (pushforwarded Gaussian $T\sharp \mathcal{N}$ and data $\nu$, in 2 different spaces). This does not result in a “GW optimal map” (one that would be, ideally, such that $c_{\mathcal{Y}}(T(x),T(x’)) \approx c_{\mathcal{X}}(x),x’)$ in addition to satisfying a push-forward constraint).
>
> > **_“In section 5.1, 5.2 links to the Figures 5.1, 5.2 are incorrect”_**
>
> ➤ We thank the reviewer for pointing out this typo.
>
> > **_“The notation GW-LR on Figure 5 should be introduced in the main text.”_**
>
> ➤ We agree on that point, so we have made the change. We thank the reviewer for this suggestion.

---

> > ### Comment · Reviewer_DBjq · 2023-11-20
> > **Response**
> >
> > I have read the authors' response. It clarifies some minor aspects and explains the differences with (Tong et.al., 2023). Nevertheless, my main concern still remains quite open: to which extent discrete OT solutions are good?
> >
> > I see that the authors have done some evaluation on the Gaussians and comparison with some methods which demonstrate that their method performs comparably or better. But I have some questions regarding the metric used and their experimental setup.
> >
> > - Why do the authors multiply the covariance matrices of input distributions by 0.01? This seems quite weird. Do the other papers also do this?
> >
> > - As the metric, the authors use the Sinkhorn divergence between plans and perform relative comparison with the other methods. I do not understand whether the metric values are good (all the methods work well to some extent) or they are bad (the all methods fail in high dimensions). Could this be further clarified? For example, what is the Sinkhorn divergence between the independent plan and the true plan for reference? This could help to understand some baseline value.
> >
> > - To my knowledge, sample estimates of Sinkhorn are slightly biased although have good sample complexity. To which extent this is an issue for this metric, especially in dimensions such as 256? For example, what is the sinkhorn divergence between the two random subsamples from the true plan? Could this and the previous number be added to the paper?
> >
> > Overall, it seems like this is one of the first works proposing ways to "distill the discrete OT" instead of optimizing some Schrodinger Bridge-based or dual-like entropic objectives. Because of this, in my opinion, it is essential to clearly understand how valid is this strategy (especially because most of experiments are on bilogical data where the precision might be crucial), and there must be a transparent compasion with as much prior methods as possible. The fact that the authors avoided the entropic benchmark suggested by the two reviewers seems weird as using it could be helpful to immediately answer the raised question. I kindly ask to clarify my questions above.

---

> ### Author Response · Authors · 2023-11-20
> **Thanks for your detailed answer**
>
> We are very grateful for your time reading our paper and our rebuttal, thanks for coming back to us and for highlighting several areas for improvements.
>
> Here are a few additional answers to your questions, as you kindly requested:
>
> > **_to which extent discrete OT solutions are good?_**
>
> We believe your question can be split in two parts:
> - what is the right model to learn flows? is it deterministic or conditional? _(model)_
> - given that choice, is OT useful to guide the estimation of such flows? _(loss)_
>
> _Our_ answer to these two questions is “Using a conditional model to map $x$, $P( . |x)$, and _not_ a deterministic $T(x)$ (integrated à la Benamou-Brenier), we believe that it makes sense to use an entropic OT guided coupling approach”.
>
> We _do not_ take a stand as far as _other_ applications of mini-batch OT solutions are concerned ([Tong+23] or [Pooladian+23]). In fact, GENOT uses “standard” (independent, not OT guided) flow matching, when learning to match white noise $z$ to samples $y_1, … y_k$ _conditionally_ to a point $x$ (the points $y_1, .. , y_k$ are matched to $x$ using entropic OT), using notations from algorithm 1.
>
> > **_”Why do the authors multiply the covariance matrices of input distributions by 0.01? This seems quite weird. Do the other papers also do this?”_**
>
> We generate a pair of source and target centered Gaussians, by sampling them from a **Wishart** distribution, with $d$ degrees of freedom. The scaling matrix we use (using wikipedia notations, https://en.wikipedia.org/wiki/Wishart_distribution) is $\mathbf{V} = 0.01 \cdot \mathbf{I}_d$. This results in matrix samples that have expectation $d\mathbf{V}=0.01 \times d \times \mathbf{I_d}$, which is therefore close to $\mathbf{I}_d$ with our range of $d$. We have switched from notation $\Sigma$ to $\mathbf{V}$ in the appendix to make this easier for the reader.
>
> > **_”As the metric, the authors use the Sinkhorn divergence between plans and perform relative comparison with the other methods. I do not understand whether the metric values are good (all the methods work well to some extent) or they are bad (the all methods fail in high dimensions). Could this be further clarified? For example, what is the Sinkhorn divergence between the independent plan and the true plan for reference? This could help to understand some baseline value._”**
>
> We are working towards adding these numbers. We will do our best to get back to you before the end of the rebuttal period, and will follow your suggestion if we cannot make it in time. We agree this can be helpful to assess the scale of the difference (but not the conclusion, which is that our method remains competitive).
>
> > **_”To my knowledge, sample estimates of Sinkhorn are slightly biased although have good sample complexity. To which extent this is an issue for this metric, especially in dimensions such as 256? For example, what is the sinkhorn divergence between the two random subsamples from the true plan? Could this and the previous number be added to the paper?”_
>
> [Mokrov+23] (Energy guided OT, submitted to ICLR) and [Guchin+23] (Neural entropic OT with diffusion processes, NeurIPS 23) have used the FID score (i.e. W2 between Gaussian approximations) to assess fit with Gaussians, we will try to add these results to our revision. There is no silver bullet, as FID is also fairly restrictive in what it captures, but we agree with your suggestion that more metrics (e.g. MMD) are always welcome.
>
> > **_Overall, it seems like this is one of the first works proposing ways to "distill the discrete OT" instead of optimizing some Schrodinger Bridge-based or dual-like entropic objectives._**
>
> We would credit [Tong+23] or [Pooladian+23] with the idea of distillating _discrete OT_, but they distill this into a _map_. Our contribution is to distill _discrete **entropic** OT_ using a conditional (stochastic) approach.
>
> > **_Because of this, in my opinion, it is essential to clearly understand how valid is this strategy (especially because most of experiments are on bilogical data where the precision might be crucial), and there must be a transparent compasion with as much prior methods as possible._**
>
> For most biologically driven problems, there is no ground truth available (except for the FOSCTTM task). In that context we believe that our experiments highlight that entropic OT is mostly useful to 1. have better robustness 2. provide uncertainty quantification for our predictions.

---

> ### Author Response · Authors · 2023-11-20
> **Comment on benchmark**
>
> > **_The fact that the authors avoided the entropic benchmark suggested by the two reviewers seems weird as using it could be helpful to immediately answer the raised question._**
>
> That benchmark, much like the 2019 Neural OT benchmark (https://arxiv.org/abs/2106.01954)  from the same team, is a very nice contribution for the community.
>
> As end-users, we feel, however, that adopting this new benchmark in the short timespan of the rebuttal is quite difficult.
>
> For instance, a few hours ago, in a previous version of this response (that we changed shortly after, using the edit function), we mentioned that we had struggled with a bug in the entropic Gaussians code of that benchmark; After further examination, we are less sure; this misunderstanding might be due to a different parameterization (compared to [Janati+20]). However, we are still investigating this aspect, because we still recover negative eigenvalues for covariance matrices with that codebase. We will update here with the result of our investigation.
>
> To summarize, we believe the benchmark is still very new (it was released over the summer, right before the ICLR deadline), and was not used, for instance, in these recent submissions from a similar team.
>
> https://arxiv.org/pdf/2304.06094.pdf energy guided OT
> https://arxiv.org/pdf/2211.01156.pdf entropic neural OT via diffusion processes
>
> We will do our best to interface this (pytorch) very promising benchmark with our codebase (jax) in coming weeks, but this will require further work.
>
> > _**I kindly ask to clarify my questions above.**_
>
> We hope these answers are satisfactory, and are of course happy to continue this conversation if our answers are not clear. We would like to express our appreciation for your time, and for your many constructive comments.

---

### Official Review · Reviewer_WAiP · 2023-11-06

**Soundness:** 3 good
**Presentation:** 3 good
**Contribution:** 3 good
**Rating:** 8
**Confidence:** 4

**Summary:**

The authors consider different versions of entropic OT problem statement: unbalanced OT formulation, quadratic OT instead of linear OT, etc. They proved that these problems can be solve using the conditional flow matching framework. They proposed corresponding computational algorithms and tested them in single-cell biology problems.

**Strengths:**

- a unified solution framework for different OT problem statements, important for practical cases

- interesting computational experiments on practical tasks in the field of single-cell biology, where challenges of standard OT problems (unbalanced OT formulation, quadratic OT instead of linear OT, etc.) appear

**Weaknesses:**

- the authors did not investigate computational limits of the proposed approach: how do comp. efficiency and accuracy scale w.r.t. dimensionality? sample size?

- the code is not provided. However, the experimental protocol for working with biological data looks very complex. So it is not easy to replicate the results of the research

- it seems the approach will not work in case of image data. Any ideas how it can be adapted for this case?

- limited number of baselines were considered to verify the performance of the proposed approach on biomedical data

- In proposition 1 the authors claim that using the proposed algorithm we can recover the desired conditional plan. However, this proposition assumes that the loss is calculated using the samples from the ground truth conditional plan. In the proposed algorithm the authors use the samples from the discrete conditional plan. So it is not clear whether based on such statistical estimate we really converge to the true solution of the corresponding continuous problem statement, especially in high-dimensional case

- from a practical viewpoint it is difficult for a non-specialist to verify results of computational experiments in biological domain, whether they are significant or not. Moreover, in biological experiments the authors uses only marginal distributions to estimates accuracy of their results. In the comment above I articulate that it is not clear to which OT plan actually the solution, delivered by the proposed algorithm, converges. Thus, it is important to estimate accuracy of the solution based on high-dimensional statistics (not only marginal distributions). I understand that in biological use cases we do not have such possibility.

However, there exist several benchmarks for OT, e.g. https://arxiv.org/abs/2306.10161 (Building the Bridge of Schrödinger: A Continuous Entropic Optimal Transport Benchmark) and https://arxiv.org/abs/2106.01954 (Do Neural Optimal Transport Solvers Work? A Continuous Wasserstein-2 Benchmark)

I wonder whether efficiency of the proposed algorithm can somehow be evaluated using high-dimensional test use cases from one of these benchmarks?

Answers to comments above are very important for me to finally assess the paper.

**Questions:**

- page 30. What is "... choose alpha = 0.7m=, but ..."?

---

> ### Author Response · Authors · 2023-11-17
> **Many thanks for your feedback.**
>
> Please have a look at our answers:
>
> > **_the authors did not investigate computational limits of the proposed approach: how do comp. efficiency and accuracy scale w.r.t. dimensionality? sample size?_**
>
> ➤ We parse your question in two different blocks: compute complexity and statistical efficiency. We discuss statistical / minibatch aspects later in the response.
>
> Just like other coupling-guided methods, estimating GENOT requires at each iteration a call to a discrete solver, to sample coupled points from two marginals. We have explored in this work (un)balanced {Kantorovich / linear, GW / quadratic, FusedGW / linear+quadratic} settings. These solvers’ complexities are all well understood, and not the crux of our contribution:
>
> The Kantorovich problem is solved using the unbalanced [Frogner+15] formulation of the Sinkhorn algorithm *[Cuturi13]*, while the GW/fused uses *[Séjourné+22]*, itself an extension of a scheme proposed in *[Peyre+16]*
>
> - For $n$ samples from two measures, and a cost function $c : \mathbb{R}^d \times  \mathbb{R}^d  \to \mathbb{R}$, Sinkhorn has quadratic $\mathcal{O}(n^2)$ time complexity. This does prevent using *very* large batches (above e.g. >20k). We just focused our efforts on running experiments with $n = 1024$. We could imagine exploring other low-rank alternatives, which are known to scale better, but this would change the overall messaging of our paper (since we emphasize entropy).
>
> - For $n$ samples, the complexity of the implementations we use (e.g. in OTT-JAX) of (F)GW will depend on the intra-domain cost functions $c_\mathcal{X}$ and $c_\mathcal{Y}$, as well as extra-domain cost for fused. As shown in the first section of [Scetbon+22], this complexity is also quadratic if all of these cost functions have a low rank decomposition (which is typically the case in large data regime, where $n>d$, and $d$ is the dimensionality of points in $\mathcal{X}$ or $\mathcal{Y}$). This notably includes the inner product and the squared Euclidean costs.
>
> We added a very short comment in line 3 of the Algorithm on p.16. We can add a more detailed discussion in the appendix, along the lines of the paragraph above, if the reviewer thinks this would be helpful.
>
>
> > **_“the code is not provided. However, the experimental protocol for working with biological data looks very complex. So it is not easy to replicate the results of the research”_**
>
> ➤ This is a good point. We have shared code in the appendix of the updated pdf. We are aware this is not ideal. We commit to implementing a readily usable GENOT implementation in a leading OT toolbox shortly, and will release code to reproduce experiments in a different, independent, repo.
>
> > **_“it seems the approach will not work in case of image data. Any ideas how it can be adapted for this case?”_**
>
> ➤ First, we believe that an important innovation of GENOT is the ability to map across spaces. To showcase this convincingly with images, we would therefore need to consider *another* modality (aside from images), e.g. texts.
>
> In a simpler image to image (same dimension) task, because FM ([Lipman+23]), as well as "coupling-guided" CFM ([Tong+23a, Pooladian+23] have proved useful, we expect that getting convincing results from GENOT would be possible, but would require significant compute. The expertise of our team being in genomics, and there being a strong need for such OT methods in that area, we have preferred to focus extensively on genomics tasks.
>
> > **_“limited number of baselines were considered to verify the performance of the proposed approach on biomedical data”_**
>
> ➤ We benchmark the performance of **GENOT-K** against competing methods in Appendix E2. For **GENOT-GW** and **GENOT-FGW** there is, to the best of our knowledge, no competitor, as we are not aware of any other method which computes neural entropic GW or neural entropic FGW. While there exists a neural Gromov **Monge** map estimator [Nekrashevich23], there is no code provided. Hence, we decided to compare **GENOT-GW** and **GENOT-FGW** in the discrete case, with out-of-sample-estimation based on linear regression.

---

> ### Author Response · Authors · 2023-11-17
> **Second part of answers**
>
> > **_“In proposition 1 the authors claim that using the proposed algorithm we can recover the desired conditional plan. However, this proposition assumes that the loss is calculated using the samples from the ground truth conditional plan. In the proposed algorithm the authors use the samples from the discrete conditional plan. So it is not clear whether based on such statistical estimate we really converge to the true solution of the corresponding continuous problem statement, especially in high-dimensional case”_**
>
> ➤ Thanks for this remark. We have clarified in the updated pdf that Proposition 3.1 is only a _theoretical justification for our loss_, not a justification of our algorithm. We have substantially changed the language in that part.
>
> In particular, we introduced a _new_ discussion on why, while not being a panacea, our mini-batch approach is _better_ suited to the goal of estimating *entropic* couplings, as compared to estimating the _unregularized_ “cursed-by-dimension-true-Monge-OT-map” (which constitutes the bulk of the literature so far).
>
> > **_“from a practical viewpoint it is difficult for a non-specialist to verify results of computational experiments in biological domain, whether they are significant or not.”_**
>
> ➤ We understand that the biological significance is hard to assess without more background knowledge on these problems. We have added additional references in the updated pdf, towards reviews or roadmap papers, in Section 5.1 and 5.2, to help the reader parse each of our experiments. We are open to adding more content if you feel this would help readability.
>
> > **_”Moreover, in biological experiments the authors uses only marginal distributions to estimates accuracy of their results. In the comment above I articulate that it is not clear to which OT plan actually the solution, delivered by the proposed algorithm, converges. Thus, it is important to estimate accuracy of the solution based on high-dimensional statistics (not only marginal distributions). I understand that in biological use cases we do not have such possibility.”_**
>
> ➤ We understand the concerns about the metrics in the single-cell data benchmarks. For **GENOT-K**, we follow the setup of previous neural OT papers which benchmark on single-cell data  [Bunne+2022, Uscidda+2023] which assess the learned Monge map only with respect to the fitting term. Yet, we would like to highlight that for GENOT-GW and GENOT-FGW we do assess the optimality of the learnt transport plan: **The FOSCTTM score assesses how close a cell is to its true match** and was shown in **Figure 5** in our original submission (now Figure 6).
>
> > ***“I wonder whether efficiency of the proposed algorithm can somehow be evaluated using high-dimensional test use cases from one of these benchmarks?”***
>
> ➤ We agree with the reviewer that it is relevant to measure GENOT's ability to recover EOT couplings that are known analytically. Due to time constraints, we are not able to evaluate GENOT on the very recent [Guschin+’2023;a]’s benchmark. Instead, we evaluate GENOT's ability to learn the known EOT coupling between Gaussians [Janati+’2020]. We use [Janati+’2020]'s original code to create this ground-truth. We considered several dimensions $d$ and $\varepsilon$ values, as done in [Mokrov+2023], [Guschin+’2023;b], or [Shi’+2023]. Please look at these results in **Appendix E2, paragraph 2** of the updated pdf.
>
> Note that the second benchmark that you suggested [Korotin'+2021] can only be used to evaluate models estimating deterministic OT maps (a.k.a. Monge maps) in the squared-Euclidean cost case. This is not so relevant for our probabilistic maps, in the entropy-regularized setup.
>
> >***“page 30. What is "... choose alpha = 0.7m=, but ..."?”***
>
> ➤Thanks for pointing out this typo.

---

### Author Response · Authors · 2023-11-17
**General response to all reviewers, and AC**

We would like to thank all reviewers for their encouraging feedback, constructive criticism, and thoughtful comments, as well as for pointing out typos.

We highlight in this general response the changes and additions, following reviewers' comments, that we have incorporated in the updated manuscript. These modifications/additions are highlighted in red. Due to time constraints, the manuscript is currently slightly longer than 9 pages. We will of course bring it back to 9 pages as time allows.

**New experiments added to the manuscript**

➤ As suggested by reviewers **WAiP**, **DBjq**, **NfWc**, and **fozM**, we measure GENOT’s ability to recover EOT couplings that are known analytically, focusing on higher dimensions. We evaluate GENOT's ability to learn the known EOT coupling, for the squared Euclidean cost, between Gaussians [Janati+’2020], as done in [Mokrov+2023], [Guschin+’2023;b], or [Shi’+2023]. We use original  [Janati+’2020]’s code and consider dimensions $d \in \{2, 8, 32, 128, 256\}$ and $\varepsilon \in \{0.1, 1, 10\}$. Results are shown in appendix E2 of the updated pdf. They show the competitiveness of GENOT, especially in large dimensions $d$, where it slightly outperforms all the baselines, for each $\varepsilon$ value. These results echo the **new** discussion on mini-batches and biases that we have added in the manuscript Sec. 3, and that we refer to below, in individual responses.

➤ Furthermore, as proposed by reviewer **fozM**, we also investigate the influence of the batch size $n$. Using the same Gaussian setup, we report the performances provided by multiple batch sizes $n \in \{4, 16, 64, 128, 512, 1024, 2048\}$. As expected, in the Gaussian case, larger mini-batches do help. We plan to expand these results in the near future to more challenging setups (e.g. GW) and complex distributions.

**Other changes to the manuscript**

➤ As requested by reviewers **WAiP** and **DBjq**, we provide code to run GENOT in appendix G4. While the first two paragraphs contain the implementation for the neural network architectures and the GENOT model, the third paragraph reproduces the experiment to generate figure 2, and thus demonstrates how to use GENOT-K. The fourth paragraph contains code to reproduce the results of figure 6 (now figure 5), and hence shows how to use GENOT-FGW. We are aware this is a far from ideal way to share code. We are going to implement GENOT in a leading OT toolbox, and will release code to reproduce experiments in a different repository.

➤ Following the comments of reviewers **WAiP**, **DBjq**, **NfWc**, and **fozM**, we have rephrased Proposition 3.1, and added context around it. We also added a discussion about the statement in the paper. In particular, we highlight that Proposition 3.1 is only a _theoretical justification for our loss_, not a justification of our algorithm.  Moreover, we introduced a _new_ discussion on why using minibatches to estimate  (F)GW or linear OT couplings might be less problematic to estimate **continuous entropic** OT than it is to estimate continuous deterministic Monge maps (which represents the bulk of proposed neural OT estimators).

---

### Meta-Review · Area_Chair_5GAV · 2023-12-08

**Metareview:**

This paper introduces a general method for computing neural OT couplings. The proposed
method can be considered in large situations including incomparable spaces and unbalanced problems imbalanced
spaces. The paper provides theoretical supports to recovery of the optimal entropic coupling as well as experiments
showing the benefits of the approach for biological datasets.


The paper have raised discussions among reviewers and the final points are that there is still several
important issues that need to be solved to make the paper supports its claims. Among those points, one can
mention
* the need for convincing experiments in high-dimension (and eventually beyond Gaussians)
* consider other types of data (image + text for instance)
* address the mini-batch issue
* provide clear experiments demonstrating that the method learns correct OT/GW solution with sufficiently small error.

**Justification For Why Not Higher Score:**

the paper still have some issues to address in order to support its claims

**Justification For Why Not Lower Score:**

na

---

### Decision · Program_Chairs · 2024-01-16

Reject